# CEACAM1 as a mediator of B-cell receptor signaling in mantle cell lymphoma

Serene Xavier [1,16], Vivian Nguyen[1,16], Vishal Khairnar [1,2,16], An Phan[1,16], Lu Yang [1,16], Michael S. Nelson [3], Ravi P. Shukla[4], Jinhui Wang[5], Aimin Li[6], Huimin Geng[7], Jaewoong Lee[8,9,10], Teresa Sadras[11], Lan V. Pham[12], Dennis D. Weisenburger[13], Wing C. Chan [6], Karl S. Lang[14], Geoffrey P. Shouse [15], Alexey V. Danilov [15], Joo Y. Song[6], Samir Parekh [4], Markus Müschen [8,17] & Vu N. Ngo [1,17] ✉

B-cell receptor (BCR) signaling plays an important role in the pathogenesis of mantle cell lymphoma (MCL), but the detailed mechanisms are not fully understood. In this study, through a genome-wide loss-of-function screen, we identify carcinoembryonic antigen-related cell adhesion molecule 1 (CEACAM1) as an essential factor in a subset of MCL tumors. Our signal transduction studies reveal that CEACAM1 plays a critical role in BCR activation through involvement in two dynamic processes. First, following BCR engagement, CEACAM1 co-localizes to the membrane microdomains (lipid rafts) by anchoring to the F-actin cytoskeleton through the adaptor protein filamin A. Second, CEACAM1 recruits and increases the abundance of SYK in the BCR complex leading to BCR activation. These activities of CEACAM1 require its cytoplasmic tail and the N-terminal ectodomain. Considering that previous studies have extensively characterized CEACAM1 as an ITIM-bearing inhibitory receptor, our findings regarding its activating role are both surprising and context-dependent, which may have implications for BCR-targeting therapies.

Mantle cell lymphoma (MCL) arises primarily from transformed pre-germinal center (GC) or GC-experienced B-cells, the vast majority of which carry the t(11;14)(q13;q32) with oncogenic activation of cyclin D1 (*CCND1*). Pathological B-cell receptor (BCR) signaling represents an important driver of malignant transformation in MCL, which is evident from the stereotyped, biased immunoglobulin heavy chain variable region gene (IGHV) repertoire in most MCL cases[1], increased expression of BCR signaling components[2,3], and strong, albeit short-lived, clinical responses to ibrutinib, which targets the BCR-downstream Bruton's tyrosine kinase (BTK)[4,5]. With the introduction of ibrutinib, the overall survival of patients has been extended by 5–8 years[6], but MCL remains an incurable disease. Mechanisms underlying BCR

[1]Department of Systems Biology, City of Hope Comprehensive Cancer Center, Monrovia, CA, USA. [2]Department of Medical Oncology, Dana-Farber Cancer Institute, Harvard Medical School, Boston, MA, USA. [3]Light Microscopy and Digital Imaging Core, Beckman Research Institute, City of Hope, Duarte, CA, USA. [4]Icahn School of Medicine at Mount Sinai, New York, NY, USA. [5]Integrative Genomics Core, Beckman Research Institute, City of Hope, Duarte, CA, USA. [6]Department of Pathology, City of Hope Medical Center, Duarte, CA, USA. [7]Department of Laboratory Medicine, University of California, San Francisco, San Francisco, CA, USA. [8]Center of Molecular and Cellular Oncology, Yale Cancer Center, Yale University, New Haven, CT, USA. [9]School of Biosystems and Biomedical Sciences, College of Health Science, Korea University, Seoul, Korea. [10]Interdisciplinary Program in Precision Public Health, Korea University, Seoul, Korea. [11]Olivia Newton-John Cancer Research Institute, Heidelberg, Melbourne, VIC, Australia. [12]Oncology Discovery, Abbvie Inc., South San Francisco, CA, USA. [13]Department of Pathology, Microbiology, and Immunology, University of Nebraska Medical Center, Omaha, NE, USA. [14]Institute of Immunology, University Hospital Essen, Essen, Germany. [15]Department of Hematology and Hematopoietic Cell Transplantation, City of Hope Medical Center, Duarte, CA, USA. [16]These authors contributed equally: Serene Xavier, Vivian Nguyen, Vishal Khairnar, An Phan, Lu Yang. [17]These authors jointly supervised this work: Markus Müschen, and Vu N. Ngo. ✉e-mail: vngo@coh.org

hyperactivity have been established in diffuse large B-cell lymphoma (DLBCL) such as activating mutations in components of the BCR pathway[7–9], and in B-cell chronic lymphocytic leukemia (B-CLL) with self-recognition of the BCR[10]. However, little is known about the mechanisms driving aberrant BCR signaling in MCL.

Carcinoembryonic antigen-related cell adhesion molecule 1 (CEACAM1) is a member of the carcinoembryonic antigen (CEA) family of transmembrane glycoproteins and is expressed on lymphocytes, natural killer (NK) cells, granulocytes, epithelial, and certain endothelial cells[11,12]. CEACAM1 is structurally composed of four immunoglobulin (Ig)-like extracellular domains—an N-terminal IgV-like domain, followed by three IgC2-like domains—along with a single-pass transmembrane region and a cytoplasmic domain that exists in two isoforms: the long isoform (CEACAM1-L), which contains two immunoreceptor tyrosine-based inhibitory motifs (ITIMs), and the short isoform (CEACAM1-S), which lacks ITIMs[13]. The longest isoform, CEACAM1-4L, includes all four Ig-like extracellular domains and a long cytoplasmic tail. While the N-terminal domain of CEACAM1 typically mediates trans-homophilic interactions, the two ITIMs on its cytoplasmic, a unique feature among the CEACAM family, regulate intracellular signaling. ITIMs are signaling domains that recruit the inhibitory phosphatases, src homology 2 (SH-2) domain-containing protein tyrosine phosphatase 1 (SHP-1) and SH-2 containing inositol 5′ polyphosphatase 1 (SHIP1), which dephosphorylate BCR and T-cell receptor (TCR) pathway substrates[14]. Of the 109 ITIM-bearing inhibitory receptors in the human genome, normal B cells express CEACAM1, CD22, CD72, paired immunoglobulin-like type 2 receptor beta (PILRB), and the Fc fragment of IgG receptor IIb (FCGR2B). Mice lacking any of the last four of these receptors develop autoimmune disease and B-lymphoproliferation driven by hyperactive BCR signaling[15–18]. However, despite substantial CEACAM1 expression levels on normal B-cells, *Ceacam1*-deficient mice do not develop autoimmune disease; instead, they have significantly reduced rather than increased numbers of B cells[19,20]. These studies suggest that CEACAM1 is functionally distinct from the other ITIM-containing inhibitory receptors on B-cells.

BCR signaling occurs within lipid rafts, which are submicroscopic plasma membrane domains that are rich in sphingolipids, cholesterol, and lipid-anchored proteins, including the SRC family kinases LYN, FYN, and BLK[21,22]. Antigen engagement leads to recruitment of the BCR to these membrane microdomains, which then quickly coalesce into larger membrane raft regions[23,24]. Clustering of the receptors within the lipid rafts brings them close to the raft-resident kinases, resulting in phosphorylation of two immunoreceptor tyrosine-based activation motifs (ITAMs) in the CD79A and CD79B signal chains of the BCR complex and initiation of downstream signaling cascades[25]. Coalescence of lipid rafts in response to BCR crosslinking is critical for efficient and sustained signaling. Yet, the way in which lipid rafts are stabilized and contribute to hyperactive BCR signaling in MCL remains unclear.

In this study, we identified CEACAM1 as a key factor that supports BCR signaling in MCL through a comprehensive approach combining genome-wide CRISPR screens, gene expression profiling, and BCR signal transduction studies. Our data shed light on the positive role of CEACAM1 in facilitating BCR signaling in MCL through bolstering lipid-raft functions and orchestrating the actions of signaling components. These findings also provide mechanistic insights into the dual role of CEACAM1 in the transmission of both activating and inhibitory signals at the BCR complex, depending on the cellular context and coordinated recruitment of signaling molecules.

## Results
### CRISPR screen identifies CEACAM1 as an essential regulator in MCL
To identify molecular targets important for MCL proliferation and survival, we performed an unbiased genome-wide CRISPR library screen[26]. The MCL cell line JEKO-1 was engineered for doxycycline-inducible expression of Cas9 and transduced with a pooled lentiviral CRISPR library of single-guide (sg) RNAs in two replicates. For sufficient coverage of the complex library, we scaled up the library transduction to ~500 cells per sgRNA. After transduction and puromycin selection for stable integrants, Cas9 expression was induced for 14 days. Genomic DNA was harvested on days 0 and 14 and subjected to deep sequencing to determine the relative abundance, i.e., depletion or enrichment, of each sgRNA between the two time points. While there was little difference in the abundance of many non-targeting-control sgRNAs between days 0 and 14, sgRNAs targeting genes known to be essential in MCL, such as CCND1[27], SOX11[28,29], genes in the BCR and NFκB pathways, or pan-dependent genes in many cancers were substantially depleted by at least twofold on day 14 (Fig. 1a, b, Supplementary Fig. 1a, b, and Supplementary Data 1). To identify additional candidates for further functional analysis, we developed a four-tier filtering strategy. To focus on genes that are specifically expressed in MCL, we first selected for genes that were substantially upregulated in MCL compared to pre-GC B-cells[30] (Z ratio > 0, P < 0.05). In a second step, we selected all genes that were depleted by at least twofold (P < 0.01) over the course of the 14-day interval. To orient our search toward genes that could be potential drug targets, we then removed 94 pan-dependent genes of the selected 435 genes because they were identified as essential genes in a broad array of cancer cell lines based on a CRISPR screen using the same library[31]. Analyzing clinical outcome and gene expression data from two studies of patients with MCL[32,33], we identified clinical outcome predictors in a final filtering step (Supplementary Data 2). To this end, we segregated the clinical cohorts into two equally strong subgroups based on higher vs lower than median mRNA levels for each of the remaining 341 genes. Twenty-one of these genes had a significant association with shorter overall survival of MCL patients in these studies (Fig. 1a and Supplementary Data 3). Among the 21 candidates, CEACAM1 had the highest number of depleted sgRNAs (Fig. 1b) and since its function was previously implicated in BCR signaling[20,34,35], we selected this molecule for further study.

We next conducted validation experiments by introducing two CEACAM1-specific sgRNAs identified from the screen into Cas9-expressing JEKO-1 cells. Our findings confirmed that depletion of CEACAM1 impaired cell viability by enhancing apoptosis and reducing S-phase progression (Fig. 1c, d and Supplementary Fig. 2). To confirm that the observed reduction in cell viability was not a result of unintended CRISPR-mediated effects on neighboring genes, such as CD79A, which is located at the 19q13.2 locus near CEACAM1, or on IgM expression, we compared CD79A and IgM expression between control and CEACAM1 sgRNA-transduced JEKO-1 cells. Our analysis revealed no significant difference in CD79A and IgM expression using flow cytometry and immunoblotting detection assays (Supplementary Fig. 3a–c). These results suggest that the reduced cell viability seen in Fig. 1d was not caused by an accidental loss of CD79A expression due to CRISPR-mediated CEACAM1 depletion. Moreover, we investigated whether the impact of CEACAM1 knockdown on JEKO-1 cell viability was linked to the disruption of β-catenin signaling. Previous studies had shown the importance of WNT-β-catenin signaling in MCL[36–38], and it was known that β-catenin interacts with CEACAM1[39]. However, our results indicated that both control and CEACAM1-knockout JEKO-1 cells exhibited β-catenin stabilization upon IgM stimulation (Supplementary Fig. 3d). This stabilization was consistent with increased phosphorylation at the serine 9 residue on GSK3β, leading to the inactivation of this negative regulator of β-catenin (Supplementary Fig. 3d). These findings strongly suggested that β-catenin signaling was independent of CEACAM1 in this context. Altogether, our results indicate that reduced cell viability in CEACAM1-depleted JEKO-1 cells was not due to unintended CRISPR-related loss of CD79A or alterations in IgM expression, as well as β-catenin signaling.

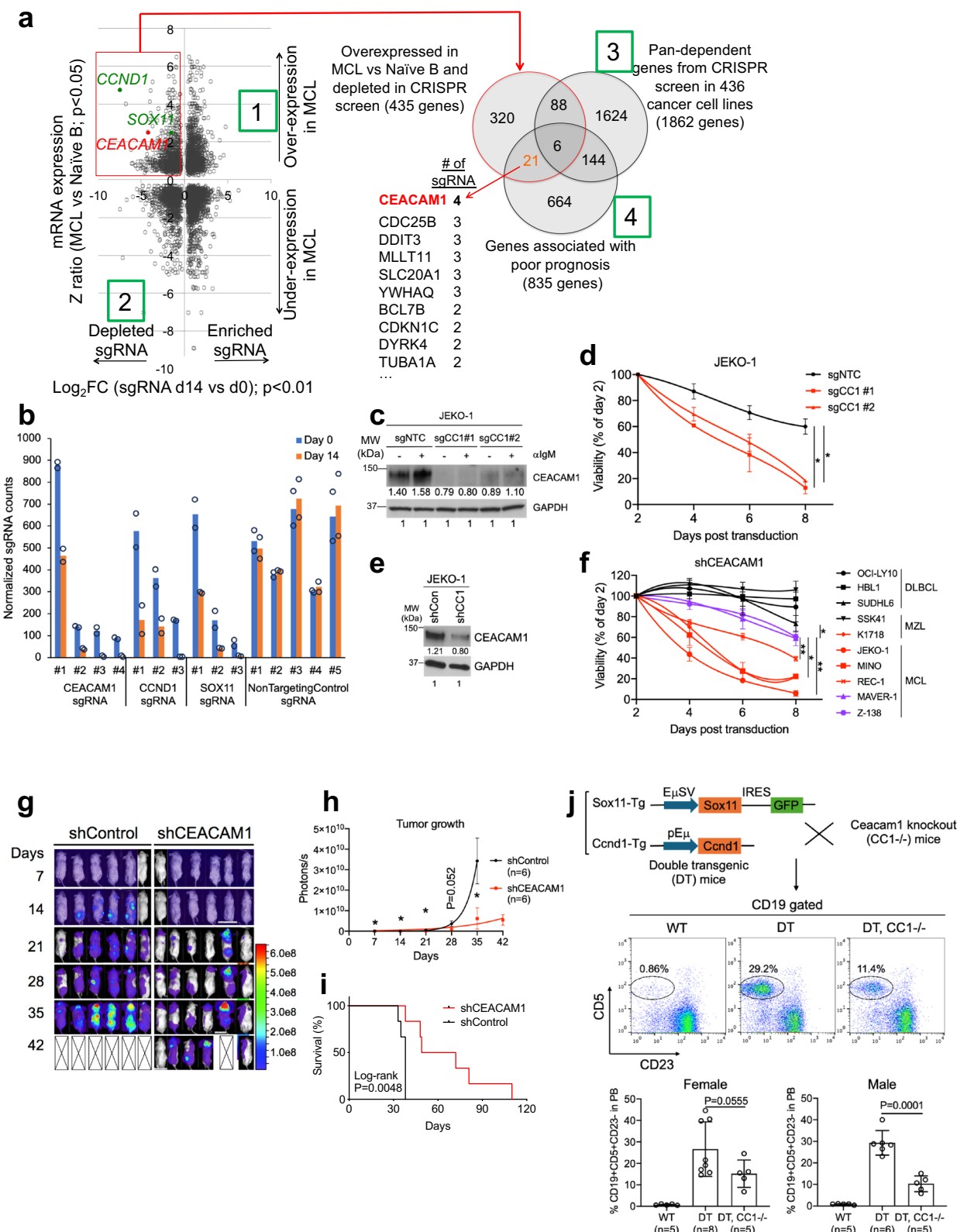

To expand the library screen results, we selected a small-hairpin RNA (shRNA) to knock down CEACAM1 (Fig. 1e) and transduced it into multiple lymphoma cell lines. CEACAM1 depletion led to reduced survival in cells with high CEACAM1 expression (JEKO-1, REC-1, MINO, MAVER-1, and K1718); however, the effect on cell survival was partial in MAVER-1 cells (Fig. 1f). In contrast, depletion of CEACAM1 had little effect on survival of lymphoma cells (SUDHL-6, OCI-LY10, HBL-1, and

SSK41) expressing low or undetectable levels of CEACAM1 (Fig. 1f; see below). CEACAM1 knockdown in the MCL cell line Z-138, which expresses a low level of CEACAM1, also induced a partial effect on cell survival (Fig. 1f). We extended these results in vivo by employing an established xenograft model of MCL using JEKO-1[40,41]. Two million control or CEACAM1 shRNA-transduced JEKO-1 cells expressing firefly luciferase were intravenously injected into immunodeficient NOD/

**Fig. 1 | CRISPR library screen identifies CEACAM1 as essential in MCL. a** Left panel. Log$_2$ fold-change of sgRNA counts between days 14 and 0 ($P < 0.01$; two-sided permutation test) are plotted against the Z ratio of mRNA expression between MCL and naïve-B cells ($P < 0.05$; two-sided $t$-test). Right panel. Overlap of selected genes depleted at least twofold and overexpressed in MCL with pan-dependent genes[31] and genes associated with poor prognosis[32]. Numbers in green boxes indicate filtering strategy (see text). **b** Shown are mean values of selected normalized sgRNA counts on days 0 and 14 from two biological replicates. **c** Immunoblots show CEACAM1 knockout in JEKO-1 cells transduced with control (gNTC) or CEACAM1 gRNAs (gCC1) followed by anti-IgM antibody stimulation (2 µg/ml, 5 min). **d** JEKO-1 cells were transduced with indicated sgRNAs, and viable, GFP$^+$-transduced cells were monitored over time by FACS. Shown are the means of GFP+ fractions compared to day-2 samples from three biological replicates. Error bars, SD. *$P < 0.05$ by a two-sided, paired $t$-test. **e** Immunoblots show CEACAM1 knockdown by shRNA in JEKO-1 cells. **f** Indicated cells were transduced with CEACAM1 shRNA and viable, propidium iodide (PI)-negative cells were assessed by FACS over time. Shown are the means of PI-negative fractions compared to day-2 samples from three biological replicates. Error bars, SD. *$P < 0.05$, **$P < 0.01$ by a two-sided, paired $t$-test. **g** CEACAM1 is required for MCL survival in vivo. JEKO-1 cells were transduced with control or CEACAM1 shRNA and intravenously transplanted into NSG mice. Shown are weekly bioluminescence images of six mice/group. **h** Line graphs show the means of bioluminescence signals measuring tumor growth in mice described in (**g**). Error bars, SEM. *$P < 0.05$ by a one-sided $t$-test. **i** Kaplan−Meier survival analysis of mice shown in (**g**). $P$ value, log-rank test. **j** Top panel, Generation of double SOX11/CCND1 transgenic (DT) and CEACAM1-deficient mice. Middle panel, Representative FACS plots show MCL-like population (CD19 + CD5 + CD23-). Bottom panel, bar graphs show means of %MCL-like cells from the peripheral blood of indicated mice and sample sizes. Error bars, SD. $P$ values, two-sided unpaired $t$-test. Source data are provided as a Source Data file.

SCID/IL2R-Gamma null (NSG) mice, and tumor growth was monitored by whole-body bioluminescence imaging. All mice showed bioluminescence signals in the spleen, bone marrow, and central nervous system, indicating involvement in these organs (Fig. 1g, h). All mice in the control group showed visible bioluminescence by day 14 after tumor implantation and became ill with marked weight loss and hind-leg paralysis, resulting in a median survival of only 37 days. In contrast, mice that received CEACAM1-depleted tumor cells had delayed bioluminescence signals and a median survival of 60 days (Fig. 1i). Together, these findings indicate that CEACAM1 is essential in a subset of MCL tumors both in vitro and in vivo.

Furthermore, we employed a transgenic mouse model of MCL to determine the role of CEACAM1 in MCL progression. The Eµ-SOX11 transgenic mice developed an MCL-like population (CD19$^+$CD5$^+$CD23$^-$) by two months of age with 100% penetrance[42]. To achieve concurrent overexpression of SOX11 and CCND1, hallmark features of MCL biology, Eµ-SOX11 mice were crossed with Eµ-CCND1 mice. The resulting double Eµ-SOX11/CCND1 transgenic (DT) mice exhibited an enhanced MCL-like phenotype compared to Eµ-SOX11 mice[43]. By further crossing DT mice with CEACAM1−/− mice, we observed that the MCL-like population was significantly reduced in male DT mice with a CEACAM1-deficient background ($n = 5$) compared to DT controls ($n = 6$) (Fig. 1j). A similar trend was also observed in female mice, though with lower statistical significance. Collectively, our data suggest that CEACAM1 plays a crucial role in MCL development in the DT mouse model.

## CEACAM1 expression in MCL

To compare CEACAM1 expression levels in other cell types, we re-analyzed previously published gene expression data[30] and found that CEACAM1 mRNA expression is higher in MCL than in normal B cells (Fig. 2a) or other B-cell malignancies (Fig. 2b, c). At the protein level, immunoblot assays showed CEACAM1 is highly expressed in primary MCL samples, in PDX models, and in MCL cell lines as compared to non-MCL cells (Fig. 2d). Flow cytometric analysis also revealed high levels of CEACAM1 surface expression on MCL PDX models (Fig. 2e) and primary MCL samples (Supplementary Fig. 4a) compared to other B-cell lymphoma or CD19$^+$ naïve B cells. Consistent with a previous report[20], we observed higher CEACAM1 expression levels in CD27$^+$ memory B cells as compared to naïve B cells from three healthy donors (Supplementary Fig. 4b). Furthermore, using immunohistochemistry (IHC), we compared CEACAM1 expression in tissue microarrays of different B-cell malignancies. CEACAM1 was found to intensely stain the surface of the lymphoma cells from a majority of primary MCL biopsies (86.9%) and CLL/SLL (93.7%), and some cases of DLBCL (25.6%), follicular lymphoma (FL) (31.3%), and classical Hodgkin lymphoma (cHL) (7.7%) (Fig. 2f and Table 1).

Because the CEACAM1 antibodies used in immunoblotting (Santa Cruz Biotechnology, clone E-1) and IHC (Abcam, clone EPR4049) only recognize the long isoforms of CEACAM1 while the CEACAM1 antibody used in flow cytometry (eBioscience, clone CD66a-B1.1) recognizes the ectodomain of CEACAM1, CEACAM3, CEACAM5, and CEACAM6, we further clarified which CEACAM members are expressed in MCL cells. To do this, we performed qPCR assays using Taqman probes specific for five CEACAM1 isoforms (Supplementary Fig. 5a) and other CEACAM molecules. The Taqman probes were validated for their specific detection of the corresponding CEACAM molecules in the CEACAM$^+$ A549 cell line, using HEK 293T as a negative control (Supplementary Fig. 5b). Our data indicate that MCL mostly expresses CEACAM1 long cytoplasmic tail (CT) isoforms, with low to undetectable levels of CEACAM1 short CT isoforms and undetectable levels of CEACAM3, CEACAM5 and CEACAM6 (Supplementary Fig. 5c, d). We also found that CEACAM1 long isoform was predominantly expressed in the marginal zone lymphoma (MZL) cell line Karpas-1718 (K1718) and modestly expressed in the activated B-cell (ABC) DLBCL cell line HBL-1 (Supplementary Fig. 5c). Together, these data indicate that CEACAM1 is highly expressed in the majority of MCL tumors compared to other CEACAM molecules.

## CEACAM1 is required for BCR signaling in MCL

To determine whether CEACAM1 contributes to BCR signaling in MCL, we compared Ca$^{2+}$ release in response to BCR engagement in CEACAM1-deficient JEKO-1 MCL cells and controls. Knockdown of CEACAM1 resulted in markedly reduced Ca$^{2+}$ signals and decreased phosphorylation of the BCR-proximal kinases LYN and SYK, as well as downstream pathways including ERK, PLCG1, and AKT (Fig. 3a−c). BCR-mediated Ca$^{2+}$ signals were also significantly reduced after CEACAM1 depletion in MAVER-1 cells and in the MZL cell line K1718, both of which express high levels of CEACAM1 (Fig. 3d, e). In addition, compared to mature splenic B cells from wild-type mice, Ceacam1$^{-/-}$ B cells displayed reduced BCR-mediated Ca$^{2+}$ signals (Fig. 3f). These results suggest that CEACAM1 is required for BCR signaling in MCL and normal B cells. To investigate further, we evaluated the association between CEACAM1 expression and BCR signaling activity in MCL. Previous studies have demonstrated that MCL tumors isolated from the lymph node (LN) microenvironment express genes that are correlated with BCR activation, in contrast to tumor cells from peripheral blood (PB)[44]. These results prompted us to re-examine gene expression data from this study, and we found that CEACAM1 expression levels were indeed higher in LN-derived MCL cells than in PB-derived MCL cells (Fig. 3g). We next evaluated the correlation between CEACAM1 expression in lymphoma cells and their response to ibrutinib, which inhibits the BCR pathway kinase BTK. Lymphoma cells with high CEACAM1 expression were found to be more sensitive to ibrutinib than those with low expression levels (Fig. 3h), except for MAVER-1 cells, which had high CEACAM1 expression but were resistant to ibrutinib (see Discussion). Together, these results indicate that CEACAM1 expression is highly correlated with BCR signaling activity.

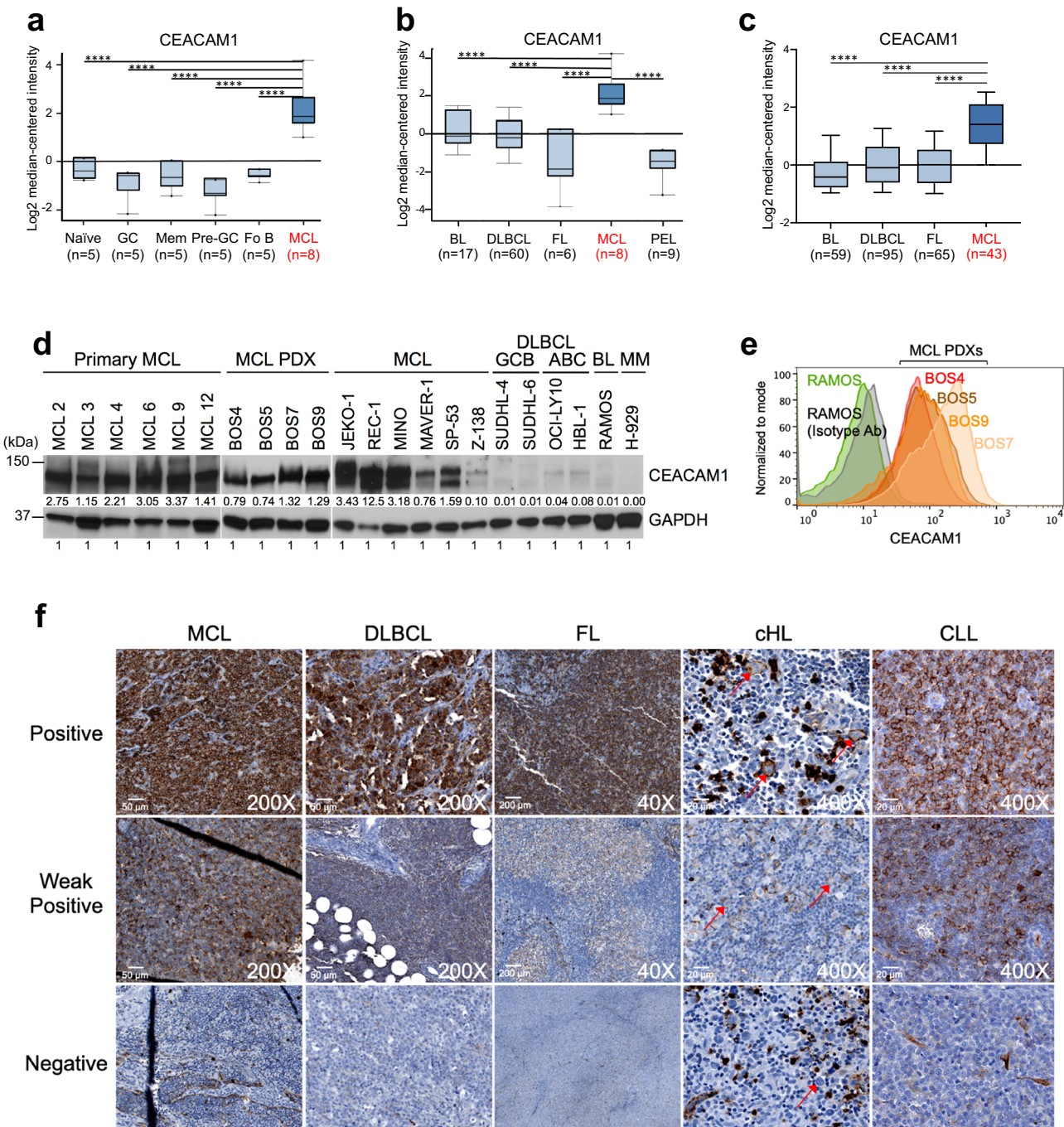

**Fig. 2 | CEACAM1 expression in lymphoma cells. a−c** Box plots show CEACAM1 mRNA expression levels in MCL and other cell types from the datasets GSE2350 (**a**, **b**) and GSE132929 (**c**). The boxes extend from the 25th to 75th percentiles with the center line as median and whiskers drawn from 10th to 90th percentiles. ****$P < 0.0001$ by a two-sided Mann−Whitney $U$- test. GC germinal center, Mem memory, Fo follicular, BL Burkitt lymphoma, DLBCL diffuse large B-cell lymphoma, FL follicular lymphoma, PEL primary effusion lymphoma. **d** Representative immunoblot analysis of CEACAM1 expression in indicated cells from at least three independent experiments. Numbers below bands represent densitometric values of CEACAM1 signals normalized over GAPDH loading controls. **e** Flow cytometry analysis of surface CEACAM1 expression in MCL PDXs compared to CEACAM1-negative RAMOS cells from at least two independent experiments. Live cell gating in this experiment and throughout the study is described in Supplementary Fig. 13. **f** Representative immunohistochemistry images showing varying CEACAM1 staining levels for indicated formalin-fixed paraffin-embedded tissue microarrays from three independent experiments. In cHL tissue, background plasma cells are also stained positive. cHL classic Hodgkin lymphoma, CLL chronic lymphocytic leukemia. See Table 1 for a summary of CEACAM1 positivity in specific diseases. Source data are provided as a Source Data file.

We next determined which structural features of CEACAM1 are involved in BCR signaling by generating mutant CEACAM1 constructs that contained either mutated ITIM tyrosine residues (CEACAM1-4L-Y493F/Y520F) or lacked the cytoplasmic tail (CEACAM1-4S). The expression of these constructs was verified by immunoblots and flow cytometry (Supplementary Fig. 6a−c). To evaluate the impact of different CEACAM1 variants on BCR signaling, we introduced full-length CEACAM1-4L, CEACAM1-4L-Y493F/Y520F, or CEACAM1-4S into JEKO-1

**Table 1 | Summary of CEACAM1 IHC staining of tissue microarrays**

| Disease | Positive | Weak | Negative | Total positive | % Positive |
|---|---|---|---|---|---|
| Mantle cell lymphoma | 16 | 4 | 3 | 20 | 86.96% |
| CLL/SLL | 8 | 7 | 1 | 15 | 93.75% |
| DLBCL | 10 | 9 | 63 | 19 | 23.17% |
| GCB | 8 | 7 | 35 | 15 | 30.00% |
| ABC | 0 | 2 | 13 | 2 | 13.33% |
| Indeterminate | 2 | 0 | 15 | 2 | 11.76% |
| Follicular lymphoma | 11 | 15 | 57 | 26 | 31.33% |
| grade 1–2 | 4 | 7 | 35 | 11 | 23.91% |
| grade 3A | 5 | 5 | 11 | 10 | 47.62% |
| grade 3B | 1 | 1 | 1 | 2 | 66.67% |
| DLBCL + FL | 0 | 2 | 7 | 2 | 22.22% |
| Indeterminate | 1 | 0 | 3 | 1 | 25.00% |
| Classic Hodgkin lymphoma | 2 | 1 | 36 | 3 | 7.69% |

cells in which endogenous CEACAM1 had been knocked out (Fig. 3i). Our results showed that only JEKO-1 cells expressing CEACAM1-4L, but not CEACAM1-4L-Y493F/Y520F or CEACAM1-4S, were able to rescue the defective BCR-mediated kinase activities after CEACAM1 knockout (Fig. 3j). Taken together, these results indicate that CEACAM1 plays an active role in BCR signaling in both normal B cells and MCL, and that the ITIM tyrosine residues on the cytoplasmic tail are required for this function.

### CEACAM1 co-localizes to lipid rafts and is required for F-actin reorganization during BCR activation

Since the cytoplasmic tail is important for CEACAM1 function (Fig. 3j), we hypothesized that molecules that interact with this domain play a key role in the BCR signaling complex. One such candidate is the F-actin binding adapter filamin A (FLNA)[45], a known CEACAM1-interacting protein[46]. To determine whether FLNA contributes to BCR signaling in MCL, we knocked down FLNA in JEKO-1 cells and performed $Ca^{2+}$ flux assays after BCR stimulation. FLNA depletion significantly reduced both $Ca^{2+}$ signals and cell survival in JEKO-1 cells (Supplementary Fig. 7). Expanding on this finding, we examined the lipid-raft signaling microenvironment to determine whether CEACAM1 and FLNA promote BCR signaling through increasing lipid-raft function. Control (gNTC) or CEACAM1-knockout (gCC1) JEKO-1 cells were stimulated with anti-IgM antibody, and lipid rafts were isolated by sucrose gradient centrifugation (Fig. 4a). As expected, we detected the two membrane raft markers LYN and flotillin-1[22] in the raft fraction (fraction I) from all four samples (Fig. 4b). In addition, LYN activation, as detected by the anti-phospho-$Y^{416}$ SRC antibody, was observed in lipid-raft fractions from the IgM-stimulated samples in a CEACAM1-dependent manner. CEACAM1 was detected in raft fractions of both unstimulated and IgM-stimulated B cells. Importantly, we found that the levels of FLNA were increased in the lipid rafts of IgM-stimulated cells, but only in the presence of CEACAM1 and not in CEACAM1-depleted cells (Fig. 4b). As an alternative approach, we used confocal immunofluorescence microscopy to confirm the role of CEACAM1 in FLNA localization and LYN activity in lipid rafts during BCR signaling. In control JEKO-1 cells, FLNA is recruited to the membrane regions that are positive for LYN after BCR crosslinking. In contrast, CEACAM1 depletion significantly reduced FLNA recruitment to LYN-enriched lipid-raft regions (Fig. 4c and Supplementary Fig. 8). In addition to impaired FLNA recruitment, CEACAM1 deficiency also had a

negative impact on the lipid raft marker GM1 staining and on LYN activation, both of which were restored upon CEACAM1 reintroduction (Fig. 4d).

Because FLNA binds F-actin[45], CEACAM1-dependent recruitment of FLNA to lipid rafts likely involves F-actin reorganization to facilitate lipid raft accumulation. To determine the role of CEACAM1 in F-actin reorganization, we used confocal immunofluorescence to observe the changes in the F-actin network in JEKO-1 cells in which CEACAM1 was deleted. Upon BCR crosslinking, the F-actin cytoskeleton was highly clustered on the cell surface and became detectable by confocal microscopy on control JEKO-1 cells (Fig. 4d). Strikingly, accumulation of the F-actin network after BCR stimulation was markedly impaired in CEACAM1-deficient MCL cells (Fig. 4d). The negative impact of CEACAM1 deficiency on F-actin accumulation after BCR crosslinking was readily rescued when CEACAM1 was reintroduced (Fig. 4d). Taken together, these data indicate that CEACAM1 supports BCR signaling by stabilizing lipid rafts through facilitating the localization and reorganization of structural elements such as FLNA and the underlying F-actin cytoskeletal network, respectively.

### CEACAM1 interactions with BCR-proximal signaling components depend on intact tyrosine residues and the ligand-binding N-terminal domain

To further investigate the function of CEACAM1 during BCR activation, we examined the interactions between CEACAM1 and SYK or SHP-1 using two MCL cell lines, JEKO-1 and MINO, both having elevated CEACAM1 expression (Fig. 2d). By employing the proximity ligation assay (PLA), we demonstrated that interactions of CEACAM1 with SYK peaked between 5 and 15 min following BCR stimulation and then decreased by 30 min both in JEKO-1 and MINO cell lines. However, increased recruitment of SHP-1 to CEACAM1 was only observed at ~30 min after BCR crosslinking (Fig. 5a, b). To investigate this further, we used co-immunoprecipitation (co-IP) as an alternative method and found comparable early trafficking kinetics of SYK, while SHP-1 recruitment to CEACAM1 occurred at a later stage following BCR activation in both JEKO-1 and MINO cells. Monitoring SYK activity through its phosphorylation revealed that, in both JEKO-1 and MINO cells, SYK activation peaked at 15 min after BCR stimulation and declined by 30 min, coinciding with the increased recruitment of SHP-1 to CEACAM1 (Fig. 5c, d). However, while the early SYK kinetics reached statistical significance, the slower SHP-1 kinetics did not, potentially due to a combination of factors such as limitations of the SHP-1 antibody, epitope masking caused by the interaction itself, or unidentified components within the complex interfering with antibody recognition. It is also possible that this interaction is particularly sensitive to minor experimental variations.

We next determined which CEACAM1 domains are involved in its interactions with BCR signaling components. To do this, we used CEACAM1-knockout JEKO-1 cells that were reconstituted with either full-length 4 L, a mutant Y493F/Y520F, or an N-domain deleted (ΔN) CEACAM1 construct. We monitored the kinetics of protein interactions using PLA and co-IP assays after BCR stimulation. As expected, tyrosine phosphorylation on the Y493F/Y520F mutant CEACAM1 was impaired upon BCR crosslinking. In contrast to the 4 L CEACAM1, this mutant was also defective in its interactions with FLNA, SYK, and SHP-1 following BCR activation (Supplementary Fig. 9). Remarkably, like the Y493F/Y520F mutant, the ΔN CEACAM1 construct was defective in ITIM phosphorylation after BCR stimulation, despite having intact tyrosine residues. As a result, ΔN CEACAM1 failed to interact with FLNA, CD79B, LYN, SYK, or SHP-1 (Fig. 6a). Notably, the full-length 4 L CEACAM1 pull-down from unstimulated samples exhibited basal phosphorylation, potentially resulting from basal interactions with LYN and/or SYK, which were also found to interact with CEACAM1 prior to stimulation (Fig. 6a). Using the PLA method, we confirmed defective ΔN CEACAM1 interactions with CD79B, LYN, SYK, and SHP-1

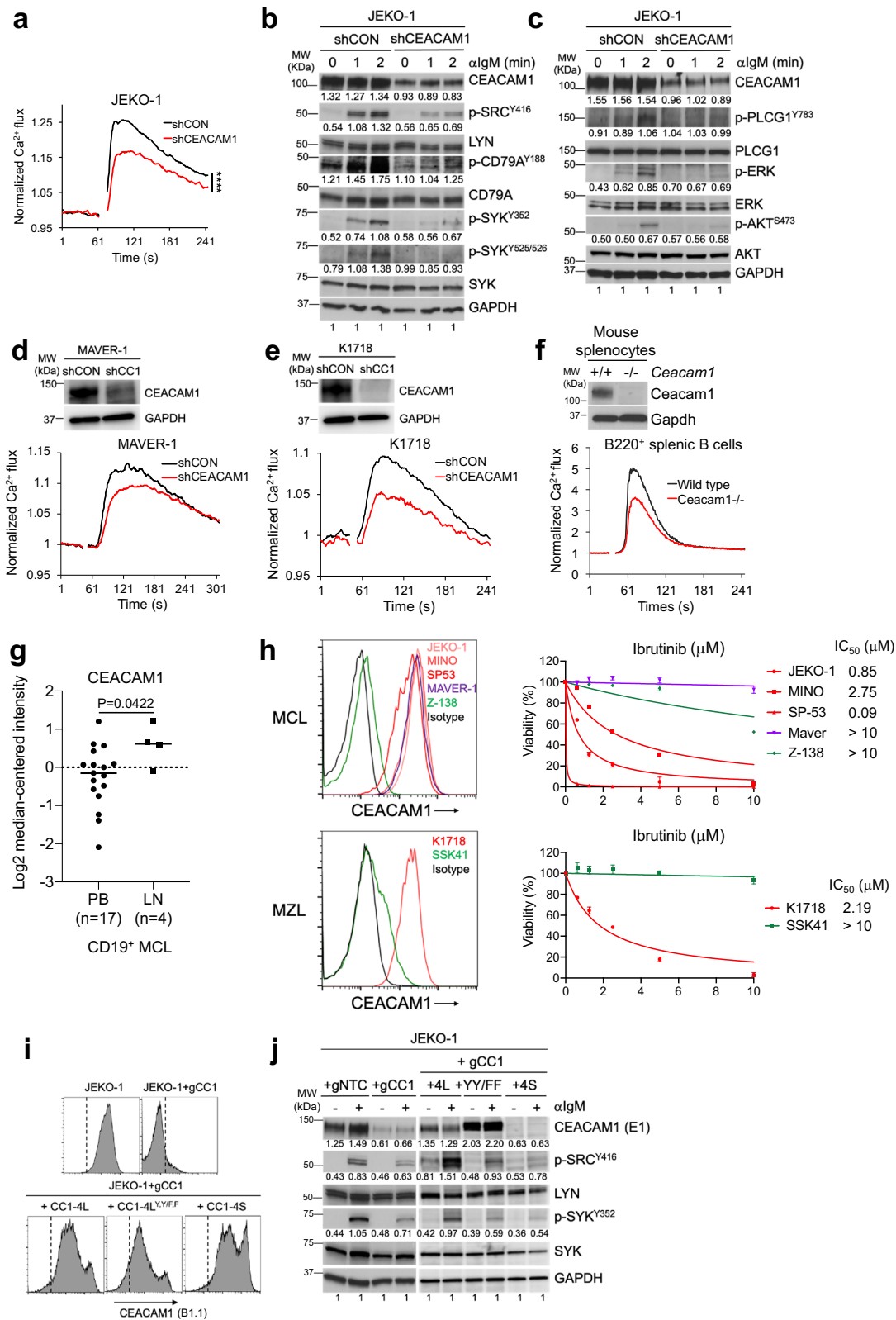

during BCR activation (Fig. 6b, c). The negative impact of the Y493F/ Y520F mutant or ΔN CEACAM1 on interactions with BCR-proximal molecules and BCR signaling was consistently reflected in impaired CD79B association with SYK compared to wildtype 4 L CEACAM1, as demonstrated by PLA (Supplementary Fig. 10). Therefore, these data indicate that CEACAM1 positively contributes to BCR signaling by initially recruiting and activating LYN and SYK at the BCR complex, and

subsequently recruiting SHP-1 to dampen the signal. Importantly, these interactions rely on the phosphorylation of tyrosine residues and the presence of the N-domain of CEACAM1.

## A dual role of CEACAM1 in BCR signaling

The aforementioned data suggest that CEACAM1 has a positive impact on the early stages of BCR signaling by activating LYN and SYK,

**Fig. 3 | CEACAM1 is required for BCR signaling. a–e** Indicated cell lines were transduced with control or CEACAM1 shRNA, followed by stimulation with anti-IgM (1 µg/mL). **a** Shown are normalized $Ca^{2+}$ signals. ****$P < 0.0001$ by two-way ANOVA. Data were representative of at least three independent experiments.
**b**, **c** Immunoblots show effects of CEACAM1 knockdown on BCR signaling components. Data were representative of three independent experiments.
**d**, **e** Immunoblots show CEACAM1 knockdown (top panels) and its effect on $Ca^{2+}$ signals (bottom panels). **f** Top panel, immunoblots of splenocytes from wild type (+/+) or Ceacam1-deficient (−/−) mouse with indicated antibodies. Bottom panel, $Ca^{2+}$ signals of indicated splenic B220[+] B cells. Data from (**d**–**f**) are representative of at least two independent experiments. **g** CEACAM1 mRNA expression levels in CD19+ sorted MCL cells from peripheral blood (PB) or lymph nodes (LN) analyzed from the GSE70910 dataset[44]. Closed circles or squares represent individual patient samples. Horizontal bars from each group indicate mean mRNA expression. *P* value is from a two-sided unpaired *t*-test with Welch's correction. **h** CEACAM1 expression

is correlated with ibrutinib response. Left panels, FACS plots showing surface CEACAM1 expression on indicated MCL and MZL cell lines. Isotype, negative isotype antibody on Z-138 cells. Right panels, Indicated cell lines were treated with indicated doses of ibrutinib for 4 days, and viable propidium iodide (PI)-negative cells were assessed by flow cytometry. Line graphs showing means of normalized PI-negative fractions from three independent experiments. Error bars, SD. Fifty-percent inhibition concentration ($IC_{50}$) values were calculated by GraphPad Prism v8. **i**, **j** ITIM tyrosine residues are required for BCR signaling. JEKO-1 cells transduced with control (gNTC) or CEACAM1 gRNA (gCC1), followed by reintroducing WT CEACAM1 (4 L), CC1-4L-Y493F/Y520F mutant (YY/FF), or short cytoplasmic tail (4S). Controls or reconstituted cell lines were verified for CEACAM1 expression by FACS using B1.1 antibody (**i**) and stimulated with 1 µg/mL of anti-IgM for 5 min, followed by immunoblotting with indicated antibodies, including the CEACAM1 E1 antibody (**j**). Data from (**i**, **j**) are representative of at least two independent experiments. Source data are provided as a Source Data file.

consistent with a previous report[20]. However, this appears contradictory to other studies proposing that CEACAM1 inhibits antigen receptor signaling in B and T cells by recruiting the negative regulator SHP-1[35,47–49]. In exploring these conflicting roles of CEACAM1, we noted that the inhibitory functions of CEACAM1 were primarily studied in T cells or B cells lacking CEACAM1 expression, followed by the introduction of CEACAM1. To establish a system comparable to these earlier studies, we selected the MCL cell line Z-138 with low CEACAM1 expression and reintroduced CEACAM1 into these cells. Overexpression of CEACAM1 in Z-138 cells did not affect IgM surface expression levels (Supplementary Fig. 11). However, CEACAM1-overexpressing Z-138 cells exhibited downregulated BCR signaling, evidenced by reduced CD79A and SYK phosphorylation after IgM stimulation, compared to the empty vector control (Fig. 7a). Similarly, overexpression of CEACAM1 in the CEACAM1-negative Burkitt lymphoma cell line ST486 did not affect the IgM surface expression (Supplementary Fig. 11) but significantly downregulated SYK activation following BCR stimulation compared to control cells (Supplementary Fig. 12). The inhibitory effect of CEACAM1 on BCR signaling was dependent on its cytoplasmic tail, as expression of CEACAM1-4S, which lacks this domain, in Z-138 cells showed little change in SYK and CD79A phosphorylation compared to empty vector control (Fig. 7a).

We then performed immunoprecipitation assays to investigate the recruitment of SYK, SHP-1, and SHP-2 by CEACAM1, comparing CEACAM1-overexpressing Z-138 cells with JEKO-1 cells, where CEACAM1 was knocked out and subsequently reintroduced using the same CEACAM1-4L expression construct. In the input (pre-immunoprecipitation) samples, reintroducing CEACAM1 into CEACAM1-knockout JEKO-1 cells led to increased SYK phosphorylation compared to controls (Fig. 7b). In contrast, CEACAM1 overexpression in Z-138 cells, again, resulted in downregulated BCR signaling, as indicated by reduced SYK phosphorylation (Fig. 7b). The opposing effects on BCR signaling in JEKO-1 and Z-138 cells upon CEACAM1 overexpression corresponded with CEACAM1's interaction patterns in the immunoprecipitated samples—predominantly associating with SYK in JEKO-1 cells, whereas in Z-138 cells, it interacted more with SHP-1 and SHP-2 (Fig. 7c). Furthermore, our complementary PLA approach confirmed significantly more CEACAM1-SHP-1 interactions than CEACAM1-SYK interactions in CEACAM1-overexpressing Z-138 cells (Fig. 7d). Collectively, these findings indicate that CEACAM1 not only plays an activating role but can also act as a negative regulator of BCR signaling in lymphoma cells with low or absent CEACAM1 expression by preferentially binding to SHP-1 instead of SYK.

## Discussion

Despite strong evidence supporting chronic activation of the BCR in MCL, the mechanisms driving pathological BCR signaling in MCL are not fully understood. In this study, we have identified the type 1

transmembrane protein CEACAM1 as a crucial component of BCR signaling in MCL and MZL. CEACAM1 facilitates BCR signaling by localizing to the membrane microdomains where it recruits SYK, leading to an abundance of this critical kinase at the BCR complex. This BCR activating function of CEACAM1 requires the intact N-domain and cytoplasmic tyrosine phosphorylation. We also uncovered two distinct patterns in the recruitment dynamics of CEACAM1 towards BCR-proximal kinases and phosphatases, which correlate with its expression status across different cellular contexts (Fig. 8). This discovery provides compelling evidence to reconcile the previously conflicting roles attributed to CEACAM1.

To dissect the role of CEACAM1 in BCR signaling, we investigated the proteins that interact with CEACAM1. Our initial focus was on FLNA, a structural adapter protein that interacts with both CEACAM1 and the F-actin cytoskeleton, the latter of which is actively remodeled during BCR activation[50]. Notably, FLNA was recently found to interact with the calcium-sensing receptor CaSR[51], suggesting its potential link to BCR-mediated $Ca^{2+}$ signaling. Our data indicate that CEACAM1 works with FLNA and the underlying three-dimensional F-actin network to facilitate successful BCR activation. This working model is similar to the "tether and trap" model proposed by refs. 52,53. According to the Tavano model, engagement of the TCR with antigen on antigen-presenting cells leads to FLNA recruitment to the immunological synapse (IS) by binding to the cytoplasmic tail of CD28. This enables actin-based recruitment and trapping of membrane rafts in the IS. Notably, a previous study by ref. 54 reported that *Ceacam1*-deficient T cells fail to recruit FLNA to the IS, resulting in reduced proliferation and survival of CD8[+] T cells after antigen activation. Thus, the role of CEACAM1 in regulating lipid raft dynamics and facilitating BCR signaling in B cells has implications for TCR signaling in T cells as well.

Our study confirms that CEACAM1 plays an important role in regulating BCR signaling, consistent with previous reports on *Ceacam1*-deficient mice[20,34]. These mice showed a decrease in the number of B cells, especially splenic marginal zone B cells that rely on strong BCR signaling[55], as well as defective SYK kinase activity and insufficient antibody response after antigen stimulation[20]. Our study also revealed a significant reduction in BCR-mediated $Ca^{2+}$ signals in *Ceacam1*-deficient B cells compared to wildtype B cells, confirming the critical activating role of CEACAM1 in BCR signaling. The positive role of CEACAM1 is further supported by its dynamic interactions with LYN, SYK, and SHP-1. Specifically, CEACAM1 is found to associate with LYN and SYK in the CD79B vicinity after BCR engagement, thereby heightening the concentrations of these crucial kinases at the BCR complex during peak activation. Subsequently, we speculate that CEACAM1 increases its interaction with SHP-1, which exerts its regulatory role at the end of BCR activation. Furthermore, our data highlight the requirement of the ligand-binding N-terminal domain of CEACAM1 for phosphorylation of ITIM tyrosine residues, and for CEACAM1 interactions with structural elements and signaling kinases. By using ΔN

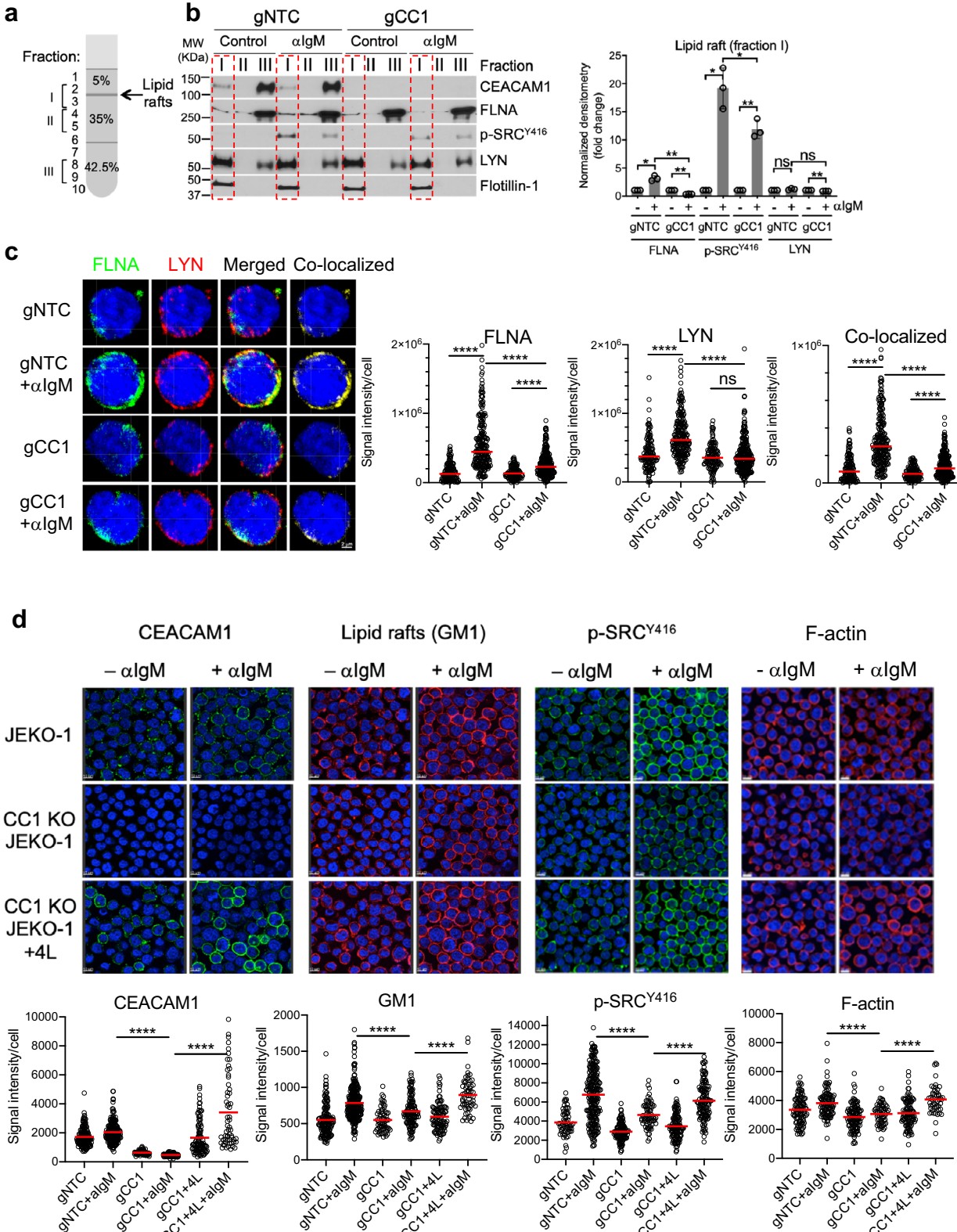

CEACAM1 lacking the homophilic interaction, we observed that ITIM tyrosine phosphorylation and CEACAM1 interactions with signaling molecules were impaired during BCR activation. These findings are consistent with the study by ref. 48, which demonstrated the absence of tyrosine phosphorylation on the cytoplasmic tail of CEACAM1 when mutations (R43G and Q44L) were introduced at the N-terminal domain to abolish homophilic binding.

Despite its demonstrated positive impact on BCR signaling, as shown in our current study and by refs. 20,54, our data also revealed the inhibitory function of CEACAM1 in the lymphoma cell lines with low or no CEACAM1 expression, consistent with previous reports[47–49]. It appears that the molecular basis of this negative role of CEACAM1 hinges on its distinct interaction with SHP-1, as opposed to SYK, observed in its positive function during the initial stages of BCR

**Fig. 4 | CEACAM1 co-localizes to lipid rafts and promotes F-actin reorganization during BCR activation. a** Diagram of sucrose-density gradient fractionation of lipid rafts. **b** Left panels, Immunoblots of fractions indicated in (**a**) from control or CEACAM1-knockout JEKO-1 cells stimulated with control or anti-IgM antibody (1 μg/ml) for 2 min. Right panels, immunoblot signal quantification of fraction I (red dashed-line box) after normalization to Flotillin-1 and unstimulated controls. Shown are the means of fold changes from three independent experiments. Error bars, SD. **$P < 0.01$, *$P < 0.05$ by a two-sided, paired $t$-test. ns not significant. **c** Left panels, Control (gNTC) or CEACAM1 knockout (gCC1) JEKO-1 cells were stimulated with 2 μg/ml anti-IgM antibody for 2 min. Shown are representative cells from confocal immunofluorescence images in Supplementary Fig.8 of control and IgM-stimulated cells co-stained with anti-FLNA (green) and anti-LYN (red) antibodies, followed by nuclear staining with DAPI (blue). Scale bar, 2 μm. Right panels, quantified fluorescent signals for each cell from the samples described. Shown are the sums of intensities in signal-positive areas per cell in arbitrary units from three independent experiments. Horizontal red bars indicate the mean. Approximately 200 cells from each sample were analyzed. ****$P < 0.0001$ by a two-sided unpaired $t$-test. ns not significant. **d** CEACAM1 is required for lipid-raft assembly. Top panels, Control, CEACAM1 knockout (CC1 KO), or CC1 KO + CEACAM1-4L (4 L) JEKO-1 cells were stimulated with 2 μg/ml anti-IgM antibody for 5 min. Shown are representative confocal immunofluorescence images of control and IgM-stimulated cells stained with anti-CEACAM1 antibody (green), GM1 via cholera toxin B (red), p-SRC$^{Y416}$ antibody (green), or F-actin via Actin-Stain 555 Phalloidin (red) followed by nuclear staining with DAPI (blue). Scale bar, 10 μm. Bottom panels, quantified fluorescent signals from the samples described in the top panels. Shown are the sums of intensities in signal-positive areas per cell in arbitrary units from three independent experiments. Horizontal red bars indicate the mean. Approximately 200 cells from each sample were analyzed. ****$P < 0.0001$ by a two-sided unpaired $t$-test. Source data are provided as a Source Data file.

activation. The variations in these binding dynamics seem to depend on the specific cellular context dictated by differing CEACAM1 expression status, suggesting the involvement of additional regulatory factors in this process. Identifying these regulators is thus essential for a more comprehensive understanding of CEACAM1's signaling outcomes in various cellular settings. One potential clue from our data is the significant difference in SYK expression levels between JEKO-1 and Z-138 cells. In JEKO-1 cells, high SYK expression may enable SYK to outcompete SHP-1 in binding to CEACAM1, ensuring active BCR signaling. In contrast, in Z-138 cells, SYK is expressed at much lower levels, allowing SHP-2 to outcompete it in binding to CEACAM1 and thereby leading to downregulation of BCR signaling. Further investigations, including experimental manipulation of SYK, SHP-1, and SHP-2 levels in their interactions with CEACAM1, should help to clarify how these factors influence BCR signaling.

Nonetheless, the role of CEACAM1 in BCR signaling may have important implications for therapies directed at this pathway. The strong correlation between CEACAM1 expression levels and the BCR signature in MCL tumors (Fig. 3g), and its relationship to ibrutinib responsiveness in MCL and MZL cell lines (Fig. 3h), suggests a potential utility of CEACAM1 as a biomarker of ibrutinib therapy in these lymphomas. However, CEACAM1 expression cannot predict ibrutinib sensitivity in patients with mutations that activate alternative pathways, thus overriding BCR signaling. For instance, the MCL cell line MAVER-1, despite its high CEACAM1 expression, is resistant to ibrutinib (Fig. 3h) due to the presence of TRAF3 deletion[56] and NRAS G12C mutation[57] that activate downstream pathways. Moreover, somatic mutations in BCR pathway components, often found in ABC DLBCL, can affect BCR clustering[7], likely rendering these lymphomas less dependent on CEACAM1. Therefore, the combined assessment of CEACAM1 expression levels and mutation analysis may serve as valuable biomarkers for predicting ibrutinib sensitivity in MCL and potentially MZL cases.

## Methods

All experiments in this study comply with relevant ethical regulations by City of Hope's Institutional Biosafety Committee (IBC) and Institutional Animal Care & Use Committee (IACUC) under approved protocols IBC 10024, IACUC 11046, and IACUC 22054.

### Cell lines and culture conditions

Human MCL lines JEKO-1, Z-138, MINO, REC-1, and MAVER-1 were obtained from the American Type Culture Collection (ATCC, Manassas, VA). SP-53, SUDHL-4, SUDHL-6, OCI-LY10, HBL-1, RAMOS, K1718, SSK41, and H-929 were kindly provided by Dr. Louis Staudt. Cells were cultured in RPMI-1640 medium (Corning, CA, USA), supplemented with 10% fetal bovine serum (FBS) (Sigma-Aldrich, MO, USA), 100 IU/mL penicillin and 100 μg/mL streptomycin (Thermo Fisher Scientific, MA, USA) in a humidified incubator at 37 °C with 5% $CO_2$. In some

experiments, JEKO-1 cells were transduced with firefly luciferase/eGFP and sorted for a greater 99% positive population. Mouse splenic B cells were isolated by depletion of non-B cells from single cell suspensions using the MagniSort™ Mouse B-cell Enrichment Kit (Thermo Fisher Scientific, Waltham, MA) and following the manufacturer's recommendations. Splenocytes were cultured in RPMI-1640 medium supplemented with 2 mM glutamine (Thermo Fisher Scientific), 1 mM sodium pyruvate (Thermo Fisher Scientific), 60 μM β-ME (Sigma-Aldrich, St. Louis, MO), and 10% FBS in a humidified incubator at 37 °C with 5% $CO_2$.

All cell lines were cultured in RPMI-1640 medium (Life Technologies, Grand Island, NY), supplemented with 10% fetal bovine serum (FBS), 100 IU/mL penicillin, and 100 μg/mL streptomycin in a humidified incubator at 37 °C with 5% $CO_2$. Cell lines were commercially authenticated using the Promega GenePrint 10 System.

### Primary MCL samples and patient-derived xenografts

Viable primary MCL cells were thawed from frozen de-identified samples provided by the Pathology Department under City of Hope's Institutional Review Board-approved protocol as previously described[58]. Cells were washed in 20% fetal bovine serum-supplemented RPMI-1640 medium and incubated overnight at 37 °C in a $CO_2$ incubator before analysis. MCL patient-derived xenografts (PDX) were acquired from the public repository of xenografts (ProXe.org) (Supplementary Data 4). Cells were intravenously transferred into NOD.Cg-$Prkdc^{scid}Il2rg^{tm1Wjl}$/SzJ (NSG) mice (The Jackson Laboratory, Bar Harbor, ME). Xenografts were obtained by mashing the mouse spleen through a 7-μm cell strainer and were viably frozen for subsequent use.

### CRISPR library screen

The genome-wide CRISPR library screen was carried out using the Human GeCKO v2 Library, 2-plasmid system (a gift from Feng Zhang; Addgene #1000000049) and following the protocol as described[26]. Briefly, the library, which contains 122,417 unique single-guide (sg) RNAs targeting the human genome with six sgRNAs per gene, was amplified using Endura ElectroCompetent cells following recommendations by the manufacturer (Lucigen, Middleton, WI). The amplified library was sequenced by next-generation sequencing to determine sgRNA distribution using oligos listed in Supplementary Data 5. The entire library, together with helper plasmids pMD2.g and psPAX2 were then transfected into HEK 293T cells and lentiviral supernatants were collected after 2 days followed by spin infection at $1200 \times g$, in two replicates of doxycycline-inducible Cas9-expressing JEKO-1 cells for 1 h in the presence of 8 μg/ml polybrene. To ensure most cells receive one sgRNA, viral titers were adjusted at an MOI <0.3. Library transduction was scaled up to 500 cells per sgRNA for sufficient coverage of the complex library. Transduced cells were selected by puromycin for 3 days, and doxycycline (1 μg/ml) was added to induce Cas9 expression, followed by culturing for an additional 14 days. Genomic DNA was

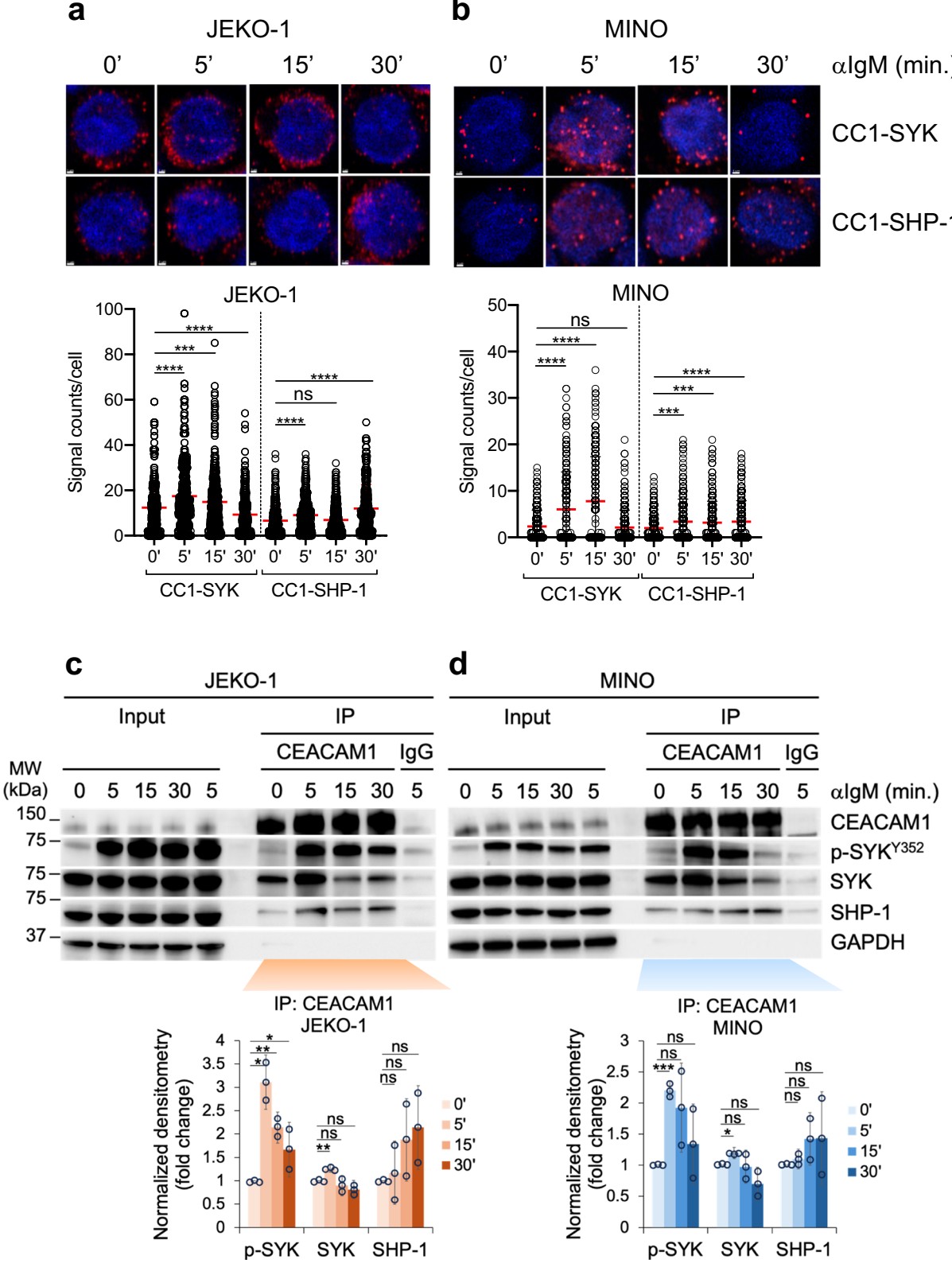

harvested on days 0 (day 3 in puromycin) and 14 and subjected to high-throughput sequencing to determine sgRNA abundance using oligos listed in Supplementary Data 5. Quantification of sgRNA depletion or enrichment was performed using the MAGeCK software as described[59]. The $P$ value for each gene was determined by a permutation test using the negative binomial model. At least two independent sgRNAs

depleted by at least twofold on day 14 as compared to day 0 ($P < 0.01$) were selected as potential candidates. To validate the screen results, top- and bottom-strand oligos of control (gNTC) or the top three CEACAM1 gRNAs were synthesized and cloned into the library backbone plasmid lentiGuide-Puro (Addgene #52963) using a Golden Gate assembly as described[26].

**Fig. 5 | CEACAM1 interactions with BCR signaling components. a**, **b** Top panels, Proximity ligation assay (PLA) showing interactions (visualized as red dots using Airyscan FAST 2D confocal microscope and a 40x/1.2NA water objective) between CEACAM1 (CC1) and the indicated proteins in JEKO-1 and MINO cells stimulated with 2 μg/ml of anti-IgM antibody for the indicated times. Bottom panels, Quantification of PLA signals shown in the top panels for ~200 cells on average from three independent experiments using QuPath 0.3.2 software. ****$P < 0.0001$, ***$P < 0.001$ by a two-tailed unpaired $t$-test. ns not significant. **c**, **d** Immunoprecipitation analysis of CEACAM1 interactions. Top panels, JEKO-1 or MINO cells were stimulated with 2 μg/ml of anti-IgM antibody for the indicated times, and CEACAM1 was immunoprecipitated with a CEACAM1-specific antibody or IgG control antibodies, followed by immunoblotting with the indicated antibodies. One percent of the total lysates was used as an input control. Bottom panels, Quantification of indicated co-IP signals shown in the top panels. Bar graphs show the means and individual densitometric values from three independent experiments for each timepoint normalized to the CEACAM1 pull-down signals. Error bars, SD. ***$P < 0.001$, **$P < 0.01$, *$P < 0.05$ by a two-sided unpaired $t$-test. ns not significant. Source data are provided as a Source Data file.

## DNA transfection and lentiviral transduction
DNA transfection was performed by mixing DNA with Lipofectamine 2000 (Life Technologies, Grand Island, NY) and following the manufacturer's instructions. For lentiviral transduction, a lentiviral vector together with helper plasmids, pMD2.g (Addgene #12259) and psPAX2 (Addgene #12260), were co-transfected into HEK 293T cells using Lipofectamine 2000. Viral supernatants were harvested 48 h after transfection and were used to transduce target cells by centrifugation at 1200×$g$ for 1 h in 8 μg/ml polybrene.

## Gene expression analysis
Transcriptome data of selected tissues were downloaded from NCBI GEO (GSE2350[30], GSE132929, and GSE70910[44]) and analyzed using R Bioconductor packages for Affymetrix Oligonucleotide Arrays[60]. Raw data (CEL files) were normalized using the robust multi-array average (RMA) method. Z-score transformation[61] was used to compare significant changes in gene expression between samples. Data were presented as log2 median-centered intensity or Z ratio.

## Validation of CEACAM1 sgRNAs and RNA interference reagents
Top- and bottom-strand oligos of control (gNTC) or the top 3 CEACAM1 gRNAs (Supplementary Data 6) were synthesized and cloned into the library backbone plasmid lentiGuide-Puro (Addgene #52963) using a Golden Gate assembly as described (see ref. 26). After verification by Sanger sequencing, the resultant lentiviral constructs were transduced into Cas9-expressing JEKO-1 cells and effects of CEACAM1 knockout on cell survival were analyzed by flow cytometry. CEACAM1 and FLNA shRNA pKLO constructs were obtained from Sigma, St. Louis, MO (Supplementary Data 7) and subcloned into a GFP-expressing lentiviral vector by replacing the puromycin $N$-acetyl-transferase gene (puro) with the fusion puro-eGFP insert at the BamHI and KpnI restriction sites.

## Cell viability measurement
Cell viability was assessed by flow cytometric analysis for the propidium iodide (PI) negative population. In some experiments, shRNA knockdown vectors co-express eGFP and the fractions of PI-, eGFP+ were analyzed by flow cytometry and compared with the early time fractions. A reduction in the GFP+ fractions over time indicates reduced cell viability.

## Immunoblotting
Cells were lysed in RIPA buffer (1% Triton X-100, 0.1% SDS, 20 mM Tris-HCl, 150 mM NaCl, and 1 mM EDTA) in the presence of protease inhibitor cocktail (Sigma, St. Louis, MO) and Halt phosphatase inhibitor cocktail (Thermo Fisher Scientific) for 30 min. Lysates were cleared by centrifugation, and protein concentrations were determined by BCA protein assay (Pierce Biotechnology). Twenty micrograms of lysates per lane were separated by 4–15% SDS-PAGE and immobilized on the nitrocellulose membranes (Thermo Fisher, Waltham, MA) for immunoblotting. Immunoblot signals were developed by a chemiluminescent detection method (Thermo Fisher Scientific) and captured by standard autoradiographic films. The following primary antibodies were used: mCeacam1 (clone CC1; Invitrogen), p-Y (4G10)

(Thermo Fisher), CD79A, CD79A pY188, LYN, and SRC pY416, SYK pY352, SYK pY525/526, PLCG1 pY783, ERK pT202/Y204, AKT pS473 SYK, PLCG1, ERK, AKT (Cell Signaling Technology, Danvers, MA), CEACAM1 (E-1), LYN, FLNA, and GAPDH (Santa Cruz Biotechnology, Dallas, TX). The following secondary antibodies were used: goat anti-mouse IgG2a, human ads-HRP, goat anti-mouse IgG2b, human ads-HRP, goat anti-mouse IgG1, human ads-HRP, goat anti-mouse IgG3, human ads-HRP, and goat anti-rabbit Ig, human ads-HRP (SouthernBiotech, AL, USA).

## Immunohistochemistry
Formalin-fixed paraffin-embedded tissue blocks were sliced into 5-μm-thick sections on positively charged glass slides. The slides were loaded on the Ventana IHC automated stainer (Ventana Medical Systems, Roche Diagnostics, Indianapolis, IN) for deparaffinization, rehydration, endogenous peroxidase activity inhibition, and antigen retrieval. Anti-CEACAM1 rabbit monoclonal antibody (clone EPR4049, Abcam, Cambridge, MA) was then added to the slides, followed by DISCOVERY anti-Rabbit HQ and DISCOVERY anti-HQ HRP detection system (Ventana Medical Systems). The stains were visualized using the DISCOVERY ChromoMap DAB Kit and counterstained with hematoxylin (Ventana Medical Systems). CEACAM1 staining was reviewed by two expert hematopathologists (JYS and WCC), and positive staining was determined as >50% of the tumor cells staining. Regarding the intensity, there were cases that had decreased intensity, which were scored as weak.

## Ceacam1 deficient and SOX11/CCND1 transgenic mice
Ceacam1$^{-/-}$ mice were kindly provided by Nicole Beauchemin (Rosalind and Morris Goodman Cancer Centre, Montreal, QC, Canada)[19]. Double SOX11/CCND1 transgenic mice were generated in Dr. Samir Parekh's laboratory. Mice were maintained under approved IACUC protocols 11046 and 22054 and housing conditions including a 12-h light/dark cycle with a temperature range of 20–24 °C and humidity of 45–65%. Approximately 20–25-week-old mice were used for analysis.

## Xenograft study
Mouse care and experimental procedures were performed in accordance with institutional guidelines and approved protocols from the Institutional Animal Care and Use Committee (IACUC) of the City of Hope (protocol 11046). Two days after transduction, two million control or CEACAM1 shRNA-transduced JEKO-1 cells were intravenously injected via the tail vein into 6–8-week-old, gender-matched immunodeficient NSG mice (Jackson Laboratories, Bar Harbor, ME). Early tumor growth was monitored weekly by whole-body bioluminescence imaging using a Spectral LagoX camera and analyzed with Aura software (Spectral Instruments Imaging, Tucson, AZ). To ensure that humane endpoints were not exceeded during tumor growth at later stages (after day 28), mice were monitored daily for common signs of pain and distress such as weight loss, labored breathing, hunched posture, lethargy, reluctance to move, or limb paralysis.

## Intracellular Ca²⁺ measurements
Cells were loaded with 1 μM of Indo-1 (Thermo Fisher Scientific) in HBSS medium at 37 °C for 30 min then washed once and rested (at 37 °C for 15 min) before analysis using a LSRII flow cytometer (Becton

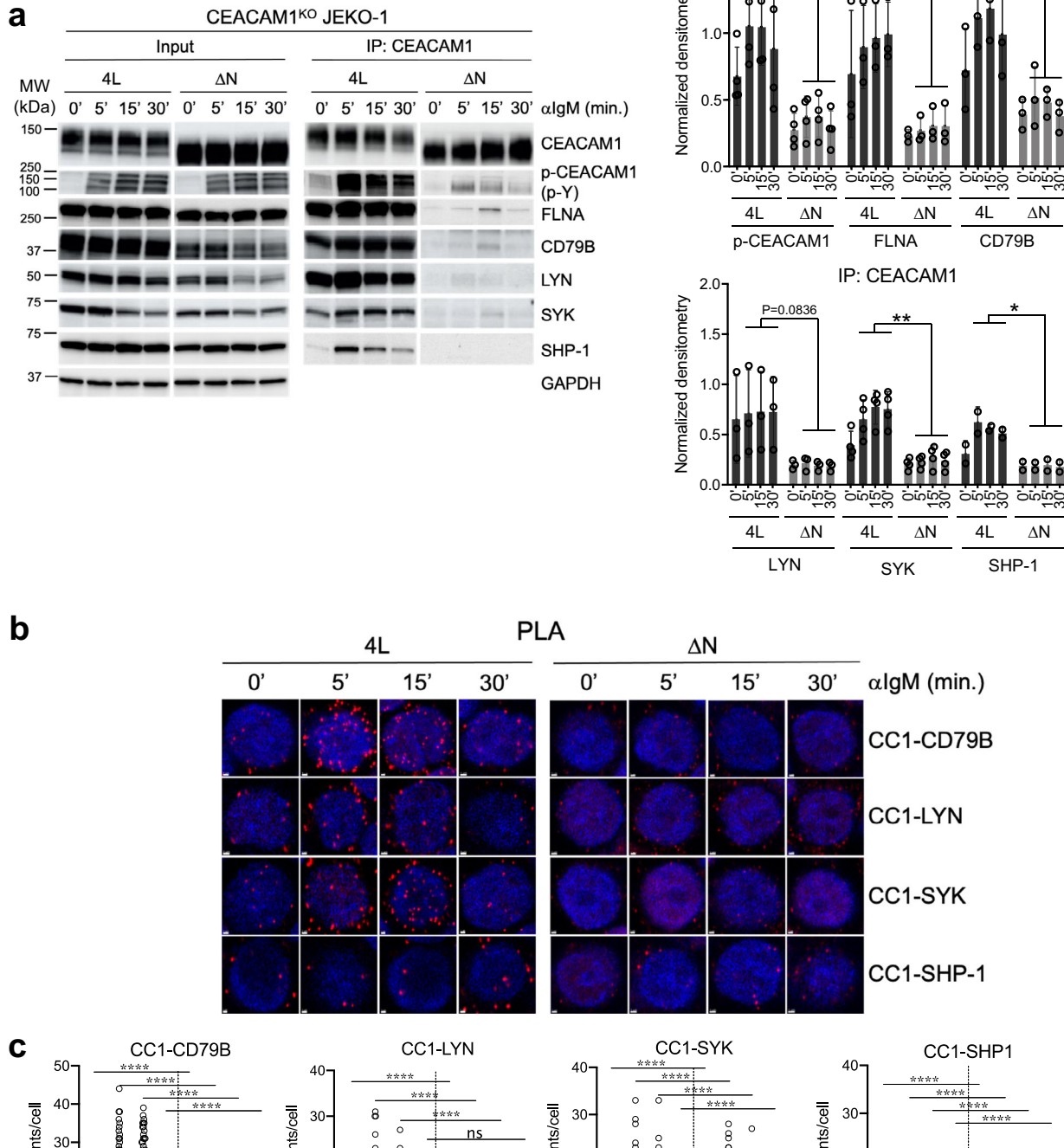

Dickinson). In some experiments, cells were also stained with fluorescently conjugated antibodies for surface marker detection at RT for 20 min and equilibrated at 37 °C for 15 min before flow cytometric acquisition. Baseline fluorescence was recorded for 1 min followed by the addition of anti-IgM antibody and continuous recording for an additional 4 to 5 min. Changes in intracellular $Ca^{2+}$ concentration were calculated as the ratio of Indo-1 emission wavelengths at 420 nm

(bound calcium) and 510 nm (free calcium). Data were acquired using FlowJo Kinetics function and plotted against time and arithmetically normalized before display.

**Lipid-raft isolation**

Fifty million cells at $10^6$/ml were stimulated with 1 µg/ml anti-IgM F(ab')₂ antibody and immediately centrifuged at 900×*g* for 4 min.

**Fig. 6 | CEACAM1 interactions with BCR signaling components require an intact N-domain. a** Left panel, Immunoprecipitation analysis of CEACAM1 interactions. CEACAM1-knockout JEKO-1 cells were transduced with either full-length (4 L) or N-domain truncated (ΔN) CEACAM1 constructs, stimulated with 2 μg/ml anti-IgM antibody for the indicated times, followed by CEACAM1 immunoprecipitation and immunoblotting with the indicated antibodies. One percent of the total lysates was used as an input control. p-Y, anti-phosphotyrosine antibody clone 4G10. Right panels, Quantification of indicated co-IP signals shown in the left panels. Bar graphs show the means and individual densitometric values from four independent experiments for each timepoint normalized to the CEACAM1 pull-down signals. Error bars, SD. **$P < 0.01$, *$P < 0.05$ by two-way ANOVA. **b** Proximity ligation assay (PLA) showing interactions (visualized as red dots using Airyscan FAST 2D confocal microscope and a 40x/1.2NA water objective) between 4 L or ΔN and the indicated proteins. **c** Quantitation of PLA signals shown in (**b**) for 100–300 cells on average from three independent experiments using QuPath 0.3.2 software. Error bars indicate means with S.D. ****$P < 0.0001$ by two-sided Mann–Whitney U-test. ns not significant. Source data are provided as a Source Data file.

Whole cell lysate (WCL) was then prepared by adding 0.5 ml 1x PBS to the cell pellet and mixing with 0.5 ml lysis buffer (0.5% Triton X-100, 10 mM Tris-HCl, 50 mM NaCl, 10 mM EDTA and protease/phosphatase inhibitors) at 4 °C for 10 min followed by brief (10 s) sonication. For the discontinuous sucrose gradient, 1 ml of WCL was mixed with 1 ml of 85% sucrose in the lysis buffer without detergent (TNE) and transferred to the bottom of a 14 × 89 mm centrifuge tube. Diluted WCL was overlaid with 4 ml of 35% sucrose in TNE and followed by 2 ml of 5% sucrose in TNE. The samples were centrifuged at $200,000 \times g$ at 4 °C for 16 h, after which 1 ml fractions were collected from the top of the tube. For immunoblot analysis, 30 μl from each fraction was mixed with 10 μl of 4x SDS sample buffer before denaturing by boiling and separation by SDS-PAGE gel electrophoresis.

### Confocal immunofluorescence microscopy and quantitative image analysis

One million cells were grown on 35 mm poly-L-lysine-coated Nº1.5 glass bottom dish (MatTek Corp., Ashland, MA) at 37 °C in CO$_2$ incubator for 1 h and stimulated with 2 μg/ml anti-IgM F(ab')2 antibody (Jackson ImmunoResearch Laboratories Inc., PA, USA) for 4 min followed by fixation in 1.5% paraformaldehyde and permeabilization in 0.1% Triton X-100 for 15 min each at RT. Fixed and permeabilized cells were stained with primary antibodies for 2 h at RT in a humidified chamber. F-actin was detected using Alexa Fluor 488-conjugated phalloidin (Cytoskeleton, Inc., Denver, CO). For lipid raft staining, cells stimulated with anti-IgM (5 μg/ml) were labeled with CT-B reagent (Vybrant™ Alexa Fluor™ 594 Lipid Raft Labeling Kit, Invitrogen) for 10 min on ice followed by fixation and permeabilization and incubation with CEACAM1 (EPR4049) rabbit antibody (Abcam, MA, USA) overnight at 4 °C in a humidified chamber. After washing 3x in 1x PBS, cells were stained with fluorescently labeled secondary antibodies for 1 h at RT, followed by washing twice in 1x PBS and nuclear staining with 4′,6-diamidino-2-phenylindole (DAPI) (Thermo Fisher Scientific) for 10 min at RT. Fluorescent images were acquired on an LSM 880 confocal microscope (Leica Microsystems Inc., IL, USA) in Airyscan mode using a 20×0.8NA objective for 2D collections or a 40×1.2NA objective for 3D reconstructions. On average, 30 z-slice optical sections were collected with a step size of 3 microns. Image acquisition was controlled with the ZEN 2.3 SP1 Black Edition software (Carl Zeiss Microscopy); brightness and contrast were identically adjusted for images acquired in the same experiments. Three-dimensional image reconstructions were visualized using the Imaris 9.5 software (Oxford Instruments, Zurich, Switzerland). Quantitative image analysis of the 20x 2D images was conducted using QuPath V0.2.0m6, an open-source bioimage analysis software[62]. Briefly, nuclei were first detected using Positive Cell Detection. Cells were then manually curated to remove bubbles or other extremely high DAPI signals as artifacts. Within the true cells, Subcellular Detection was performed to generate a measurement of signal-positive area. The sum of the total intensities in all signal-positive areas was generated per cell for plotting. The scripts for image analysis are available in the Supplementary Software.

### Proximity ligation assay

Proximity ligation assay (PLA) was performed using Duolink® In Situ Detection Reagents Red (Sigma-Aldrich, MO, USA) following the manufacturer's instructions. Briefly, one million cells were seeded on 35 mm poly-L-lysine-coated Nº1.5 glass bottom dish (MatTek Corp., Ashland, MA) at 37 °C in a CO$_2$ incubator for 1 h and stimulated with 5 μg/ml of AffiniPure F(ab')$_2$ fragment donkey anti-human IgM, Fc5μ fragment specific antibody (Jackson ImmunoResearch Laboratories Inc, PA, USA) at 37 °C for the indicated time points (0, 5, 15, or 30 min). Cells were fixed using 1.5% paraformaldehyde for 15 min at RT and permeabilized using 0.1% Triton X-100 for 15 min at RT followed by blocking using Duolink® Blocking Solution for 1 h at 37 °C. Cells were incubated overnight in a humidified chamber at 4 °C with a combination of two primary antibodies of interest. Cells were then washed twice with 1X Duolink® Wash Buffer A for 3 min each and subjected to PLA labeling and amplification procedures according to the manufacturer's recommendations. Briefly, the PLUS and MINUS PLA probes were diluted 1:5 in Duolink® Antibody Diluent. The PLA PLUS or MINUS probes were prepared using Duolink® In Situ PLA® Probe Anti-mouse PLUS or Anti-rabbit MINUS (Sigma-Aldrich) kit. Two microlitres of Conjugation Buffer was added to 20 μL of the antibody (1 mg/mL of AffiniPure Donkey Anti-Mouse IgG (H + L) or AffiniPure Donkey Anti-Rabbit IgG (H + L), Jackson ImmunoResearch Inc.). The antibody solutions were transferred to one vial of lyophilized oligonucleotides each (PLUS or MINUS) and incubated at RT overnight. Two microlitres of Stop Reagent was added to the reaction and incubated at RT for 30 min followed by the addition of 24 μL of Storage Solution. From this preparation, 12.5 ng/μl of PLUS and MINUS probes were diluted in probe diluent (1:5 dilution) and added to the cells and incubated for 1 h at 37 °C. After the incubation, cells were washed twice with 1X Duolink® Wash Buffer A (3 min each). For the ligation step, 5x Duolink® Ligation buffer was prepared in high-purity water at 1:5 dilution and ligase enzyme (1:40 dilution) was added to the mixture, which was added to the cells and incubated for 30 min at 37 °C. Cells were then washed twice with 1X Duolink® Wash Buffer A (3 min each). For the amplification step, 5x Amplification buffer was prepared in high-purity water at 1:5 dilution and polymerase enzyme (1:80 dilution) was added and mixed well. Cells were incubated with the amplification mixture for 1.5 h at 37 °C in dark. After the incubation, cells were washed twice with 1X Duolink® Wash Buffer B (3 min each). Cell nuclei were stained using DAPI and imaged using LSM 880 with Airyscan FAST confocal microscope (Leica Microsystems Inc., IL, USA). PLA puncta in all images were analyzed with cell and subcellular spot detection using QuPath V.0.3.2 and representative ROIs were generated using Zen Blue 3.1. Additional information regarding reagents used in this study is listed in Supplementary Data 8.

### Immunoprecipitation

Cells ($3-4 \times 10^7$ per condition) were lysed in 1 ml of IP buffer (50 mM Tris-HCl, 150 mM NaCl, 1% Triton X-100, and 2 mM EDTA) supplemented with protease inhibitor cocktail (Sigma-Aldrich, MO, USA) and phosphatase inhibitor (Thermo Fisher Scientific, MA, USA) on ice for 30 min. Lysates were sonicated at 15 joules. For 10 s and cleared by centrifugation at $14,000 \times g$ for 15 min at 4 °C. One percent of the lysate (~10 μl) was saved as input and the rest was incubated with 1 μg of anti-CEACAM1 antibody (Santa Cruz Biotechnology, TX, USA) or isotype control antibody (SouthernBiotech, AL, USA) together with 30 μl of protein G Sepharose beads (Pierce Biotechnology, MA, USA) by rotation at 4 °C for 8 h. Antibody-bound beads were washed six times in

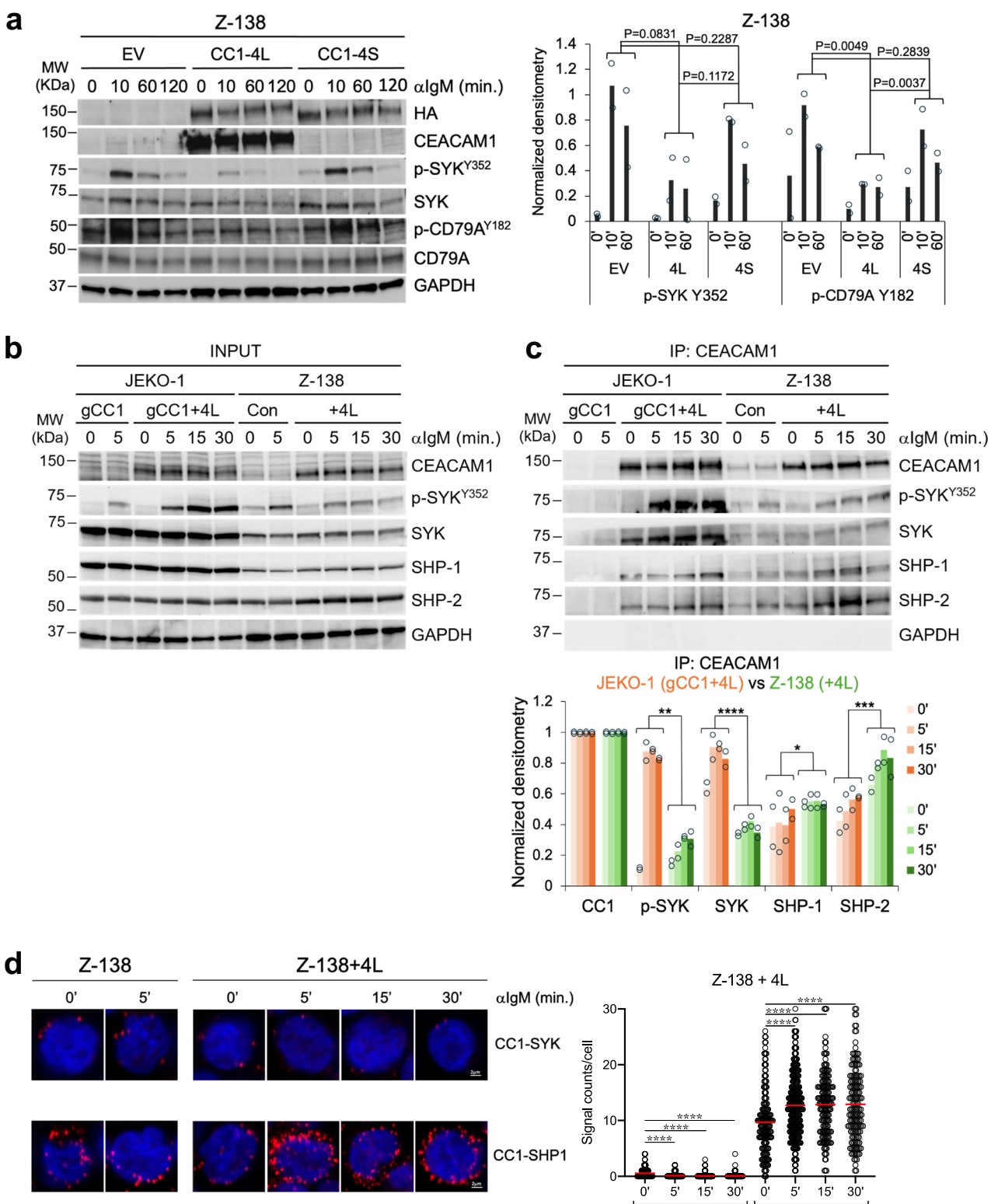

1 ml of IP buffer by centrifugation at 8000 × *g* for 1 min at 4 °C. Washed Sepharose beads were resuspended in 1x sample buffer containing 5% beta-mercaptoethanol, heat-denatured for 5 min at 95 °C, and analyzed by SDS-PAGE and immunoblotting.

**Quantitative real-time PCR**
Quantitative real-time PCR (qPCR) reactions were performed using Taqman Universal PCR Master Mix (Thermo Fisher Scientific, Waltham,

MA) and analyzed by the QuantStudio 7 Real-Time PCR System (Thermo Fisher Scientific, Waltham, MA). Taqman probes for CEACAM1 long variants (Hs00989786_m1), CEACAM1 short variants (Hs00266109_m1), CEACAM3 (Hs00174351_m1), CEACAM5 (Hs00944025_m1), CEACAM6 (Hs03645554_m1), and GAPDH (Hs02786624_g1) were purchased from Thermo Fisher. Relative mRNA expression was normalized to GAPDH signals and calculated using the ddCt method.

**Fig. 7 | A dual role of CEACAM1 in BCR signaling. a** Left panel, CEACAM1 suppresses BCR signaling in Z-138 cells. Z-138 cells were transduced with control empty vector (EV), WT CEACAM1 (CC1-4L), or short cytoplasmic tail CEACAM1 (CC1-4S). Transduced cells were stimulated with 2 μg/mL of anti-IgM F(ab')2 fragments for the indicated times, followed by immunoblot analysis probed with the indicated antibodies. HA, hemagglutinin, a protein tag in-framed with CEACAM1 to detect the 4S isoform. Right panel, Quantification of the immunoblot signals shown in the left panel. Bar graphs show the means of densitometric values from the indicated time points normalized to GAPDH loading controls from two independent experiments. *P* values, one-sided permutation test. **b, c** CEACAM1-knockout JEKO-1 cells (gCC1) or Z-138 cells were transduced with either empty vector control or full-length (4 L) CEACAM1 construct, stimulated with 2 μg/ml anti-IgM antibody for the indicated times, followed by CEACAM1 immunoprecipitation and immunoblotting with indicated antibodies. One percent of the total lysates was

used as an input control (**b**). The samples shown in (**b**) derive from the same experiment, but different gels for CEACAM1, p-SYK$^{Y352}$, SYK, SHP-1, GAPDH, and another for SHP-2 were processed in parallel. Bar graphs show the quantification of the co-IP signals shown in (**c**). Shown are the means of densitometric values from two independent experiments for each timepoint normalized to the CEACAM1 pull-down signals. ****$P < 0.0001$, ***$P < 0.001$, **$P < 0.01$, *$P < 0.05$ by two-way ANOVA. **d** Left panels, Proximity ligation assay (PLA) showing interactions (visualized as red dots using Airyscan FAST 2D confocal microscope and a 40x/1.2NA water objective) between CEACAM1 (CC1) and the indicated proteins in Z-138 cells or Z-138 cells transduced with CEACAM1-4L followed by stimulation with 2 μg/ml of anti-IgM antibody for the indicated times. Right panels, Quantification of PLA signals shown in the right panels for -200 cells on average from three independent experiments using QuPath 0.3.2 software. ****$P < 0.0001$ by a two-sided unpaired *t*-test with Welch's correction. Source data are provided as a Source Data file.

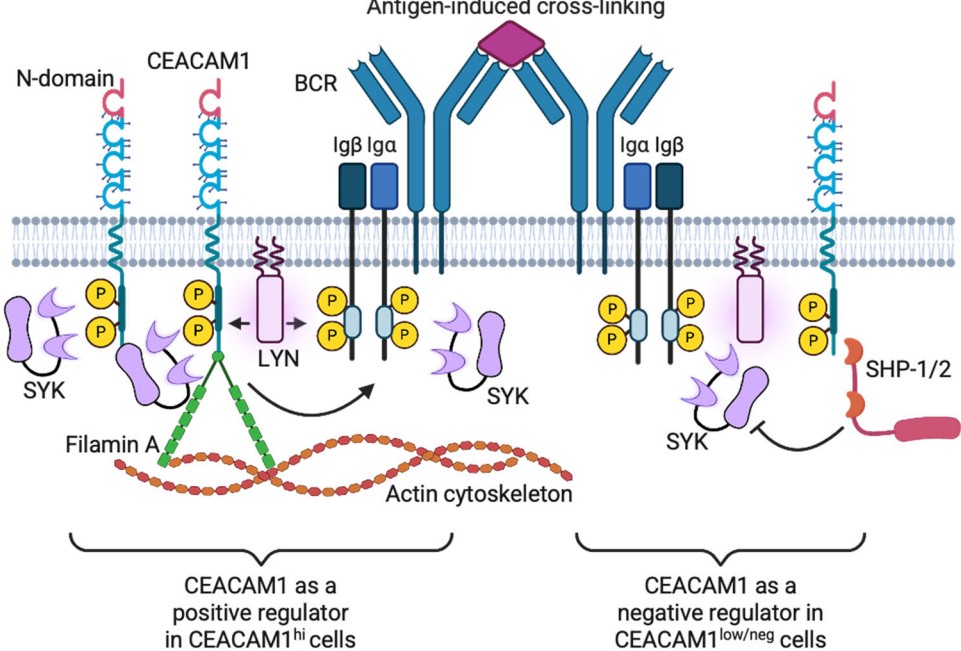

**Fig. 8 | Proposed model for CEACAM1 function in BCR signaling.** Following antigen stimulation in B-cell lymphomas with abundant CEACAM1 expression, CEACAM1 binds to FLNA, which anchors to the actin cytoskeleton and lipid rafts, and recruits SYK to the proximity of CD79A/B to enhance BCR activity. In cells with

low or no CEACAM1 expression (and potentially low SYK expression), SHP-1 and SHP-2 outcompete for CEACAM1 binding, leading to signal attenuation. Created in BioRender. Ngo, V. (2025) https://BioRender.com/dl16wxh.

## CEACAM1 expression vectors

CEACAM1 expression vectors were constructed by replacing the luciferase-IRES-ZsGreen cassette from the lentiviral vector pHIV-Luc-ZsGreen (Addgene #39196) with the CEACAM1-IRES-Neo cassette using Gibson cloning at the EcoRI and ClaI cut sites. PCR primers for generating Gibson fragments corresponding to CEACAM1-4L, 4S, 4L-Y493F/Y520F, IRES, and Neo are listed in Supplementary Data 9. Mutations (Y→F) at the two ITIM tyrosine residues were accomplished by site-directed mutagenesis (Stratagene, San Diego, CA) following the manufacturer's instructions and confirmed by Sanger sequencing. The final CEACAM1 constructs were confirmed by DNA sequencing and validated by flow cytometry and immunoblotting.

## Statistics and reproducibility

This study aims to understand the mechanisms of CEACAM1 in supporting BCR signaling in MCL. The study was designed to confirm the CRISPR screen results and investigate the precise molecular mechanisms of CEACAM1 interactions with BCR-proximal signaling elements. CEACAM1 as an essential gene in MCL was confirmed by gene knockout or knockdown using CRISPR or shRNA in vitro and in vivo. CEACAM1 mRNA and protein expression were compared in MCL, normal B

cells, and other B-cell tumors, using gene expression profiling, immunoblotting, immunohistochemistry, and flow cytometry on primary tumor samples and cell lines. The mechanistic roles of CEACAM1 in promoting BCR signaling was investigated by introducing wildtype or mutant CEACAM1 constructs into JEKO-1 cells from which the endogenous CEACAM1 had been knocked out. The activities of BCR-proximal kinases were examined by immunoblotting, and BCR activation was assessed using Ca²⁺ flux assays. To investigate CEACAM1 interactions with proteins proximal to BCR signaling environment, we used the lipid raft fractionation method, confocal immuno-fluorescence microscopy, immunoprecipitation, and Proximity Ligation Assay on the MCL cell lines JEKO-1, MINO, and Z-138. All experiments were repeated at least twice.

Data were shown as mean ± s.d. unless otherwise stated. Statistical analyses were performed using GraphPad Prism 8 (GraphPad Software Inc., La Jolla, CA). A paired, two-sided (unless otherwise stated) *t*-test, Mann–Whitney *U*- test, or permutation test was used to analyze statistical significance between two groups. Two-way ANOVA was used for comparing groups defined by two independent variables. Significance was considered at $P < 0.05$. Kaplan–Meier survival analysis was used to estimate the overall survival of mice, and a log-rank test was used to

compare the difference between the two groups. No statistical method was used to predetermine the number of animals. No data or animals were excluded from the analyses.

### Reporting summary

Further information on research design is available in the Nature Portfolio Reporting Summary linked to this article.

## Data availability

All data are available in the main text and the Supplementary Information. Source data are provided with this paper.

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

## Acknowledgements

We would like to thank Nicole Beauchemin for sharing *Ceacam1*$^{-/-}$ mice and John E. Shively for critical reading of the manuscript. This article is dedicated to the memory of Dr. Bernhard B. Singer, whose inspiration and unwavering support sustained the study. National Institute of Health grant P30CA033572 (Cancer Center Support Grant to City of Hope). National Institute of Health grant P50CA107399 (Lymphoma SPORE Developmental Research Program, VNN). National Institute of Health grants R03CA245996, R01CA262754 (VNN). Concern Foundation grant (VNN). The Leukemia Lymphoma Society Career Development grant (VK). The Alex's Lemonade Stand Foundation grant (VK). The Lymphoma Research Foundation grant (TS). Howard Hughes Medical Institute grant (MM). National Cancer Institute grants R01CA252222, R01CA244899, R01CA262754, K12CA270375, and CA196521 (SP). National Research Foundation of Korea (NRF) grant RS-2024-00339966 (JL).

## Author contributions

Conceptualization, supervision, and writing—original draft: V.N.N. and M.M. Methodology: S.X., V.K., V.N., A.P., L.Y., J.W., M.S.N. and A.L. Investigation: S.X., V.K., V.N., A.P., L.Y., M.S.N., R.P.S., A.L., J.Y.S., S.P., and V.N.N. Visualization: S.X., V.K., V.N., A.P., and V.N.N. Data curation: L.Y. Resources: J.L., L.V.P., D.D.W., W.C.C., K.S.L., G.P.S., A.V.D., J.Y.S., and T.S. Formal analysis: L.Y. and H.G. Software: M.S.N. Writing —review and editing: S.X., V.K., M.S.N., J.L., T.S., D.D.W., W.C.C., and M.M., and V.N.N.

## Competing interests

S.P. serves on the Advisory Board at Grail, consults for Regeneron, Genentech/Roche, and Poseida Therapeutics, and receives research support from Grail, Celgene/BMS Corporation, Caribou, imCORE, and Poseida Therapeutics. The remaining authors declare no competing interests.
