## [Transparent Peer Review file · Nature Communications]

CEACAM1 as a Mediator of B-cell Receptor Signaling in Mantle Cell Lymphoma

Corresponding Author: Dr Vu Ngo

Version 0:

Reviewer comments:

Reviewer #1

(Remarks to the Author)

The paper identifies CEACAM1 as a novel amplifier of BCR signalling with a specific role in MCL. Since BCR signalling has been associated with the pathogenesis and clinical outcome of these non-Hodgkin lymphomas, but in a distinct role compared to other B cell cancers, the results are novel and interesting. They have implications for understand how the BCR signalling drives malignant cell growth specifically in this disease. They could also be important for the responses of normal B cells and contribute to the discussion about the complexity of mechanisms (positive and negative) through which CEACAM proteins regulate kinase signalling pathways. The manuscript is generally well presented and contains abundant data supporting most of the conclusions.

Strong points of the paper:

MCL primary cells and cell lines typically have higher CEACAM1 expression compared to normal B cells or other types of non-Hodgkin lymphoma.

Knocking out or down CEACAM1 reduces MCL cell line growth or survival.

Knocking out or down CEACAM1 reduces BCR signalling in MCL lines at the earliest steps of BCR-proximal kinase activation.

CEACAM1 recruits Syk and filamin A and this requires intracellular tyrosines and the N-terminal ligand binding domain.

In non-MCL cell lines, CEACAM1 recruits SHP1 and inhibits BCR signalling.

Overall the data make a strong case for the importance of CEACAM1 overexpression in MCL augmenting BCR-dependent pathogenesis.

However, there are a few major and a number of minor points that should be addressed before publication.

1. The conclusions that CEACAM1 “stabilizes membrane microdomains (lipid rafts) by anchoring to the F-actin cytoskeleton through adaptor protein filamin A” or that it “orchestrates lipid raft dynamics “ are indirect. There is currently little evidence in the paper that the expression of CEACAM1 influences the composition of lipid rafts or recruitment of the BCR into lipid rafts in a way that would regulate BCR signalling. From Figure 4B, it seems that a small proportion of CEACAM1 (not quantified) is constitutively in raft fractions of the gradient. While this proportion may help recruit FLNA, the significance of this for BCR signalling is unclear as we do not know if FLNA is essential for BCR signalling in these cells. There are no statistical comparisons of the differences in the protein presence in the different fractions over several experiments. There is also no information on the partitioning of the BCR under these conditions. It seems that the results can be interpreted that CEACAM1 simply enhances Lyn and Syk activation, irrespective of its partial presence in the raft fractions. Fig 4C and D are even more difficult to interpret. The images are nicely quantified and substantiate the effect of CEACAM1 absence on Src phosphorylation and actin polymerisation, but they show an increase in total staining of Lyn, FLNA or GM1 after BCR crosslinking, which is difficult to interpret in terms of raft stability or dynamics. It is a bit puzzling why the intensity of these proteins/markers increases just a few minutes after BCR triggering. Most previous studies showed only changes in distribution/colocalisation. Thus, unless substantial new evidence is presented on the role of the lipid rafts and filamin A in the mechanism, I suggest toning down the above conclusions in the title and the abstract.

2. The prioritisation of CEACAM1 for these studies is logical and interesting. However, the paper does not present the actual results of the CRISPR screen beyond the dot plot in Fig 1A. A supplementary table with full-screen results is a standard in

the field and would be an important part of the paper. Similarly important is the list of genes implicated in patient survival in MCL aggregated from previous studies. Since the cited papers containing the original data do not highlight CEACAM1 themselves, it would be important to provide a reanalysis of those data to illustrate how CEACAM1 expression correlates with progression.

Fig 1 d,f and in other figures throughout the paper, the loss of CEACAM-1 targeted cells is called "loss of viability". Is this actually cell death as implied, or is it loss of proliferation and thus outgrowth of the GFP- cells?. Perhaps a more generic term is "cell growth" or "proliferation and survival".

Extended data Fig 3a - please indicate what is the green part of the sequence schemes.

Fig 3A,B. Statistics and numbers of experiments are missing. Why was PLCG1 chosen? Most BCR studies focus on PLCG2, which is more expressed in B cells.

Fig 3H - please use empty histograms, it is currently difficult to see the data for the individual cell lines.

Fig3J, presumably the WB is done with the E1 antibody, which is binding to the intracellular part? Please indicate this in the figure/legend.

Extended data Fig 5B - statistical analysis of the signalling data is needed.

Figure 5A: The data analysis does not show the statistical significance of increased CEACAM1 interactions with SYK after crosslinking, just higher values compared to SHP1. Fig 7D suggest that the SHP1 PLA is working in a different cell line, but the interpretation of Fig5A would require a positive verification that the SHP1 signal is meaningful compared to some control.

Figure 5C, D. The IP shows that there is an increase in p-SYK associated with CEACAM1 after anti-IgM. However, there seems to be a substantial amount of non-phosphorylated Syk bound to CEACAM1 before stimulation. This correlates with basal CEACAM phosphorylation in Fig 6. Perhaps this is worth highlighting. However, the statistical comparisons focus on comparing the pSYK to SHP1 data, which is not a good comparison as they are developed using different antibodies. A better comparison is to compare to the unstimulated control. It is also very unusual to use a nested t-test for the analysis. Two-way ANOVA pairing the control and stimulation in each experiment is more typical in these settings. While the slow kinetics of SHP1 are interesting, they are not reproduced in Fig 6A.

Figure 6A - There is no statistical analysis of the IP results.

Extended data 7 and 6A - there are multiple bands in the phospho-CEACAM blot in lysate but only one in the IP. What is the antibody used here?

Figure 7 - The results with the N-terminal deletion are very interesting. What is the expression level of this construct on the cell surface compared to 4L? Can the authors speculate on whether they think that cis or trans interactions are involved? Is there any role of the CEACAM1 in the homotypic adhesion of the cells?

Figure 7 - The preferential recruitment of SHP1 in CEACAM1-low cells is also interesting. Not explained, but it opens new ideas. However, the comparisons on the WB seem selective. Again, possibly ANOVA with appropriate pairing would be less biased.

The cartoon is helpful, but also potentially misleading. I don't think there is any evidence that Syk or SHP1 bind using both their SH2 domains to the pairs of phospho-tyrosines. There is also a possibility that Syk binds to the phosphotyrosine in a hemITAM fashion, i.e. to a CEACAM1 dimer (10.1126/scisignal.aan3676).

(Remarks on code availability)

Reviewer #2

(Remarks to the Author)

In this manuscript by Xavier et al. the authors study the role of CEACAM1, a transmembrane protein and adhesion molecule, for signaling downstream of the B cell receptor in mantle cell lymphoma. The authors show that CEACAM1 stabilizes lipid rafts via anchoring to the cytoskeleton via filamin A. CEACAM1 also recruits SYK to the BCR complex. In careful in vitro work the authors have, therefore, unravelled novel functions of CEACAM1 in particular with regards to its signaling and its activating role for signaling downstream of the BCR.

The manuscript is well written and the experiments are carefully performed. However, any evidence that this pathway is playing a role in vivo is lacking. Is there an inhibitor of CEACAM1? What are the effects on signaling if an inhibitor of filamin A, if available, is employed?

Other points are:

1. Is there a relevance of CEACAM1 as a tumor marker, similar to CEA?
2. Figure 1f: Does CEACAM1 play a role in any chromosomal translocation events as frequently found in lymphomas?

3. Please include in vivo experiments to prove the relevance for lymphoma progression.
4. Do CEACAM1-deficient lymphoma cells proliferate less? Cell cycle? Apoptosis? Are these signaling studies relevant for studying proliferation?
5. Is a CEACAM1 knockout murine model available for further studies and what is this mouse's phenotype?
6. Is there a correlation of CEACAM1 with CD5 expression? I.e. does CEACAM1 also play a role in CLL cells?
7. Figure 2f: This is a comparison of very different entities, in which CEACAM1 may play very different roles. It would be helpful to add CLL here.

Minor points:

1. It seems as if figure 7c is cited out of order.
2. It may be helpful to add changes of the cytoskeleton to figure 8.
3. It is suggested to organize the results section in such a way that one subsection with a given result corresponds to one figure. Currently, the figure number changes within a subsection.

(Remarks on code availability)

Reviewer #3

(Remarks to the Author)

Overall Review: In this manuscript, Xavier & coauthors, using a comprehensive approach combining genome-wide CRISPR loss-of-function screens, gene expression profiling and BCR-signal transduction studies, have identified CEACAM1 as a critical factor for mantle cell lymphoma (MCL) proliferation and survival, playing a pivotal role in B-cell receptor (BCR) signaling. Their findings demonstrate that CEACAM1 activates BCR signaling by binding the adaptor protein filamin A (FLNA) which anchors them to the F-actin cytoskeleton and stabilizes lipid rafts. This process enhances the recruitment of LYN and SYK within the BCR complex, ultimately leading to robust BCR activation. The N-terminal as well as the C-terminal domains of CEACAM1 are essential for its interaction with FLNA, CD79B, LYN and SYK, promoting downstream phosphorylation, positively contributing to the early stages of BCR signaling by activating LYN and SYK. The strengths of this manuscript include the genome-wide CRISPR loss-of-function screens, incorporation of publicly available RNA-seq and clinical data and well designed experiments. Weakness include limited data in primary MCL samples, particularly related to CEACAM1 mechanism of action enhancing the BCR-signaling, with mechanistic data restricted to a single MCL cell line with no (functional) validation in the context of the tumor microenvironment's influence or in vivo therapeutic implications. Below are suggestions to improve the publication:

MAJOR COMMENTS:

1. CEACAM1-dependent MCL viability was validated using CEACAM1-shRNA across various lymphoid cell lines. However, CEACAM1 silencing appears to impact viability only in lymphoid cell lines with high CEACAM1 expression, all of which are MCL-derived. Notably, these results have not been compared between MCL cell lines with low CEACAM1 expression (e.g., Z138) and those with high expression (e.g., REC1) (Fig. 2d).
2. Tumor engraftment and in vivo viability in MCL xenograft mouse models were significantly reduced upon CEACAM1-silencing. While these findings are intriguing, the experiments were conducted exclusively with the JEKO-1 cell line. Given the important heterogeneity observed among MCL cases, it is essential to include additional cell lines to validate the in vivo effects of CEACAM1 overexpression on MCL progression and viability.
3. CEACAM1 is highly expressed in the majority of MCL tumors compared to other lymphoid neoplasms and normal B cells. Notably, CEACAM1 has been shown to be more highly expressed in SOX11+ MCL than in SOX11- MCL (Balsas P, et al., Blood, 2021). It would be valuable to investigate the correlation between SOX11, which is exclusively expressed in MCL, and CEACAM1 expression across different MCL cell lines and primary samples. Additionally, examining the impact of ectopic SOX11 overexpression on CEACAM1 expression in JVM2 cell lines could provide further insights. Finally, analyzing survival dependency in CEACAM1+ versus CEACAM1- JVM2 cell lines would help clarify the functional relationship between them.
4. BCR-mediated Ca²⁺ signaling was significantly reduced upon CEACAM1 depletion in MAVER-1 cells. However, cell survival and sensitivity to Ibrutinib remained comparable between CEACAM1-high MAVER-1 and the CEACAM1-low Z138 cell lines. To establish whether Ibrutinib sensitivity is influenced by CEACAM1 expression, I recommend: (1) Conduct similar experiments using CEACAM1-silenced JEKO-1 cell lines and (2) comparing CEACAM1-low and CEACAM1-high MCL cells derived from primary samples. (3) Perform analogous studies in Ibrutinib-resistant cell lines, such as Z138 or JVM2, to explore potential differences in sensitivity and resistance mechanisms.
5. Results shown in Fig. 3 and 4 should be confirmed in more than one MCL cell line and comparing CEACAM low/high MCL cells from primary samples.
6. CEACAM1 also can regulate negatively the BCR signaling in lymphoma cells with low or absent CEACAM1 expression by preferentially binding to SHP-1/2 instead of SYK. However recruitment of SHP-1 is high in MCL cell lines expressing high levels of CEACAM1 (Fig 5a). To clarify this hypothesis I recommend: (1) conduct similar experiments and analyze the levels of SHP-1 bound to CEACAM1 in CEACAM1-silenced cell lines and (2) comparing CEACAM1-low and CEACAM1-high MCL cells derived from primary samples.
7. CEACAM1 has recently been identified as an immune checkpoint in various tumors. To further validate its functionality in MCL, BCR-signaling studies in 3D coculture systems could provide insights into how the tumor microenvironment influences CEACAM1 activity and its potential role as an immune checkpoint in MCL. Additionally, comparing CEACAM1 expression in primary samples from peripheral blood and lymph nodes could help determine whether its expression is crucial for interactions between MCL cells and the tumor microenvironment or how its expression is influenced by the

microenvironment.

8. CEACAM1 inhibitory experiments with already described blocking antibodies in in vitro and in vivo MCL models would demonstrate its therapeutic implications in MCL.

MINOR COMMENTS:

1. It is difficult to follow the text because the data is not there or it is not displayed under the correct name. e.g. I couldn't find Extended Data 3a, b, c. In extended Data 5, I only see 5a. Fig. 5a&c, showing significantly reduced both Ca²⁺ signals and cell survival in JEKO-1 cell are not shown in the corresponding Extended data 5. I couldn't find Extended Data Fig. 6.
2. CEACAM1-4S it is not detectable in JEKO-1 cells+gCC1 cell lines (Fig 3 j).

(Remarks on code availability)

Reviewer #4

(Remarks to the Author)

In this manuscript by Xavier et al. entitled "CEACAM1 Orchestrates Lipid Raft Dynamics and B-cell Receptor Signaling in Mantle Cell Lymphoma" the authors report about a crucial role of CEACAM1 in the modulation of B cell receptor signaling especially in mantle cell lymphoma cells.

They provide convincing evidence for an upregulation of CEACAM1 in lymphoma cells. Furthermore, they demonstrate that some signaling mediators are differentially affected by IgM stimulation in the context of an altered CEACAM1 expression. Nevertheless, the study suffers from some inconsistencies with regard to the obtained data. Additionally, conclusions are drawn that are not entirely justified by the methods used.

Specific concerns and comments:

The authors claim that SYK is the kinase that is involved in CEACAM1-mediated modulation of BCR signaling. In Fig.3, 5 and 7 they therefore analyzed SYK and its phosphorylation. Why did they analyze SRC-Y416 instead of SYK-Y352 in Fig.4b and Fig.4d?

Fig.6a: CEACAM1 phosphorylation is unaltered by the deletion of the N domain and although the same amount of CEACAM1 was precipitated in the 4L and ΔN experiment, no (or only less) phosphorylated CEACAM1 can be found in the precipitate of the ΔN experiment. How can this be the case?

Fig.6b,c: According to proximity ligation assay, IgM stimulation does not induce increased interaction of CEACAM1 with SHP1 in 4L-transfected cells. This looks completely different in Fig.6a – there, much more SHP1 is detected in the CEACAM1 precipitate of 4L cells (at 5 min incubation roughly 10 times higher). Which finding is right?

In Fig.6a and Fig.7c IgG controls for IP are missing.

Fig.7a: What is "HA"? No explanation anywhere.

Fig.7b: Here, it is explicitly stated that analyses were conducted on two separate membranes. This implies that all other Western blots shown, were conducted on one membrane. This is impossible because of the overlapping molecular weights of the detected proteins.

One important question: is the expression of the BCR itself affected by altered CEACAM1 expression. This needs to be clarified.

Except for Fig.5c, 5d, 7a and 7c, no Western blot is quantified! For Fig.7a and 7c statistical analysis based on TWO independent experiments (as stated in the figure legend)! How can you apply a statistical analysis on these data?

Overall, the number of animals/experiments should be clearly stated for each figure panel.

In the proximity ligation assay experiments, the number of analyzed cells covers a very wide range: Fig.5b 200-500 cells, Fig.6c 100-300 cells, Fig.7d ~200 cells. What is the rationale for this? Why are the numbers of analyzed cells different at all? Such a procedure could skew the results. Additionally, in how many independent experiments the analyzed cells were generated? 200 cells do not mean 200 independent experiments!

Regarding Fig.5c and Fig.7c: it is simply impossible to directly compare the signals of two different proteins obtained by Western blot! Beside the amount of protein, signal strength is ultimately affected by the titer, the affinity and avidity of the antibody used. These parameters are undoubtedly different concerning the used antibodies to detect SYK and SHPs. As a consequence, the lower signal of SHPs (compared to SYK) can be generated by a larger amount of SHP (compared to SYK), simply because the antibody performs less. Therefore, I think that a preference of CEACAM1 for SYK or SHPs depending on its expression level can not be deduced from these data.

(Remarks on code availability)

Reviewer #5

(Remarks to the Author)

BCR signaling is critical in MCL pathogenesis, though its mechanisms are not fully understood. In this study, Xavier et al identified CEACAM1 as essential in a subset of MCL tumors through a genome-wide CRISPR library screen. Mechanically, CEACAM1 stabilizes membrane microdomains via F-actin anchoring and enhances SYK recruitment to activate BCR signaling. These functions depend on its cytoplasmic ITIM domain and N-terminal ectodomain. Interestingly, CEACAM1, known as an inhibitory receptor, exhibits an activating role in the BCR signaling in MCL cells, providing new insights into BCR-targeting therapies in MCL.

The findings of this study are novel and significantly advance our understanding of MCL pathobiology. However, further refinements are needed to enhance the impact and value of this manuscript.

Major comments:

- 1) The authors should present a comprehensive overview of the quality of the genome-wide CRISPR screening results. Specifically, the Human GeCKO v2 library contains 1,000 control sgRNAs designed not to target any site in the genome. Data related to these control sgRNAs should be summarized and displayed in a figure for quality assessment. Additionally, the screening results for pan-dependent genes (1,682 genes) should also be presented in a figure. A good example for reference is Extended Data Fig. 1 in PMID: 29925955. Furthermore, the entire dataset of genome-wide CRISPR screening results, including Log2 fold change (Log2FC), p-values, and rankings, should be provided as a supplementary table for transparency and reproducibility.
- 2) One of the significant advantages of genome-wide CRISPR screening is the ability to understand the global gene essentiality in signaling pathways. Considering the critical role of aberrant BCR signaling in MCL cell biology, it would be important to report the CRISPR screening results for genes involved in BCR signaling, particularly IgM, CD79B, CD79A, CD19, LYN, and SYK. This analysis would provide insights into the essentiality of these genes in MCL cells.
- 3) BTK serves as a pivotal hub connecting BCR signaling to the NFκB pathway. In this context, the authors should detail the CRISPR screening results for key genes in the NFκB signaling pathway, including BTK, CARD11, BCL10, MALT1, IKBKB, IKBKG, CHUK, NFKB1, REL, IRF4, and SPIB. Reporting the essentiality of these genes would elucidate the dependency of MCL cells on NFκB signaling and highlight potential therapeutic targets.
- 4) Figure 2D is particularly intriguing as it shows that MCL cells exclusively express CEACAM1. However, the molecular mechanism underlying this higher expression of CEACAM1 in MCL cells remains unclear. Does SOX11 regulate CEACAM1 mRNA expression, potentially driving its overexpression? To provide a deeper understanding, it would be informative to include immunoblot data for SOX11, SYK, and LYN in Figure 2D
- 5) In the discussion section, the authors speculated that "high SYK expression may enable SYK to outcompete SHP-1 in binding to CEACAM1," thereby influencing CEACAM1's role in BCR activation. To substantiate this hypothesis, the authors should create SYK-overexpressing Z-138 cells and compare them with mock-transduced cells. Specifically, the levels of p-SYK and p-CD79A following BCR stimulation should be evaluated and presented in Figure 7A. This experiment would provide direct evidence for the proposed mechanism and clarify SYK's role in CEACAM1-mediated signaling.
- 6) To investigate the function of CEACAM1 in greater detail and to evaluate whether CEACAM1 and LYN share similar functions, the authors should perform RNA-seq on JEKO1 and MINO cells transduced with sgCEACAM1, sgLYN, or the control sgNTC.

Minor comments:

- 7) The meaning of "4L" in "WT CEACAM1 (4L)" is unclear. The authors should provide a detailed explanation of this term in the manuscript, ensuring clarity for the readers.
- 8) In Page 8 line 8, Fig.2d should be Fig. 2f.

(Remarks on code availability)

Version 1:

Reviewer comments:

Reviewer #1

(Remarks to the Author)

The authors addressed my previous concerns. I am happy with the publication of the manuscript.

Reviewer #2

(Remarks to the Author)

The authors have addressed my comments. Very few comments remain:

1. Line 358: ...led to increased SYK.....
2. Line 383: FLNA reacts with several partners, not just CEACAM1. One such example is calcium-sensing receptor (Casr).

This begs the question whether FLNA may also contribute to BCR-signaling via Casr and Ca²⁺ (see PMID: 37802982). This should also be mentioned.

Reviewer #3

(Remarks to the Author)

The authors have shown that CEACAM1 enhances the recruitment of LYN and SYK within the BCR complex, contributing to the early stages of BCR signaling by activating LYN and SYK, through binding to the adaptor protein filamin A (FLNA) which anchors them to the F-actin cytoskeleton and stabilizes lipid rafts in MCL. Since BCR-signaling has been associated with the pathogenesis and clinical outcome of MCL, these results are relevant to understand how this pathway drives MCL cell growth.

The authors provided a strong response to the comments and criticisms of the reviewers. In general, the justification of the study and the integrated approach, now using additional mice models and several MCL cell lines, make this study very relevant to the field and, therefore, is interesting for Nature Comms readers.

Reviewer #4

(Remarks to the Author)

552864_0_art_file_9797974_sltpvh

The authors wish to demonstrate a crucial role of CEACAM1 in the modulation of B cell receptor signaling especially in mantle cell lymphoma cells.

They provide convincing evidence for an upregulation of CEACAM1 in lymphoma cells. Furthermore, they demonstrate that some signaling mediators are differentially affected by IgM stimulation in the context of an altered CEACAM1 expression. Nevertheless, the study suffers from many inconsistencies with regard to the obtained data. Additionally, conclusions are drawn that are not justified by the methods used.

Specific concerns and comments:

Fig.4c: How can it be that a 2 min IgM incubation upregulates the expression of Filamin A and Lyn in gNTC cells?

Additionally, this finding contrasts with Fig.4b. There, abundance of Filamin A and Lyn is unaltered by IgM treatment of gNTC cells. Furthermore, according to Fig.4b there is no difference in the abundance of Filamin A and Lyn between gNTC and gCC1 cells. In Fig.4c (images) this looks different. There, both proteins are less present in gCC1 cells.

Fig.4d: How can it be that a 5 min IgM incubation doubles the expression of CEACAM1 in CC1 KO+4L cells (image and graph)? The endogenous CEACAM1 gene cannot contribute to the upregulation anymore, because it is knocked-out and the transfected CEACAM1 4L has its own promoter that should not be influenced by IgM stimulation. What happened? Why is this ultra-rapid upregulation not present in the WT cells? Similarly, why is the lipid raft marker GM1 upregulated after 5 min of IgM stimulation, but now in CC1 KO cells (clearly seen in the image)? The authors claim that after IgM stimulation abundance of GM1 is somewhat higher in WT cell compared to CC1 KO cells. This can't be seen in the image provided. Nevertheless, a little less marker protein does not necessarily indicate a reduced stability of the lipid rafts in CC1 KO cells. In line with that, the lipid raft marker Flotillin-1 is not reduced in the gCC1 cells compared to gNTC cells in Fig.4b. Since the finding concerning GM1 is the only argument that is provided by the authors to indicate a stabilizing effect of CEACAM1 on lipid rafts, this conclusion is more than questionable.

The authors claim that SYK is the kinase that is involved in CEACAM1-mediated modulation of BCR signaling. In Fig.3, 5 and 7 they therefore analyzed SYK and its phosphorylation. Why did they analyze SRC-Y416 instead of SYK-Y352 in Fig.4b and Fig.4d?

Sufficient explanation.

Fig.6a: CEACAM1 phosphorylation is unaltered by the deletion of the N domain and although the same amount of CEACAM1 was precipitated in the 4L and ΔN experiment, no phosphorylated CEACAM1 can be found in the precipitate of the ΔN experiment. How can this be the case?

Furthermore, in the input fraction, 4L and ΔN CEACAM1 show different molecular size, presumable because of the N domain deletion. But if so, why are the protein bands for phospho-4L and phospho-ΔN of the same size?

Sufficient explanation.

Fig.6b,c: According to proximity ligation assay, IgM stimulation does not induce increased interaction of CEACAM1 with SHP1 in 4L-transfected cells. This looks completely different in Fig.6a – there, much more SHP1 is detected in the CEACAM1 precipitate of 4L cells (at 5 min incubation roughly 10 times higher). Which finding is right?

Insufficient explanation. I doubt that the different results obtained with IP and PLA are a matter of sensitivity.

Furthermore, the authors suggest a lower sensitivity of IP compared to PLA in the revised manuscript, but their claim of a slower association of SHP with CEACAM1 (as compared to the Syk-CEACAM1 interaction) relies on the IP data (Fig. 5c,d). The PLA data (Fig. 5 a,b) show significant SHP-CEACAM1 interaction as soon as 5 min after IgM stimulation, that is the same as for Syk.

In Fig.6a and Fig.7c IgG control for IP are missing.

Sufficient explanation.

Fig.7a: What is "HA"? No explanation anywhere.

Corrected.

Fig.7b: Here, it is explicitly stated that analyses were conducted on two separate membranes. This implies that all other Western blots shown, were conducted on one membrane. This is impossible because of the overlapping molecular weights

of the detected proteins.

Sufficient explanation.

One important question: is the expression of the BCR itself affected by altered CEACAM1 expression. This needs to be clarified.

Sufficient explanation.

Except for Fig.5c, 5d, 7a and 7c, no Western blot is quantified! For Fig.7a and 7c statistical analysis based on TWO independent experiments (as stated in the figure legend)! How can you apply a statistical analysis on these data? ???

Overall, the number of animals/experiments should be clearly stated for each figure panel.

In the proximity ligation assay experiments, the number of analyzed cells covers a very wide range: Fig.5b 200-500 cells, Fig.6c 100-300 cells, Fig.7d ~200 cells. What is the rationale for this? Why are the numbers of analyzed cells different at all? Such a procedure could skew the results. Additionally, in how many independent experiments the analyzed cells were generated? 200 cells do not mean 200 independent experiments!

Sufficient explanation.

Regarding Fig.5c and Fig.7c: it is simply impossible to directly compare the signals of two different proteins obtained by Western blot! Beside the amount of protein, signal strength is ultimately affected by the titer, the affinity and avidity of the antibody used. These parameters are undoubtedly different concerning the used antibodies to detect SYK and SHPs. As a consequence, the lower signal of SHPs (compared to SYK) can be generated by a larger amount of SHP (compared to SYK), simply because the antibody performs less. Therefore, I think that a preference of CEACAM1 for SYK or SHPs depending on its expression level can not be deduced from these data.

Sufficient explanation.

In summary, regarding the inconsistencies/ weaknesses of the study that compromise two central findings of the study (lipid raft stabilization by CEACAM1, altered signaling molecule interaction depending on CEACAM1 expression level) I recommend rejection of the study.

Reviewer #5

(Remarks to the Author)

The authors have addressed my comments appropriately and provided satisfactory responses.

Version 2:

Reviewer comments:

Reviewer #4

(Remarks to the Author)

The authors addressed my major concerns, no further comments.

Responses to Reviewers' Comments:

Reviewer #1 (Remarks to the Author): expert in BCR, cytoskeleton

The paper identifies CEACAM1 as a novel amplifier of BCR signalling with a specific role in MCL. Since BCR signalling has been associated with the pathogenesis and clinical outcome of these non-Hodgkin lymphomas, but in a distinct role compared to other B cell cancers, the results are novel and interesting. They have implications for understanding how the BCR signalling drives malignant cell growth specifically in this disease. They could also be important for the responses of normal B cells and contribute to the discussion about the complexity of mechanisms (positive and negative) through which CEACAM proteins regulate kinase signalling pathways. The manuscript is generally well presented and contains abundant data supporting most of the conclusions.

Strong points of the paper:

MCL primary cells and cell lines typically have higher CEACAM1 expression compared to normal B cells or other types of non-Hodgkin lymphoma.

Knocking out or down CEACAM1 reduces MCL cell line growth or survival.

Knocking out or down CEACAM1 reduces BCR signalling in MCL lines at the earliest steps of BCR-proximal kinase activation.

CEACAM1 recruits Syk and filamin A and this requires intracellular tyrosines and the N-terminal ligand binding domain.

In non-MCL cell lines, CEACAM1 recruits SHP1 and inhibits BCR signalling.

Overall the data make a strong case for the importance of CEACAM1 overexpression in MCL augmenting BCR-dependent pathogenesis.

Response: We appreciate the Reviewer's positive feedback on the importance of CEACAM1's role in enhancing pathologic BCR signaling. While our results have strong translational implication in lymphoma, particularly the MCL model use in this work, we believe they could also inspire further research into the role of CEACAM1 in BCR signaling more broadly, including in normal healthy B cells.

However, there are a few major and a number of minor points that should be addressed before publication.

1. The conclusions that CEACAM1 "stabilizes membrane microdomains (lipid rafts) by anchoring to the F-actin cytoskeleton through adaptor protein filamin A" or that it "orchestrates lipid raft dynamics" are indirect. There is currently little evidence in the paper that the expression of CEACAM1 influences the composition of lipid rafts or recruitment of the BCR into lipid rafts in a way that would regulate BCR signalling. From Figure 4B, it seems that a small proportion of CEACAM1 (not quantified) is constitutively in raft fractions of the gradient. While this proportion may help recruit FLNA, the significance of this for BCR signalling is unclear as we do not know if FLNA is essential for BCR signalling in these cells. There are no statistical comparisons of the differences in the protein presence in the different fractions over several experiments. There is also no information on the partitioning of the BCR under these conditions. It seems that the results can be interpreted that CEACAM1 simply enhances Lyn and Syk activation, irrespective of its partial presence in the raft fractions. Fig 4C and D are even more difficult to interpret. The images are nicely quantified and substantiate the effect of CEACAM1 absence on Src phosphorylation and actin polymerisation, but they show an increase in total staining of Lyn, FLNA or GM1 after BCR crosslinking, which is difficult to interpret in terms of raft stability or dynamics. It is a bit puzzling why the intensity of these proteins/markers increases just a few minutes after BCR triggering. Most previous studies showed only changes in distribution/colocalisation. Thus, unless substantial new evidence is presented on the role of the lipid rafts and filamin A in the mechanism, I suggest toning down the above conclusions in the title and the abstract.

Response: We thank the Reviewer for pointing out the indirect evidence of CEACAM1 influence on the composition of lipid rafts or on BCR recruitment into lipid rafts. Indeed, the current manuscript does not specifically examine these processes as downstream effects of CEACAM1. Instead, we investigated CEACAM1's role in relation to BCR signaling by linking its involvement to BCR-associated processes, including lipid raft colocalization, activation of the lipid raft-resident LYN kinase, and F-actin reorganization

through its known interaction with filamin A. Accordingly, we agree with the Reviewer on moderating our conclusions in the title and abstract as follows.

Revised title: **“CEACAM1 as a Mediator of B-cell Receptor Signaling in Mantle Cell Lymphoma”**

Revised sentences in **Abstract**: *“Our signal transduction studies revealed that CEACAM1 plays a critical role in BCR activation through involvement in two dynamic processes. First, following BCR engagement, CEACAM1 co-localizes to the membrane microdomains (lipid rafts) by anchoring to the F-actin cytoskeleton through the adaptor protein filamin A.”*

Regarding the increased staining signals of LYN, FLNA, or GM1 after BCR crosslinking, while we do not have a definitive explanation, we speculate that this enhancement may be attributed to the specific cellular context used in these experiments.

As for statistical comparisons of the protein presence in the different fractions, we have now included the quantification and comparisons of signals in the lipid raft fraction (fraction I) between IgM-stimulated and unstimulated samples and between stimulated control and CEACAM1 knockout samples in **Fig. 4b**, as shown below. The source data of the statistical comparisons are included in **Source Data Fig 4** in **Supplementary Files**.

Fig. 4: CEACAM1 co-localizes to lipid rafts during BCR activation. **a** Diagram of sucrose-density gradient fractionation to isolate lipid rafts. **b** *Left panels*, Immunoblot analysis for fractions indicated in **(a)** from control or CEACAM1 knockout JEKO-1 cells stimulated with control or anti-IgM antibody (1 μg/ml) for 2 min. *Right panels*, immunoblot signal quantification of fraction I after normalization to Flotillin-1 (loading control) and unstimulated samples. Shown are normalized means of fold changes from three independent fractionation experiments. ** P < 0.01, * P < 0.05 by a two-sided, paired t test. ns, not significant.

2. The prioritisation of CEACAM1 for these studies is logical and interesting. However, the paper does not present the actual results of the CRISPR screen beyond the dot plot in Fig 1A. A supplementary table with full-screen results is a standard in the field and would be an important part of the paper. Similarly important is the list of genes implicated in patient survival in MCL aggregated from previous studies. Since the cited papers containing the original data do not highlight CEACAM1 themselves, it would be important to provide a reanalysis of those data to illustrate how CEACAM1 expression correlates with progression.

Response: We thank the Reviewer for the helpful suggestion. We have provided the entire dataset of the CRISPR screen results in the **Supplementary Data 1** tab and the gene list associated with poor prognosis in MCL in the **Supplementary Data 2** tab, both in **Supplementary Data 1 to 5** in **Supplementary Files**.

Fig 1 d,f and in other figures throughout the paper, the loss of CEACAM-1 targeted cells is called “loss of viability”. Is this actually cell death as implied, or is it loss of proliferation and thus outgrowth of the GFP- cells?. Perhaps a more generic term is “cell growth” or “proliferation and survival”.

Response: **Fig. 1d,f** directly measured the reduction in viable cell numbers, as did in **Fig. 3h**, which showed a decrease in viable cells under drug treatment. In **Fig. 1d,f**, the decline in viable cell numbers was indeed due

to increase apoptosis and reduced cell cycle progression. We have provided additional data measuring apoptosis and cell cycle progression after CEACAM1 depletion in **Extended Data Fig. 2**, as shown below. Per the Reviewer's suggestion, we also further clarified in the text on **page 7** as follows:

“Our findings confirmed that depletion of CEACAM1 impaired cell viability by enhancing apoptosis and reducing S-phase progression (Fig. 1c, d; Extended Data Fig. 2).”

Extended Data Fig. 2: Effects of CEACAM1 knockdown on apoptosis and cell cycle. **A**, Immunoblots show CEACAM1 depletion by shRNA transduction in JEKO-1 cells compared to control in four independent experiments. **B**, Representative flow cytometry plots of JEKO-1 cells transduced with control or CEACAM1 shRNA and stained for Annexin V and propidium iodide (PI). *Right panel*, quantification of % apoptotic cells (Annexin V+PI+) from four independent experiments. **C**, Representative cell cycle analysis of JEKO-1 cells transduced with control or CEACAM1 shRNA. *Right panel*, quantification of each cell cycle phase for the indicated times post-transduction from four independent experiments.

Extended data Fig 3a - please indicate what is the green part of the sequence schemes.

Response: This figure is now **Extended Data Fig. 5a**. We have clarified this in the figure legend as follows:

“The green area indicates the distinct amino acid sequence resulted from alternative splicing in the cytoplasmic tail of the short isoforms (variants 2 and 5).”

Fig 3A,B. Statistics and numbers of experiments are missing. Why was PLCG1 chosen? Most BCR studies focus on PLCG2, which is more expressed in B cells.

Response: Two-way ANOVA statistics has been included in **Fig. 3a** and the corresponding figure legend. All data in Fig. 3a,b are representative from at least three independent experiments, which has been described in the figure legend.

We also examined but did not see any effect of CEACAM1 knockdown on PLCG2 phosphorylation. Therefore, we only showed the effect on PLCG1.

Fig 3H - please use empty histograms, it is currently difficult to see the data for the individual cell lines.

Response: The figure has been fixed using empty histograms.

Fig. 3: h CEACAM1 expression is correlated with ibrutinib response.

Fig3J, presumably the WB is done with the E1 antibody, which is binding to the intracellular part? Please indicate this in the figure/legend.

Response: Thank you for catching this missing label. Yes, we used the E1 antibody and have provided this information both in the figures and figure legends.

Extended data Fig 5B - statistical analysis of the signalling data is needed.

Response: Two-way ANOVA statistics has been included in **Extended Data Fig. 7B** (previously Extended Data Fig. 5B) and in the corresponding figure legend, as shown below.

Extended Data Fig. 7: FLNA is required for BCR signaling and survival in MCL. **A**, JEKO-1 cells transduced with control or FLNA shRNA and analyzed by immunoblotting with indicated antibodies. **B**, Ca^{2+} flux signals in indicated control or FLNA-knockdown JEKO-1 cells following anti-IgM stimulation (1 μ g/mL). **** $P < 0.0001$ by two-way ANOVA. **C**, Effects of FLNA knockdown on cell survival. JEKO-1 cells were transduced with control or FLNA shRNA co-expressing GFP. Viable, GFP+ cells were monitored over time by flow cytometry. Shown are the means of GFP+ fractions compared to day-2 samples from at least two independent experiments. Error bars, SD. * $p < 0.05$ by two-way ANOVA.

Figure 5A: The data analysis does not show the statistical significance of increased CEACAM1 interactions with SYK after crosslinking, just higher values compared to SHP1. Fig 7D suggest that the SHP1 PLA is working in a different cell line, but the interpretation of Fig5A would require a positive verification that the SHP1 signal is meaningful compared to some control.

Response: We thank the Reviewer for pointing out the missing comparison between increased CEACAM1 interactions with SYK after crosslinking. We do acknowledge that comparing the interactions between SYK and SHP-1 with CEACAM1 would be problematic due to the inherent differences in the SYK and SHP-1 antibodies. Therefore, in **Fig. 5a,b**, we now perform the statistical comparisons between CEACAM1-SYK or CEACAM1-SHP1 interactions with their respective unstimulated control, as shown below.

Fig. 5: CEACAM1 interactions with BCR signaling components. a, b Top panels, Proximity ligation assay (PLA) showing interactions (visualized as red dots using Airyscan FAST 2D confocal microscope and a 40x/1.2NA water objective) between CEACAM1 (CC1) and the indicated proteins in JEKO-1 and MINO cells stimulated with 2 μ g/ml of anti-IgM antibody for the indicated times. **Bottom panels,** Quantification of PLA signals shown in the top panels for approximately 200 cells on average from three independent experiments using QuPath 0.3.2 software. **** P<0.0001, *** P<0.001 by a two-tailed unpaired t test. ns, not significant.

Figure 5C, D. The IP shows that there is an increase in p-SYK associated with CEACAM1 after anti-IgM. However, there seems to be a substantial amount of non-phosphorylated Syk bound to CEACAM1 before stimulation. This correlates with basal CEACAM phosphorylation in Fig 6. Perhaps this is worth highlighting. However, the statistical comparisons focus on comparing the pSYK to SHP1 data, which is not a good comparison as they are developed using different antibodies. A better comparison is to compare to the unstimulated control. It is also very unusual to use a nested t-test for the analysis. Two-way ANOVA pairing the control and stimulation in each experiment is more typical in these settings. While the slow kinetics of SHP1 are interesting, they are not reproduced in Fig 6A.

Response: We agree with the Reviewer that SYK may play a role in basal CEACAM1 phosphorylation before IgM stimulation. Likewise, LYN, which is also found associated with CEACAM1 prior to stimulation, could similarly induce this phosphorylation. We have highlighted this observation in the section describing the result of this experiment in **Fig. 6a** on **page 15** as follows:

“Notably, the full-length 4L CEACAM1 pull-down from unstimulated samples exhibited basal phosphorylation, potentially resulting from basal interactions with LYN and/or SYK, which were also found to interact with CEACAM1 prior to stimulation (Fig. 6a).”

The Reviewer was correct in pointing out the problem of statistical comparisons between SYK and SHP-1 due to the use of two different antibodies. We have revised the comparisons of each protein interaction to their

corresponding unstimulated controls and used the two-sided t-test to perform pair-wise comparisons of each time point to the unstimulated control, as shown below.

Fig. 5: CEACAM1 interactions with BCR signaling components. **c, d** Immunoprecipitation analysis of CEACAM1 interactions. *Top panels*, JEKO-1 or MINO cells were stimulated with 2 mg/ml of anti-IgM antibody for the indicated times and CEACAM1 was immunoprecipitated with a CEACAM1-specific antibody or IgG control antibodies followed by immunoblotting with indicated antibodies. One percent of total lysates was used as input control. *Bottom panels*, Quantification of indicated co-IP signals shown in the top panels. Bar graphs show the means and individual densitometric values from three independent experiments for each timepoint normalized to the CEACAM1 pull-down signals. Error bars, SD. *** $P < 0.001$, ** $P < 0.01$, * $P < 0.05$ by a two-sided unpaired t test. ns, not significant.

While we observed slower SHP-1 kinetics in both JEKO-1 and MINO cells, as shown in **Fig. 5c,d**, the small difference in signal intensity at later time points was not statistically significant, as indicated in the revised **Fig. 5c,d**. This may explain the inconsistency in SHP-1 kinetics noted by the Reviewer in **Fig. 6a**. We acknowledge that the co-immunoprecipitation and immunoblotting approach may lack the sensitivity needed to detect changes in SHP-1 recruitment kinetics. Despite the inconclusive findings on SHP-1 kinetics, our conclusion regarding the differences in CEACAM1 recruitment of SYK and SHP-1 in different cellular contexts remains unaffected. Accordingly, we have revised our conclusions to reflect these limitations in interpreting the data in **Fig. 5c,d** and **Fig. 6a** on **page 14 and 15** as follows:

“However, while the early SYK kinetics was statistically significant, the slower SHP-1 kinetics did not reach statistical significance, possibly due to immunoprecipitation and immunoblotting approach, which may lack the sensitivity needed to detect changes in SHP-1 recruitment kinetics.”

Figure 6A - There is no statistical analysis of the IP results.

Response: We have included statistical analysis of the IP results from at least three independent experiments, as shown below. The source data of the statistical comparisons are included in **Source Data Fig 6** in **Supplementary Files**.

Fig. 6: CEACAM1 interactions with BCR signaling components require an intact N-domain. *Left panel*, Immunoprecipitation analysis of CEACAM1 interactions. *Right panels*, Quantification of indicated co-IP signals shown in the left panels. Bar graphs show the means and individual densitometric values from four independent experiments for each timepoint normalized to the CEACAM1 pull-down signals. Error bars, SD. ** $P < 0.01$, * $P < 0.05$ by two-way ANOVA.

Extended data 7 and 6A - there are multiple bands in the phospho-CEACAM1 blot in lysate but only one in the IP. What is the antibody used here?

Response: The multiple bands in the lysate were the result of the anti-phosphotyrosine (p-Y) clone 4G10 antibody since there is currently no reliable anti-phospho-CEACAM1 antibody. We have clarified this in these figure legends.

Figure 7 - The results with the N-terminal deletion are very interesting. What is the expression level of this construct on the cell surface compared to 4L? Can the authors speculate on whether they think that cis or trans interactions are involved? Is there any role of the CEACAM1 in the homotypic adhesion of the cells?

Response: We thank the Reviewer for raising this important question. We observed that the expression levels of the N-terminal deleted CEACAM1 were consistently higher than that of the 4L wildtype version as evaluated by western blots. Detection of these differences on the cell surface is currently challenging because the flow antibody only recognizes the N-terminal domain. We addressed the significance of the N domain on **page 19 in the Discussion section**, aligning our findings with a previous study that showed impaired ITIM tyrosine phosphorylation upon mutating two residues critical for homophilic interaction of this domain. Depending on specific cell types, we observed that overexpression of CEACAM1 sometimes induces cell clumping, suggesting trans interactions at play. Determining which cell contexts and the significance of CEACAM1 interactions with itself and others like the BCR is actively pursued in our laboratory.

Figure 7 - The preferential recruitment of SHP1 in CEACAM1-low cells is also interesting. Not explained, but it opens new ideas. However, the comparisons on the WB seem selective. Again, possibly ANOVA with appropriate pairing would be less biased.

Response: **Fig. 7** highlighted the differences in CEACAM1 recruitment preference and outcomes in lymphoma cells with high or low CEACAM1 expression levels, aiming to reconcile previous findings on CEACAM1's negative role due to the ITIM motifs on its cytoplasmic tail. We appreciate the Reviewer's comment on this important finding. We addressed these findings on **pages 19 and 20 of the Discussion** and proposed a potential explanation for CEACAM1's dual role. Specifically, the lower expression levels of SYK in CEACAM1-low cells may allow SHP-1 to competitively bind with CEACAM1 under our experimental conditions, where CEACAM1 is ectopically expressed in these cells.

We thank the Reviewer for suggesting the suitable statistical test. We have used two-way ANOVA, as recommended, in both **Fig. 7a,c**, and revised the comparison between stimulated vs. unstimulated groups within the same antibody.

The cartoon is helpful, but also potentially misleading. I don't think there is any evidence that Syk or SHP1 bind using both their SH2 domains to the pairs of phospho-tyrosines. There is also a possibility that Syk binds to the phosphotyrosine in a hemITAM fashion, i.e. to a CEACAM1 dimer (10.1126/scisignal.aan3676).

Response: We thank the Reviewer for helping to improve the accuracy of the illustration. We have revised the diagram as shown here.

Fig. 8: Proposed model for CEACAM1 function in BCR signaling.

Reviewer #2 (Remarks to the Author) expert in BCR, lipid rafts

In this manuscript by Xavier et al. the authors study the role of CEACAM1, a transmembrane protein and adhesion molecule, for signaling downstream of the B cell receptor in mantle cell lymphoma. The authors show that CEACAM1 stabilizes lipid rafts via anchoring to the cytoskeleton via filamin A. CEACAM1 also recruits SYK to the BCR complex. In careful *in vitro* work the authors have, therefore, unravelled novel functions of CEACAM1 in particular with regards to its signaling and its activating role for signaling downstream of the BCR.

The manuscript is well written and the experiments are carefully performed. However, any evidence that this pathway is playing a role *in vivo* is lacking. Is there an inhibitor of CEACAM1? What are the effects on signaling if an inhibitor of filamin A, if available, is employed?

Response: We appreciate the Reviewer's positive feedback. We have generated additional *in vivo* data as shown in our response to your comment #3 below.

To the best of our knowledge, the only anti-CEACAM1 antibody CM24 developed by cCAM Biotherapeutics had advanced to phase 1 clinical trial aiming to eliminate CEACAM1-positive tumors. However, the lack of a positive result from this trial implicates further investigations are required for the development of effective inhibitors of CEACAM1 (Gotz et al., *Front Immunol.* 2023, doi: [10.3389/fimmu.2023.1295232](https://doi.org/10.3389/fimmu.2023.1295232)).

We are not aware of any pharmacological inhibitor of filamin A (FLNA). In the revised **Extended Data Fig. 7**, we showed that knockdown of FLNA in JEKO-1 cells resulted in down-regulation of BCR signaling (assayed by IgM-stimulated Ca²⁺ signaling) and decrease in cell survival. These results are consistent with the role of FLNA in facilitating BCR signaling through its interactions with CEACAM1 and F-actin cytoskeleton. We anticipate that a future chemical inhibitor of FLNA may similarly suppress BCR signaling and cell survival.

Other points are:

1. Is there a relevance of CEACAM1 as a tumor marker, similar to CEA?

Response: Yes, CEACAM1 has been identified as a potential biomarker in several cancer entities such as lung cancer and gastric cancer (reviewed by Gotz et al, 2024 *Eur J Clin Invest*, doi: [10.1111/eci.14337](https://doi.org/10.1111/eci.14337)).

2. Figure 1f: Does CEACAM1 play a role in any chromosomal translocation events as frequently found in lymphomas?

Response: We are not aware of CEACAM1 playing a role in chromosomal translocation in lymphomas, at least from our latest search on PubMed and Google Scholar using key words CEACAM1 and chromosomal translocation.

3. Please include *in vivo* experiments to prove the relevance for lymphoma progression.

Response: In addition to our *in vivo* xenograft model of MCL using JEKO-1 cells shown in **Fig. 1j-i**, we have employed a transgenic mouse model of MCL developed in Dr. Samir Parekh's laboratory to investigate the role of CEACAM1 in lymphoma development. In this transgenic mouse model, SOX11 is expressed in B cells under the immunoglobulin enhancer. The transgenic mouse developed MCL-like population (CD19+CD5+CD23-) by two months of age with 100% penetrance (Kuo *et al. Blood* 2018, PMID: 29615403). The E μ -SOX11 mice were crossed with E μ -CCND1 to achieve concurrent overexpression of SOX11 and CCND1, hallmark features of MCL biology. The double E μ -SOX11/CCND1 transgenic (DT) mice display enhanced MCL-like phenotype compared to E μ -SOX11 mice (Edwards *et al. Blood* 2020, 136 Supplement 1:6-7, <https://doi.org/10.1182/blood-2020-143038>). We further crossed DT mice with CEACAM1^{-/-} mice to examine the impact of CEACAM1 deficiency on lymphoma progression. We observed that the MCL-like population in DT mice with a CEACAM1^{-/-} background was significantly reduced in male mice (n=5) compared to DT mice (n=6). A similar trend was observed in female mice, though with lower statistical significance. These findings indicate that CEACAM1 plays a crucial role in MCL development in the DT mouse model. We have included these additional data in **Fig. 1j (see below)** and in the revised text on **page 9**.

Fig. 1j: CEACAM1 is required for lymphoma progression in MCL mouse model. *Top panel*, Diagram depicts generation of double SOX11/CCND1 transgenic (DT) mice in the Ceacam1-deficient background. *Middle panel*, Representative flow cytometry plots show the gate of MCL-like cell population (CD19+CD5+CD23-). *Bottom panel*, box plots show the percentage of CD19+CD5+CD23- MCL-like cells isolated from peripheral blood in both male and female mice from indicated genotypes. P values are from unpaired t test with Welch's correction.

4. Do CEACAM1-deficient lymphoma cells proliferate less? Cell cycle? Apoptosis? Are these signaling studies relevant for studying proliferation?

Response: Yes, we showed that depletion of CEACAM1 resulted in downregulation of BCR signaling and cell survival. We have provided additional data showing the impact of CEACAM1 knockdown on cell cycle and apoptosis in Extended Data Fig. 2, as shown below:

Extended Data Fig. 2: Effects of CEACAM1 knockdown on apoptosis and cell cycle. **A**, Immunoblots show CEACAM1 depletion by shRNA transduction in JEKO-1 cells compared to control in four independent experiments. **B**, Representative flow cytometry plots of JEKO-1 cells transduced with control or CEACAM1 shRNA and stained for Annexin V and propidium iodide (PI). *Right panel*, quantification of % apoptotic cells (Annexin V+PI+) from four independent experiments. **C**, Representative cell cycle analysis of JEKO-1 cells transduced with control or CEACAM1 shRNA. *Right panel*, quantification of each cell cycle phase at the indicated times post-transduction from four independent experiments.

5. Is a CEACAM1 knockout murine model available for further studies and what is this mouse's phenotype?

Response: Yes, CEACAM1 knockout mice are available for the current study. CEACAM1 deficient mice exhibit reduced B cell numbers together with impaired antibody production (Khairnar et al 2015, PMID: 25692415) and defective CD8+ T cell responses to chronic viral infection (Khairnar et al 2018, PMID: 29967450). We showed in **Fig. 3f** that Ceacam1^{-/-} splenic B cells are defective in BCR-mediated Ca²⁺ signaling. In the revised manuscript, we have provided additional data using Ceacam1^{-/-} mice in response to comment #3 above.

6. Is there a correlation of CEACAM1 with CD5 expression? I.e. does CEACAM1 also play a role in CLL cells?

Response: Yes, we believe that CEACAM1 may also have a role in CLL. However, due to the lack of suitable cell line models for CLL (all currently available CLL cell line models have been immortalized with EBV thus limiting their usage), we only focus on MCL in the current manuscript.

7. Figure 2f: This is a comparison of very different entities, in which CEACAM1 may play very different roles. It would be helpful to add CLL here.

Response: We appreciate the Reviewer's suggestion and have added the staining for CLL in **Fig. 2f** (see below). Like MCL, we identified positive CEACAM1 staining in a very high frequency (~94%) of CLL samples (n=16), as shown in **Table 1** below.

Fig. 2f: Representative immunohistochemistry images showing varying CEACAM1 staining levels for indicated formalin-fixed paraffin-embedded tissue microarrays.

Table 1: Summary of CEACAM1 IHC staining of tissue microarrays

Disease	Positive	Weak	Negative	Total positive	% Positive
Mantle cell lymphoma	16	4	3	20	86.96%
CLL/SLL	8	7	1	15	93.75%
DLBCL	10	11	61	21	25.61%
GCB	8	7	35	15	30.00%
ABC	0	2	13	2	13.33%
Indeterminate	2	0	15	2	11.76%
Follicular lymphoma	11	15	57	26	31.33%
grade 1-2	4	7	35	11	23.91%

grade 3A	5	5	11	10	47.62%
grade 3B	1	1	1	2	66.67%
DLBCL+FL	0	2	7	2	22.22%
Indeterminate	1	0	3	1	25.00%
Classic Hodgkin lymphoma	2	1	36	3	7.69%

Minor points:

1. It seems as if figure 7c is cited out of order.

Response: Thank you! We have revised the text on page 17 to reflect the correct order of Fig. 7 reference.

2. It may be helpful to add changes of the cytoskeleton to figure 8.

Response: Since the current study does not focus on the phenotype of the cytoskeleton in the two main cellular contexts, our illustration only depicts its functional interaction with filamin A in the context of BCR signaling. However, we are open to the Reviewer's specific features of cytoskeletal changes to improve the diagram in Fig. 8.

3. It is suggested to organize the results section in such a way that one subsection with a given result corresponds to one figure. Currently, the figure number changes within a subsection.

Response: We thank the Reviewer for this helpful comment. We have carefully ensured that most figures align with the description of their respective subsections. However, due to space constraints, we had to distribute the data across Fig. 5 and Fig. 6 for a single subsection to enhance clarity and improve data visualization. In addition, we have occasionally cited results from previous figures in new subsections to help readers quickly reference the data.

Reviewer #3 (Remarks to the Author): expert in Mantle Cell Lymphoma

Overall Review: In this manuscript, Xavier & coauthors, using a comprehensive approach combining genome-wide CRISPR loss-of-function screens, gene expression profiling and BCR-signal transduction studies, have identified CEACAM1 as a critical factor for mantle cell lymphoma (MCL) proliferation and survival, playing a pivotal role in B-cell receptor (BCR) signaling. Their findings demonstrate that CEACAM1 activates BCR signaling by binding the adaptor protein filamin A (FLNA) which anchors them to the F-actin cytoskeleton and stabilizes lipid rafts. This process enhances the recruitment of LYN and SYK within the BCR complex, ultimately leading to robust BCR activation. The N-terminal as well as the C-terminal domains of CEACAM1 are essential for its interaction with FLNA, CD79B, LYN and SYK, promoting downstream phosphorylation, positively contributing to the early stages of BCR signaling by activating LYN and SYK.

The strengths of this manuscript include the genome-wide CRISPR loss-of-function screens, incorporation of publicly available RNA-seq and clinical data and well designed experiments. Weakness include limited data in primary MCL samples, particularly related to CEACAM1 mechanism of action enhancing the BCR-signaling, with mechanistic data restricted to a single MCL cell line with no (functional) validation in the context of the tumor microenvironment's influence or in vivo therapeutic implications. Below are suggestions to improve the publication:

MAJOR COMMENTS:

1. CEACAM1-dependent MCL viability was validated using CEACAM1-shRNA across various lymphoid cell lines. However, CEACAM1 silencing appears to impact viability only in lymphoid cell lines with high CEACAM1 expression, all of which are MCL-derived. Notably, these results have not been compared between MCL cell lines with low CEACAM1 expression (e.g., Z138) and those with high expression (e.g., REC1) (Fig. 2d).

Response: We appreciate the Reviewer for highlighting the missing MCL cell lines (REC-1 and Z-138) in the survival analysis of CEACAM1 knockdown. The omission was due to initial technical challenges in transducing these cells. We have since resolved the issue and included the results of CEACAM1 knockdown in REC-1 and

Z-138 cells in **Fig. 1f**, as shown below. As expected, cell survival dependence on CEACAM1 is significantly higher in CEACAM1^{hi} REC-1. The lower dependency of Z-138 cells on CEACAM1 is consistent with the results observed in MAVER-1 cells. Both Z-138 and MAVER-1 cells exhibit resistance to ibrutinib, indicating reduced reliance on BCR signaling due to somatic mutations in TRAF2 and TRAF3, respectively, which lead to active non-canonical NF-κB signaling.

Fig. 1f: Effects of CEACAM1 knockdown on cell survival. Indicated cells were transduced with CEACAM1 shRNA shown and viable, propidium iodide (PI)-negative cells were assessed by flow cytometry over time. Shown are the means of PI negative fractions compared to day-2 samples from at least two independent experiments. Error bars, SD. * P<0.05, ** P<0.01 by a one-sided, paired t test.

2. Tumor engraftment and *in vivo* viability in MCL xenograft mouse models were significantly reduced upon CEACAM1-silencing. While these findings are intriguing, the experiments were conducted exclusively with the JEKO-1 cell line. Given the important heterogeneity observed among MCL cases, it is essential to include additional cell lines to validate the *in vivo* effects of CEACAM1 overexpression on MCL progression and viability.

Response: The Reviewer has raised a valid point concerning the sole JEKO-1 cell line being used *in vivo*. Unlike JEKO-1 cells, genetic manipulation in other CEACAM1^{high} MCL cell lines such as MINO and REC-1 is technically more challenging due to low lentiviral transduction efficiency (MINO) or decreased cell viability after lentiviral transduction (REC-1). While these limitations still allow short-term, *in vitro* experiments on these cell lines, acquiring a large number of CEACAM1-silenced cells sufficient for *in vivo* mouse experiments become technically challenging and cost-prohibitive, precluding us from performing them.

Despite these challenges, we have investigated the role of CEACAM1 in a transgenic mouse model of MCL developed by Dr. Samir Parekh. The E_μ-SOX11 transgenic mice developed MCL-like population (CD19+CD5+CD23-) by two months of age with 100% penetrance (Kuo *et al. Blood* 2018, PMID: 29615403). To achieve concurrent overexpression of SOX11 and CCND1, hallmark features of MCL biology, E_μ-SOX11 mice were crossed with E_μ-CCND1 mice. The resulting double E_μ-SOX11/CCND1 transgenic (DT) mice exhibited an enhanced MCL-like phenotype compared to E_μ-SOX11 mice (Edwards *et al. Blood* 2020, 136 Supplement 1:6-7, <https://doi.org/10.1182/blood-2020-143038>). By further crossing DT mice with CEACAM1^{-/-} mice, we observed that the MCL-like population was significantly reduced in male DT mice with a CEACAM1^{-/-} background (n=5) compared to DT controls (n=6). A similar trend was observed in female mice, though with lower statistical significance. These findings suggest that CEACAM1 plays a crucial role in MCL development in the DT mouse model. We have included these additional data in **Fig. 1j (as shown)** and in the revised text on **page 9**.

Fig. 1j: CEACAM1 is required for lymphoma progression in MCL mouse model. *Top panel*, Diagram depicts generation of double SOX11/CCND1 transgenic (DT) mice in the Ceacam1-deficient background. *Middle panel*, Representative flow cytometry plots show the gate of MCL-like cell population (CD19+CD5+CD23-). *Bottom panel*, box plots show the percentage of CD19+CD5+CD23- MCL-like cells isolated from peripheral blood in both male and female mice from indicated genotypes. P values are from unpaired t test with Welch's correction.

3. CEACAM1 is highly expressed in the majority of MCL tumors compared to other lymphoid neoplasms and normal B cells. Notably, CEACAM1 has been shown to be more highly expressed in SOX11+ MCL than in SOX11- MCL (Balsas P, et al., Blood, 2021). It would be valuable to investigate the correlation between SOX11, which is exclusively expressed in MCL, and CEACAM1 expression across different MCL cell lines and primary samples. Additionally, examining the impact of ectopic SOX11 overexpression on CEACAM1 expression in JVM2 cell lines could provide further insights. Finally, analyzing survival dependency in CEACAM1+ versus CEACAM1- JVM2 cell lines would help clarify the functional relationship between them.

Response: We thank the Reviewer for sharing an important paper that provides the evidence of CEACAM1 expression in MCL, particularly in the SOX11+ subtype. Indeed, the question whether SOX11 regulates CEACAM1 was also raised by us during our study. However, we did not think this is the case based on the following observations: 1) the MCL cell line Z-138 expresses high levels of SOX11 but very low levels of CEACAM1 and 2) many SOX11-negative B cell malignancies such as B-CLL express CEACAM1 at high levels. Nonetheless, we appreciate the Reviewer's comments on the potential insights we may learn from these analyses and performed the experiments as suggested.

We first investigated the correlation between SOX11 and CEACAM1 expression across MCL primary samples by analyzing the mRNA expression of these genes from the public dataset GSE16455 that has 15 SOX11+ conventional and 7 SOX11- indolent MCL samples (Fernandez et al., Cancer Res 2010, PMID: 20124476). As shown in **Fig. 1 below**, the average CEACAM1 mRNA expression in SOX11+ MCL is higher than those in SOX11- MCL (**Fig. 1A, middle panel**). However, there is only a weak correlation between SOX11 and CEACAM1 ($R^2=0.1317$, **Fig. 1A, bottom panel**), although this could also be due to the limited number of samples from the dataset.

To functionally determine whether SOX11 affects CEACAM1 expression, we over-expressed SOX11 in the SOX11-negative MCL cell line JVM-2 and analyzed surface CEACAM1 expression by flow cytometry. As shown in **Fig. 1B**, the empty vector (EV) or SOX11 lentiviral vector that co-expresses GFP was verified for SOX11 expression by transfection into HEK-293T cells followed by immunoblotting. JVM-2 cells were subsequently transduced with these vectors and the infected cell populations (GFP+) were gated and compared for their CEACAM1 expression on day 2 and day 4 after lentiviral transduction. We did not see any changes in CEACAM1 expression between EV-infected and SOX11-infected JVM-2 cells on both days of analysis. This result indicates that, in the context of JVM-2 cells, SOX11 does not affect CEACAM1 expression levels.

Regarding the survival dependency in CEACAM1+ versus CEACAM1- JVM2 cell lines, we do not think that this cell line is a suitable system to address the role of CEACAM1 in BCR signaling and survival of MCL for the following reasons: 1) JVM-2 cells express low levels of CEACAM1 and IgM, and these cells do not mount calcium response upon IgM stimulation, suggesting downregulation of the BCR signaling machinery (unpublished observations); and 2) JVM-2 cells were developed by EBV-mediated immortalization which could confound the analysis of CEACAM1 manipulation on cell survival.

Fig. 1: Correlation between SOX11 and CEACAM1 expression. **A**, SOX11 and CEACAM1 mRNA expression from primary MCL samples were compared and correlated. **B**, Diagrams show control and SOX11 expression vectors used to transfect 293T cells to validate SOX11 expression as shown in the Western blot. JVM-2 cells were subsequently transduced with these vectors and the infected cell populations (GFP+) were gated and compared for CEACAM1 expression on day 4 after lentiviral transduction by flow cytometry.

4. BCR-mediated Ca²⁺ signaling was significantly reduced upon CEACAM1 depletion in MAVER-1 cells. However, cell survival and sensitivity to Ibrutinib remained comparable between CEACAM1-high MAVER-1 and the CEACAM1-low Z138 cell lines. To establish whether Ibrutinib sensitivity is influenced by CEACAM1 expression, I recommend: (1) Conduct similar experiments using CEACAM1-silenced JEKO-1 cell lines and (2) comparing CEACAM1-low and CEACAM1-high MCL cells derived from primary samples. (3) Perform analogous studies in Ibrutinib-resistant cell lines, such as Z138 or JVM2, to explore potential differences in sensitivity and resistance mechanisms.

Response: We greatly appreciate the Reviewer's keen observations regarding ibrutinib resistance of MAVER-1 and Z-138 cells, which exhibit opposing CEACAM1 expression levels. While the suggested experiments would be valuable in addressing CEACAM1's role in ibrutinib sensitivity, this aspect is beyond the scope of our study. Here, we used ibrutinib response as an indicator of ongoing BCR signaling to correlate with CEACAM1 expression. However, we found that CEACAM1 expression alone does not strongly correlate with BCR signaling, as demonstrated by the resistance of CEACAM1^{high} MAVER-1 cells. This resistance can be attributed to somatic mutations in MAVER-1 cells, which activate the non-canonical NF- κ B pathway, making cell survival independent of BCR signaling or CEACAM1 expression. This issue was further discussed in the last paragraph of the Discussion section in the original manuscript version.

Nonetheless, we respond to the Reviewer's suggested experiments as follows:

- (1) Conduct similar experiments using CEACAM1-silenced JEKO-1 cell lines: We have performed this experiment and observed enhanced sensitivity of CEACAM1-silenced JEKO-1 to ibrutinib, as shown below in Fig. 2.

Fig. 2: CEACAM1 depletion enhances ibrutinib sensitivity. JEKO-1 cells were transduced with control or CEACAM1 shRNA and treated with indicated concentrations of ibrutinib for 24 hours followed by analysis of viable cells. Line graphs show means of three independent experiments. Error bars, SD. * $P < 0.05$ from a paired two-sided t-test.

- (2) Comparing CEACAM1-low and CEACAM1-high MCL cells derived from primary samples: We also conducted ibrutinib sensitivity experiments for primary MCL samples with a wide range of CEACAM1 expression levels (data not shown). However, we did not observe any correlation between CEACAM1 expression and ibrutinib response. This outcome was expected due to the reasons stated above, i.e., genetic heterogeneity, including somatic mutations that activate alternative pathways, making the lymphoma cells independent of CEACAM1 or BCR signaling.
- (3) Perform analogous studies in ibrutinib-resistant cell lines, such as Z138 or JVM2, to explore potential differences in sensitivity and resistance mechanisms: As mentioned earlier, this study does not focus on drug resistance mechanisms.

5. Results shown in Fig. 3 and 4 should be confirmed in more than one MCL cell line and comparing CEACAM low/high MCL cells from primary samples.

Response: Unlike JEKO-1 cells, genetically manipulating other CEACAM1^{high} MCL cell lines, such as MINO (highly resistant to lentiviral transduction) and REC-1 (low transduction efficiency), is technically challenging, preventing us from performing similar experiments as in JEKO-1 cells. Despite these limitations, we have made every effort to incorporate multiple cell lines when possible. For example, we used MAVER-1, K1718, and mouse splenic B cells to examine the effects of CEACAM1 depletion on BCR signaling, as shown in Fig. 3d-f.

Regarding experiments on primary MCL cells, we agree with the Reviewer about their importance. However, obtaining definitive conclusions on CEACAM1's role requires genetic manipulation of CEACAM1 levels in primary cells, which remains technically challenging. Simply comparing CEACAM1-high and CEACAM1-low levels in primary samples is insufficient for drawing conclusions about ibrutinib sensitivity owing to the underlying heterogeneity of these samples, as highlighted in MAVER-1 and Z-138 cells above.

6. CEACAM1 also can regulate negatively the BCR signaling in lymphoma cells with low or absent CEACAM1 expression by preferentially binding to SHP-1/2 instead of SYK. However recruitment of SHP-1 is high in MCL cell lines expressing high levels of CEACAM1 (Fig 5a). To clarify this hypothesis I recommend: (1) conduct similar experiments and analyze the levels of SHP-1 bound to CEACAM1 in CEACAM1-silenced cell lines and (2) comparing CEACAM1-low and CEACAM1-high MCL cells derived from primary samples.

Response: We thank the Reviewer for the helpful suggestions. Indeed, in Fig. 7c, we performed the IP experiment as suggested, using CEACAM1-knockout JEKO-1 cells as a control for CEACAM1 pull-down in CEACAM1-knockout JEKO-1+4L (wildtype CEACAM1) cells. As expected, CEACAM1 depletion abolished SHP-1/2 interaction signals. In JEKO-1 cells, where CEACAM1 acts as a positive regulator, it interacts with both SYK and SHP-1, yet still promotes positive BCR signaling outcomes. In contrast, when CEACAM1 was introduced into CEACAM1^{low} Z-138 cells, it acts as a negative regulator of BCR signaling.

To further investigate this observation, we compared CEACAM1 interactions with SYK and SHP-1/2 between JEKO-1 and Z-138 cells. Our data indicate that CEACAM1-SYK interactions are significantly more abundant in JEKO-1 than in Z-138 cells. Conversely, CEACAM1-SHP-1/2 interactions are more prevalent in Z-138 cells than in JEKO-1 cells. We speculate that the low SYK expression levels in Z-138 cells may allow more competitive binding of SHP-1/2 to CEACAM1. We are currently testing this hypothesis in a follow-up study.

Regarding the second suggested experiment, we acknowledge that results from primary samples would provide valuable insights. However, conducting IP experiments using primary samples is technically challenging due to limited availability of material.

7. CEACAM1 has recently been identified as an immune checkpoint in various tumors. To further validate its functionality in MCL, BCR-signaling studies in 3D coculture systems could provide insights into how the tumor microenvironment influences CEACAM1 activity and its potential role as an immune checkpoint in MCL. Additionally, comparing CEACAM1 expression in primary samples from peripheral blood and lymph nodes could help determine whether its expression is crucial for interactions between MCL cells and the tumor microenvironment or how its expression is influenced by the microenvironment.

Response: We thank the Reviewer for these insightful recommendations. The idea of CEACAM1 interaction with the tumor microenvironment is important owing to its homophilic interaction with neighboring cells. While we are enthusiastic about further studies along this direction, we believe that this investigation deserves its own focus in future study and is out of scope of the current study.

8. CEACAM1 inhibitory experiments with already described blocking antibodies in in vitro and in vivo MCL models would demonstrate its therapeutic implications in MCL.

Response: We agree with the Reviewer on the important implication of inhibiting CEACAM1 therapeutically. Again, these studies need extensive validation of the blocking antibodies, which is beyond the scope of the current manuscript. We will follow up with these excellent suggestions in future studies.

MINOR COMMENTS:

1. It is difficult to follow the text because the data is not there or it is not displayed under the correct name. e.g. I couldn't find Extended Data 3a, b, c. In extended Data 5, I only see 5a. Fig. 5a&c, showing significantly reduced both Ca²⁺ signals and cell survival in JEKO-1 cell are not shown in the corresponding Extended data 5. I couldn't find Extended Data Fig. 6.

Response: We suspect that the missing figures may be due to technical issues with the software, as we have not received similar complaints from other Reviewers. Based on my experience, some Word or PDF files may experience delays in displaying the figures or may not display them at all ***unless users click on the blank image areas***. We apologize for this inconvenience and will repost the mentioned figures here. Please note the revised figure labels: previous Extended Data Fig. 3 is now **Extended Data Fig. 5**; previous Extended Data Fig. 5 is now **Extended Data Fig. 7**; previous Extended Data Fig. 6 is now **Extended Data Fig. 8**.

Extended Data Fig. 5 (previously Extended Data Fig. 3): CEACAM1 expression in lymphoma cells by qPCR analysis. **A**, Diagrams of CEACAM1 coding variants and corresponding NCBI gene accession numbers show varying amino acid (aa) lengths and locations of Taqman probes. TM, transmembrane. ITIM, immunoreceptor tyrosine based inhibitory motif. The green area indicates the distinct amino acid sequence resulted from alternative splicing in the cytoplasmic tail of the short isoforms (variants 2 and 5). **B**, Validation of Taqman probes. Expression analysis by qPCR using Taqman probes specific for each CEACAM molecules was performed on mRNA harvested from the human lung cancer cell line A549 and HEK-293T cells as positive and negative control, respectively. Shown are the means of mRNA expression levels after normalization to GAPDH signals from 4 independent amplification experiments. Error bars, SD. **C,D**, qPCR analysis of mRNA expression levels for indicated CEACAM molecules in indicated cell lines, MCL PDX or primary MCL samples using specific Taqman probes validated in (**B**). Shown are the means of mRNA expression levels after normalization to GAPDH signals from 4 independent amplification experiments. Error bars, SD.

Extended Data Fig. 7 (previously Extended Data Fig. 5): FLNA is required for BCR signaling and survival in MCL. **A**, JEKO-1 cells transduced with control or FLNA shRNA and analyzed by immunoblotting with indicated antibodies. **B**, Ca^{2+} flux signals in indicated control or FLNA-knockdown JEKO-1 cells following anti-IgM stimulation ($1\mu g/mL$). **** $P < 0.0001$ by two-way ANOVA. **C**, Effects of FLNA knockdown on cell survival. JEKO-1 cells were transduced with control or FLNA shRNA co-expressing GFP. Viable, GFP+ cells were monitored over time by flow cytometry. Shown are the means of GFP+ fractions compared to day-2 samples from at least two independent experiments. Error bars, SD. * $p < 0.05$ by two-way ANOVA.

Extended Data Fig. 8 (previously Extended Data Fig. 6): CEACAM1 is required for optimal assembly of lipid rafts. Control (gNTC) or CEACAM1 knockout (gCEACAM1) JEKO-1 cells were stimulated with 2 μ g/ml anti-IgM antibody for 2 min. Shown are representative confocal immunofluorescence images (acquired with a 20x/0.8NA objective) of control and IgM-stimulated cells co-stained with anti-FLNA (green) and anti-LYN (red) antibodies followed by nuclear staining with DAPI (blue). Scale bar, 10 μ m.

2. CEACAM1-4S it is not detectable in JEKO-1 cells+gCC1 cell lines (Fig 3 j).

Response: We apologize for not being clear in explaining our use of the antibody to detect the CEACAM1-4S isoform in the western blot. As shown in **Extended Data Fig. 6 below** (previously Extended Data Fig. 4), the CEACAM1 antibody clone E1 does not detect the 4S isoform, which can instead be detected by flow cytometry using the CEACAM1 antibody clone B1.1. Therefore, we used the accompanying flow cytometry plot in **Fig. 3i** with clone B1.1 to confirm 4S expression. We have now updated **Fig. 3j** to include the E1 label for the CEACAM1 antibody and have modified the figure legend accordingly.

Figure 3

Fig. 3i,j: Controls or reconstituted cell lines were verified for CEACAM1 expression by flow cytometry using B1.1 antibody (i) and stimulated with 1 ug/mL of anti-IgM F(ab')₂ fragments for 5 min followed by immunoblot analysis with indicated antibodies including the CEACAM1 E1 antibody (j).

Extended Data Fig. 6

Reviewer #4 (Remarks to the Author): expert in CEACAM1

In this manuscript by Xavier et al. entitled “CEACAM1 Orchestrates Lipid Raft Dynamics and B-cell Receptor Signaling in Mantle Cell Lymphoma” the authors report about a crucial role of CEACAM1 in the modulation of B cell receptor signaling especially in mantle cell lymphoma cells.

They provide convincing evidence for an upregulation of CEACAM1 in lymphoma cells. Furthermore, they demonstrate that some signaling mediators are differentially affected by IgM stimulation in the context of an altered CEACAM1 expression.

Nevertheless, the study suffers from some inconsistencies with regard to the obtained data. Additionally, conclusions are drawn that are not entirely justified by the methods used.

Specific concerns and comments:

The authors claim that SYK is the kinase that is involved in CEACAM1-mediated modulation of BCR signaling. In Fig.3, 5 and 7 they therefore analyzed SYK and its phosphorylation. Why did they analyze SRC-Y416 instead of SYK-Y352 in Fig.4b and Fig.4d?

Response: We thank the Reviewer for this question. In **Fig. 4**, we demonstrated the association of CEACAM1 within a key BCR signaling environment, such as lipid rafts. To investigate this, we used LYN, a lipid raft marker, along with its activated form, phospho-LYN (commonly detected using SRC-pY416 antibody). SYK is located in the cytoplasm and not present in the lipid raft fraction.

Fig.6a: CEACAM1 phosphorylation is unaltered by the deletion of the N domain and although the same amount of CEACAM1 was precipitated in the 4L and ΔN experiment, no (or only less) phosphorylated CEACAM1 can be found in the precipitate of the ΔN experiment. How can this be the case?

Response: The reason there were multiple bands in the input samples, where phospho-CEACAM1 was expected, was the use of phosphotyrosine (pY) antibody clone 4G10. This antibody is useful for detecting

CEACAM1 tyrosine-phosphorylation only when CEACAM1 is immunoprecipitated (there is currently no reliable phospho-tyrosine CEACAM1 antibody). The pY 4G10 antibody can detect phospho-tyrosine residues and useful for detecting p-CEACAM1 in the input samples. We have clarified the use of this antibody in the figure legend.

Fig.6b,c: According to proximity ligation assay, IgM stimulation does not induce increased interaction of CEACAM1 with SHP1 in 4L-transfected cells. This looks completely different in Fig.6a – there, much more SHP1 is detected in the CEACAM1 precipitate of 4L cells (at 5 min incubation roughly 10 times higher). Which finding is right?

Response: Although both PLA and IP were used as complementary approaches, we acknowledge that they may not yield consistent results due to their difference in sensitivity. Despite this challenge, our interpretation of slower SHP-1 kinetics (compared to SYK) in CEACAM1^{high} cells, such as JEKO-1 and MINO cells, was deduced from several observations, as shown by PLA (Fig. 5a,b; at 30' in Fig. 6c) and IP (Fig. 5c,d). However, the small difference in the western blot signal intensity at later time points was not statistically significant, as indicated in the revised Fig. 5c,d. This may explain the inconsistency in SHP-1 kinetics noted by the Reviewer in Fig. 6a, although the high signal at 5' of the IP blot in this figure could also be caused by more CEACAM1 pull-down at this time point.

While the co-immunoprecipitation and immunoblotting approach may lack the sensitivity needed to detect significant changes in SHP-1 recruitment kinetics, our conclusion regarding the differences in CEACAM1 recruitment of SYK and SHP-1 in different cellular contexts remains unaffected. Accordingly, we have revised our conclusions to reflect these limitations in interpreting the data in **Fig. 5c, d** and **Fig. 6a** on **page 14 and 15** as follows:

“However, while the early SYK kinetics was statistically significant, the slower SHP-1 kinetics did not reach statistical significance, possibly due to immunoprecipitation and immunoblotting approach, which may lack the sensitivity needed to detect changes in SHP-1 recruitment kinetics.”

In Fig.6a and Fig.7c IgG controls for IP are missing.

Response: In IP experiments, IgG controls are used to rule out that the co-precipitated signals are not due to contamination from the whole cell lysate. However, the best control is internal controls using the same IP antibody. In our case, the lack of or reduced co-IP signals in the delta-N IP samples in **Fig. 6a** serve as internal controls for 4L IP samples. Similarly, in **Fig. 7c**, internal controls are JEKO-1+gCC1 and Z-138+Con (empty vector). In **Fig. 5c,d**, since there was no internal control samples, IgG controls were used.

Fig.7a: What is “HA”? No explanation anywhere.

Response: We apologize for leaving an explanation out. We have included this in the figure legend of this figure: *“HA, hemagglutinin, a protein tag in-framed with CEACAM1 to detect the 4S isoform”*.

Fig.7b: Here, it is explicitly stated that analyses were conducted on two separate membranes. This implies that all other Western blots shown, were conducted on one membrane. This is impossible because of the overlapping molecular weights of the detected proteins.

Response: In **Fig. 7b**, two separate membranes were used due to the missing of the SYK figure in membrane 1 during our manuscript preparation, prompting us to re-run these samples for SYK and essentially related signals on a separate membrane. The missing SYK figure from membrane 1 was later found and now included. The Reviewer raised a legitimate concern about probing several proteins with similar molecular weights on the same membrane. However, this practice has been made possible due to the commercially available isotype-specific secondary antibodies. With due diligence through careful validation of antibodies used in our experiments, we are highly confident about the specificity of proteins detected on these blots.

One important question: is the expression of the BCR itself affected by altered CEACAM1 expression. This needs to be clarified.

Response: We thank the Reviewer for raising this important point. When altering CEACAM1 expression either by depletion or overexpression, we checked for potential effects on the expression of the BCR. For example, concerning about the off-target effect of CRISPR-mediated knockout of CEACAM1 on CD79A, a component of the BCR and a neighboring gene located at the 19q13.2 locus near CEACAM1, we showed that there were no changes in CD79A expression in CEACAM1 sgRNA-transduced JEKO-1 cells by both flow cytometry and immunoblotting (**Extended Data Fig. 3a,b**). We also did not see any effects on IgM expression in JEKO-1 cells after CEACAM1 knockout by sgRNA. This result has been included in **Extended Data Fig. 3c**, as shown here.

Extended Data Fig. 3c: CEACAM1 knockout does not affect IgM expression. Contour flow cytometry plot comparing IgM expression between control gRNA (gNTC)- or CEACAM1 gRNA (gCEACAM1)-transduced JEKO-1 cells.

Regarding the effect of CEACAM1 overexpression on IgM expression, we showed in **Extended Data Fig. 11** that overexpression of CEACAM1 in both Z-138 and ST486 cells did not change the IgM surface expression levels, as shown here.

Extended Data Fig. 11: Effects of CEACAM1 expression on IgM surface levels. Z-138 and ST486 cells were transduced with empty vector control or CEACAM1-4L (4L) and analyzed for CEACAM1 and IgM expression by flow cytometry.

Except for Fig.5c, 5d, 7a and 7c, no Western blot is quantified! For Fig.7a and 7c statistical analysis based on TWO independent experiments (as stated in the figure legend)! How can you apply a statistical analysis on these data?

Response: Signal quantifications have now been included for those blots with missing values.

For the statistical analysis in **Fig. 7a,c**, we used two-way ANOVA to compare two categorical independent variables (that is, two cell lines and time variables). Although our sample sizes are small, with only two replicates that may or may not be statistically significant, the time variables from the two cell lines under

comparison are independent, making the comparison between the cell lines statistically significant. We include a few examples of two-way ANOVA tests below. A complete list of all the tests for these figures can be found in the corresponding **Source Data Fig 7** in the **Supplementary Files**.

Table Analyzed					Table Analyzed				
Fig 7A pSYK Y352					Fig 7C-JEKO-1, Z138 SYK				
Two-way RM ANOVA	Matching: Across row				Two-way RM ANOVA	Matching: Across row			
Assume sphericity?	Yes				Assume sphericity?	Yes			
Alpha	0.05				Alpha	0.05			
Source of Variation	% of total variation	P value	P value sum	Significant?	Source of Variation	% of total variation	P value	P value summary	Significant?
Replicates x EV vs 4L	1.047	0.4852	ns	No	Replicates x Jeko-1 vs Z-138	0.5044	0.421	ns	No
Replicates	33.12	0.0917	ns	No	Replicates	1.63	0.3218	ns	No
EV vs 4L	55.9	0.025	*	Yes	Jeko-1 vs Z-138	85.42	<0.0001	****	Yes
Induction time	7.023	0.2926	ns	No	Induction time	8.389	0.1992	ns	No
Table Analyzed					Table Analyzed				
Fig 7A pSYK Y352					Fig 7C-JEKO-1, Z138 SHP-1				
Two-way RM ANOVA	Matching: Across row				Two-way RM ANOVA	Matching: Across row			
Assume sphericity?	Yes				Assume sphericity?	Yes			
Alpha	0.05				Alpha	0.05			
Source of Variation	% of total variation	P value	P value sum	Significant?	Source of Variation	% of total variation	P value	P value summary	Significant?
Replicates x EV vs 4S	10.46	0.014	*	Yes	Replicates x Jeko-1 vs Z-138	32.75	0.009	**	Yes
Replicates	33.94	0.2926	ns	No	Replicates	21.54	0.005	**	Yes
EV vs 4S	21.41	0.0069	**	Yes	Jeko-1 vs Z-138	25.13	0.0159	*	Yes
Induction time	33.88	0.0088	**	Yes	Induction time	6.943	0.7841	ns	No

Overall, the number of animals/experiments should be clearly stated for each figure panel.

In the proximity ligation assay experiments, the number of analyzed cells covers a very wide range: Fig.5b 200-500 cells, Fig.6c 100-300 cells, Fig.7d ~200 cells. What is the rationale for this? Why are the numbers of analyzed cells different at all? Such a procedure could skew the results. Additionally, in how many independent experiments the analyzed cells were generated? 200 cells do not mean 200 independent experiments!

Response: Thank you for the helpful suggestions. The number of animals and experiments have been included in the figures previously missing these values.

In PLA assays, we analyzed an approximate 200 cells in most cases. Some cell loss may occur during the wash steps, leading to aggregate counting of cell numbers on these slides and introducing variation. However, since these experiments were repeated at least three times, we are confident that the results are representative.

Regarding Fig.5c and Fig.7c: it is simply impossible to directly compare the signals of two different proteins obtained by Western blot! Beside the amount of protein, signal strength is ultimately affected by the titer, the affinity and avidity of the antibody used. These parameters are undoubtedly different concerning the used antibodies to detect SYK and SHPs. As a consequence, the lower signal of SHPs (compared to SYK) can be generated by a larger amount of SHP (compared to SYK), simply because the antibody performs less. Therefore, I think that a preference of CEACAM1 for SYK or SHPs depending on its expression level can not be deduced from these data.

Response: We agree with the Reviewer's concern about comparing between two different antibodies and have revised these figures. In **Fig. 5c**, we now compare stimulated and unstimulated controls separately for each antibody. Similarly, in **Fig. 7c**, for each antibody, we compare between JEKO-1 and Z-138 cells. We believe these revised comparisons are now more appropriate for interpreting our results.

For example, in **Fig. 5c**, there is a significant increase in p-SYK and SYK interactions with CEACAM1 at early time points, whereas SHP-1 recruitment occurs at later time points, though it does not reach statistical significance. In **Fig. 7c**, CEACAM1 predominantly associates with SYK in JEKO-1 cells, whereas in Z-138 cells, it interacts more with SHP-1 and SHP-2.

Reviewer #5 (Remarks to the Author): expert in CRISPR screens, lymphomas

BCR signaling is critical in MCL pathogenesis, though its mechanisms are not fully understood. In this study, Xavier et al identified CEACAM1 as essential in a subset of MCL tumors through a genome-wide CRISPR library screen. Mechanically, CEACAM1 stabilizes membrane microdomains via F-actin anchoring and enhances SYK recruitment to activate BCR signaling. These functions depend on its cytoplasmic ITIM domain and N-terminal ectodomain. Interestingly, CEACAM1, known as an inhibitory receptor, exhibits an activating role in the BCR signaling in MCL cells, providing new insights into BCR-targeting therapies in MCL.

The findings of this study are novel and significantly advance our understanding of MCL pathobiology. However, further refinements are needed to enhance the impact and value of this manuscript.

Major comments:

1) The authors should present a comprehensive overview of the quality of the genome-wide CRISPR screening results. Specifically, the Human GeCKO v2 library contains 1,000 control sgRNAs designed not to target any site in the genome. Data related to these control sgRNAs should be summarized and displayed in a figure for quality assessment. Additionally, the screening results for pan-dependent genes (1,682 genes) should also be presented in a figure. A good example for reference is Extended Data Fig. 1 in PMID: 29925955. Furthermore, the entire dataset of genome-wide CRISPR screening results, including Log₂ fold change (Log₂FC), p-values, and rankings, should be provided as a supplementary table for transparency and reproducibility.

2) One of the significant advantages of genome-wide CRISPR screening is the ability to understand the global gene essentiality in signaling pathways. Considering the critical role of aberrant BCR signaling in MCL cell biology, it would be important to report the CRISPR screening results for genes involved in BCR signaling, particularly IgM, CD79B, CD79A, CD19, LYN, and SYK. This analysis would provide insights into the essentiality of these genes in MCL cells.

3) BTK serves as a pivotal hub connecting BCR signaling to the NFκB pathway. In this context, the authors should detail the CRISPR screening results for key genes in the NFκB signaling pathway, including BTK, CARD11, BCL10, MALT1, IKKB, IKBK, CHUK, NFKB1, REL, IRF4, and SPIB. Reporting the essentiality of these genes would elucidate the dependency of MCL cells on NFκB signaling and highlight potential therapeutic targets.

Response for 1) 2) and 3): We thank the Reviewer for the insightful comments and suggestions. We have provided the entire dataset of the CRISPR screen results in **Supplementary Data 1**. The results of control sgRNAs and pan-dependent genes can be found in **Extended Data Fig. 1A** and genes in the BCR and NFκB signaling pathways in **Extended Data Fig. 1B**, as shown below, together with reference to source data as **Supplementary Data 3, 4, and 5** in the figure legend.

Extended Data Fig. 1: Results of controls in JEKO-1 CRISPR library screen. Box plots show Log₂ fold-change (LFC) of negative control gRNAs (Supplementary Data 3) and pan-dependent genes (Supplementary Data 4) (A) and genes in the BCR and NFκB signaling pathways (Supplementary Data 5) (B).

4)

Figure 2D is particularly intriguing as it shows that MCL cells exclusively express CEACAM1. However, the molecular mechanism underlying this higher expression of CEACAM1 in MCL cells remains unclear. Does SOX11 regulate CEACAM1 mRNA expression, potentially driving its overexpression? To provide a deeper understanding, it would be informative to include immunoblot data for SOX11, SYK, and LYN in Figure 2D.

Response: We thank the Reviewer for pointing to the potential mechanism of SOX11 regulating CEACAM1 expression. Please refer to our more detailed response to a similar question from comment #3 of Reviewer 3. Briefly, we observed only a weak correlation between SOX11 and CEACAM1 expression ($R^2=0.1317$, **Fig. 1A, bottom panel**) from a public dataset (GSE16455 including 15 SOX11+ conventional and 7 SOX11- indolent MCL samples). Furthermore, over-expressing SOX11 in the SOX11-negative MCL cell line JVM-2 did not affect CEACAM1 surface expression levels as shown in **Fig. 1B**. Additional study using a larger cohort will be required to conclusively address this question.

Per the Reviewer's suggestion, we ran additional immunoblots for primary MCL samples and performed correlation analysis for SOX11 expression with CEACAM1, SYK, and LYN. SOX11 expression appears to correlate well with CEACAM1 and LYN but weaker with SYK, as shown in **Fig. 3 below**. However, the mechanism of SOX11 regulating CEACAM1 expression is beyond the scope of our study, and we believe that these questions deserve more detailed investigation in future study.

Fig. 3: Correlation of SOX11 expression with CEACAM1, SYK, and LYN. Immunoblots showing primary MCL samples probed with indicated antibodies and the signal intensities of protein expression were quantified by densitometry. Scatter plots showing the correlation of SOX11 protein expression levels with indicated proteins.

5)

In the discussion section, the authors speculated that "high SYK expression may enable SYK to outcompete SHP-1 in binding to CEACAM1," thereby influencing CEACAM1's role in BCR activation. To substantiate this hypothesis, the authors should create SYK-overexpressing Z-138 cells and compare them with mock-transduced cells. Specifically, the levels of p-SYK and p-CD79A following BCR stimulation should be evaluated

and presented in Figure 7A. This experiment would provide direct evidence for the proposed mechanism and clarify SYK's role in CEACAM1-mediated signaling.

Response: We thank the Reviewer for the very helpful suggestions. We are actively testing this hypothesis in an upcoming manuscript that focuses on the same experiments recommended by the Reviewer. However, we believe that a separate study is necessary to fully explore the nuances and details of these experiments, as they fall beyond the scope of the current manuscript.

6)

To investigate the function of CEACAM1 in greater detail and to evaluate whether CEACAM1 and LYN share similar functions, the authors should perform RNA-seq on JEKO1 and MINO cells transduced with sgCEACAM1, sgLYN, or the control sgNTC.

Response: We thank the Reviewer for highlighting the need to understand CEACAM1's function in greater detail and its potential shared function with LYN. Despite these two molecules are structurally distinct and carry different roles in BCR signaling, both exhibit dual regulatory roles, acting positively or negatively depending on the cellular context.

The Reviewer's proposed experiments to compare the role of CEACAM1 and LYN through gene expression will serve as a valuable starting point for generating new hypotheses and conducting in-depth studies on the detailed functions of both CEACAM1 and LYN.

In our study, we also asked whether CEACAM1 and LYN are functionally related because our data showed that LYN phosphorylation was defective after CEACAM1 depletion (**Fig. 3b in the main text**). To investigate this, we first examined whether CEACAM1 directly affects LYN kinase activity. To do this, we immunoprecipitated LYN from the lysates of control and CEACAM1-depleted JEKO-1 cells and performed an *in vitro* LYN kinase assay. In this assay, ATP was added to LYN pull-down to induce autophosphorylation and the products were analyzed by immunoblots using anti-phospho-tyrosine antibody (e.g., 4G10 clone). We did not find a substantial difference in LYN phosphorylation levels between the control and CEACAM1-depleted cells (as shown in **Fig. 4 below**). This result indicates that specific LYN kinase activity is not directly dependent on CEACAM1, but it could be indirectly influenced by CEACAM1 interactions with FLNA and the F-actin network leading to stabilization of lipid rafts and the BCR signaling microenvironment.

These preliminary findings suggest that much remains unknown about the comparative functions of CEACAM1 and LYN, and that these studies, including those proposed by the Reviewer, deserve further investigation, and should be addressed through future dedicated research.

Fig. 4: LYN in vitro kinase assay. Control or CEACAM1 shRNA-transduced JEKO-1 cells were immunoprecipitated with anti-LYN antibody and LYN pull-downs were treated with ATP followed by immunoblot analysis with indicated antibodies. Numbers below bands are densitometric values normalized to LYN signals.

Minor comments:

7)

The meaning of "4L" in "WT CEACAM1 (4L)" is unclear. The authors should provide a detailed explanation of this term in the manuscript, ensuring clarity for the readers.

Response: Thank you for the suggestion. A detailed description has been provided on page 4 of the Introduction, as shown below:

“CEACAM1 is structurally composed of four immunoglobulin (Ig)-like extracellular domains—an N-terminal IgV-like domain, followed by three IgC2-like domains—along with a single-pass transmembrane region and a cytoplasmic domain that exists in two isoforms: the long isoform (CEACAM1-L), which contains two immunoreceptor tyrosine-based inhibitory motifs (ITIMs), and the short isoform (CEACAM1-S), which lacks ITIMs. The longest isoform, CEACAM1-4L, includes all four Ig-like extracellular domains and a long cytoplasmic tail.”

8)

In Page 8 line 8, Fig.2d should be Fig. 2f.

Response: We believe we cited the correct **Fig. 2d**, which refers to the Western blot showing low or undetectable levels of CEACAM1 in the mentioned lymphoma cell lines. **Fig. 2f** shows representative histology of CEACAM1 staining in primary tumor microarrays. However, this confusion could be due to our citing Fig. 2d out-of-order. We have corrected this by using “see below”.

Response to Reviewers' comments

Reviewer #1 (Remarks to the Author):

The authors addressed my previous concerns. I am happy with the publication of the manuscript.

Response: We thank you for time and support.

Reviewer #2 (Remarks to the Author):

The authors have addressed my comments. Very few comments remain:

Response: The authors would like to thank you for your constructive and helpful comments to improve the manuscript.

1. Line 358: ...led to increased SYK.....

Response: Typo corrected.

2. Line 383: FLNA reacts with several partners, not just CEACAM1. One such example is calcium-sensing receptor (Casr). This begs the question whether FLNA may also contribute to BCR-signaling via Casr and Ca²⁺ (see PMID: 37802982). This should also be mentioned.

Response: We appreciate the insightful advice and helpful information regarding FLNA. In Extended Data Fig. 7, we showed reduced BCR-mediated Ca²⁺ signaling and cell survival following knockdown of FLNA in JEKO-1 cells to provide a functional link between FLNA, BCR activation, and Ca²⁺ response. We agree that mentioning this new piece of puzzle should stimulate future inquiry and have added the following sentence on **line 385**:

“Notably, FLNA was recently found to interact with the calcium-sensing receptor CaSR⁵¹, suggesting its potential link to BCR-mediated Ca²⁺ signaling.”

Reviewer #3 (Remarks to the Author):

The authors have shown that CEACAM1 enhances the recruitment of LYN and SYK within the BCR complex, contributing to the early stages of BCR signaling by activating LYN and SYK, through binding to the adaptor protein filamin A (FLNA) which anchors them to the F-actin cytoskeleton and stabilizes lipid rafts in MCL. Since BCR-signaling has been associated with the pathogenesis and clinical outcome of MCL, these results are relevant to understand how this pathway drives MCL cell growth.

The authors provided a strong response to the comments and criticisms of the reviewers. In general, the justification of the study and the integrated approach, now using additional mice models and several MCL cell lines, make this study very relevant to the field and, therefore, is interesting for Nature Comms readers.

Response: Your supportive comments are greatly appreciated.

Reviewer #5 (Remarks to the Author):

The authors have addressed my comments appropriately and provided satisfactory responses.

Response: We sincerely appreciate your time and efforts.

Reviewer #4 (Remarks to the Author):

The authors wish to demonstrate a crucial role of CEACAM1 in the modulation of B cell receptor signaling especially in mantle cell lymphoma cells.

They provide convincing evidence for an upregulation of CEACAM1 in lymphoma cells. Furthermore, they demonstrate that some signaling mediators are differentially affected by IgM stimulation in the context of an altered CEACAM1 expression.

Nevertheless, the study suffers from many inconsistencies with regard to the obtained data. Additionally, conclusions are drawn that are not justified by the methods used.

Specific concerns and comments:

Fig.4c: How can it be that a 2 min IgM incubation upregulates the expression of Filamin A and Lyn in gNTC cells? Additionally, this finding contrasts with Fig.4b. There, abundance of Filamin A and Lyn is unaltered by IgM treatment of gNTC cells.

Response: The new concern raised by the Reviewer likely stems from a misunderstanding between the interpretation of fluorescence and immunoblotting data—two distinct techniques with different purposes. In **Fig. 4c**, we do not believe that the expression levels of FLNA and LYN increased following IgM stimulation. Rather, the enhanced fluorescent signals observed after anti-IgM incubation are attributed to the relocalization of FLNA and LYN to lipid rafts during IgM activation, as detected by immunofluorescence, which is a preferred technique to visualize subcellular protein localization. Since CEACAM1 is known to interact with FLNA—which binds cytoskeletal F-actin, a key player in B cell activation—FLNA was also examined. The enhanced immunofluorescent signal for LYN post IgM stimulation is consistent with prior findings, including those shown in Fig. 2C from the study by Chew and Lam (*JBC* 2007), as shown below.

Fig. 2C, recruitment of LPXN to the plasma membrane upon BCR activation. BJAB cells were stimulated with 10 ug/ml anti-human IgM F(ab) 2 fragment for various time points, cytopspin onto glass slides, and stained with anti-LPXN (red) and anti-Lyn (green) antibodies. (Chew and Lam, *J. Biol. Chem.* **282**:37, 27181-27191 (2007); PMID: 17640867)

The Reviewer correctly noted that no changes in the abundance of FLNA and LYN were observed upon IgM treatment in the immunoblot data (**Fig. 4b**), aligning with our own observations throughout the study (Figs. 3b, 3j; Fig. 4b; and Fig. 6a). However, it is important to emphasize that this does not contradict the immunofluorescence results, as these methods serve complementary but distinct purposes: immunoblotting is optimal for evaluating total protein expression levels, while immunofluorescence reveals spatial distribution of proteins within cells. An increase in fluorescence intensity may reflect the relocalization of proteins to specific subcellular compartments, rather than an actual change in their total cellular abundance.

Furthermore, according to Fig.4b there is no difference in the abundance of Filamin A and Lyn between gNTC and gCC1 cells. In Fig.4c (images) this looks different. There, both proteins are less present in gCC1 cells.

Response: The Reviewer was correct in pointing out that there is no difference in the abundance of FLNA and LYN between gNTC and gCC1 cells in the immunoblots shown in **Fig. 4b**. This aligns with our observation that CEACAM1 knockout does not affect the expression levels of these proteins, a finding consistently seen throughout our study. In **Fig. 4c**, however, the immunofluorescence signals of FLNA and LYN appear reduced in gNTC cells compared to gCC1 cells, likely due to our initial selection of non-representative images from the source data set (Extended Data Fig. 8, shown here). Upon quantifying multiple fields from the same source, as presented in the right panels of **Fig. 4c**, we confirm that there is no significant difference in mean signal intensity of FLNA and LYN between gNTC and gCC1 cells. We acknowledge the confusion caused by the previous image selection and have replaced those panels with different single-cell images (as indicated in Extended Data Fig. 8 below) that more accurately reflect the quantified data shown in the revised **Fig. 4c** (shown below).

Extended Data Fig. 8: CEACAM1 is required for optimal assembly of lipid rafts. Control (gNTC) or CEACAM1 knockout (gCEACAM1) JEKO-1 cells were stimulated with 2 μ g/ml anti-IgM antibody for 2 min. Shown are representative confocal

immunofluorescence images (acquired with a 20x/0.8NA objective) of control and IgM-stimulated cells co-stained with anti-FLNA (green) and anti-LYN (red) antibodies followed by nuclear staining with DAPI (blue). Scale bar, 10 μ m.

Revised Fig. 4c: The four single-cell panels in gCC1 cells were replaced with new images as indicated in Extended Data Fig. 8 above.

Fig.4d: How can it be that a 5 min IgM incubation doubles the expression of CEACAM1 in CC1 KO+4L cells (image and graph)? The endogenous CEACAM1 gene cannot contribute to the upregulation anymore, because it is knocked-out and the transfected CEACAM1 4L has its own promoter that should not be influenced by IgM stimulation. What happened? Why is this ultra-rapid upregulation not present in the WT cells? Similarly, why is the lipid raft marker GM1 upregulated after 5 min of IgM stimulation, but now in CC1 KO cells (clearly seen in the image)? The authors claim that after IgM stimulation abundance of GM1 is somewhat higher in WT cell compared to CC1 KO cells. This can't be seen in the image provided. Nevertheless, a little less marker protein does not necessarily indicate a reduced stability of the lipid rafts in CC1 KO cells. In line with that, the lipid raft marker Flotillin-1 is not reduced in the gCC1 cells compared to gNTC cells in Fig.4b. Since the finding concerning GM1 is the only argument that is provided by the authors to indicate a stabilizing effect of CEACAM1 on lipid rafts, this conclusion is more than questionable.

Response: As previously discussed, we do not believe that IgM stimulation alters CEACAM1 expression. Instead, it leads to the re-localization of CEACAM1 to membrane microdomains, forming aggregates that become detectable by immunofluorescence. The greater signal intensities between CC1 KO JEKO-1+4L and WT cells likely reflects the nature of the ectopic CEACAM1-4L expression construct used.

A similar rationale applies to GM1. Although the mean GM1 signal intensity is modestly higher in WT cells compared to CC1 KO cells, this difference is statistically significant. These changes in GM1 immunofluorescence signals further support a role for CEACAM1 in lipid raft localization. Importantly, the immunoblot signals of flotillin-1 in Fig. 4b remain unchanged by CC1 KO, supporting its use as a loading control.

The Reviewer was correct in pointing out that "a little less marker protein does not necessarily indicate a reduced stability of the lipid rafts in CC1 KO cells". Indeed, the current study does not specifically address lipid raft stabilization as a downstream function of CEACAM1. Rather, our focus is on CEACAM1's role in BCR signaling, particularly its contributions to BCR-associated processes such as lipid raft colocalization, activation of the lipid raft-resident LYN kinase, and F-actin reorganization through its known interaction with FLNA. To reflect this refined focus, we have moderated our conclusions in both the title and abstract accordingly as follows.

Revised title: "**CEACAM1 as a Mediator of Orchestrates Lipid Raft Dynamics and B-cell Receptor Signaling in Mantle Cell Lymphoma**"

Revised sentences in **Abstract**: "Our signal transduction studies revealed that CEACAM1 plays a critical role in BCR activation by ~~orchestrating~~ through involvement in two dynamic processes. First, following BCR

engagement, CEACAM1 stabilizes co-localizes to the membrane microdomains (lipid rafts) by anchoring to the F-actin cytoskeleton through the adaptor protein filamin A.”

Fig.6b,c: According to proximity ligation assay, IgM stimulation does not induce increased interaction of CEACAM1 with SHP1 in 4L-transfected cells. This looks completely different in Fig.6a – there, much more SHP1 is detected in the CEACAM1 precipitate of 4L cells (at 5 min incubation roughly 10 times higher). Which finding is right?

Insufficient explanation. I doubt that the different results obtained with IP and PLA are a matter of sensitivity. Furthermore, the authors suggest a lower sensitivity of IP compared to PLA in the revised manuscript, but their claim of a slower association of SHP with CEACAM1 (as compared to the Syk-CEACAM1 interaction) relies on the IP data (Fig. 5c,d). The PLA data (Fig. 5 a,b) show significant SHP-CEACAM1 interaction as soon as 5 min after IgM stimulation, that is the same as for Syk.

Response: We appreciate the Reviewer's persistence in thoroughly investigating the underlying cause of the inconsistent CEACAM1-SHP-1 interaction results observed by PLA and IP experiments. Unlike the robust early CEACAM1-SYK interaction consistently observed using both PLA and IP techniques, the CEACAM1-SHP-1 interaction, although detectable by both methods, does not show clear kinetic patterns. In our earlier response to this issue, we speculated that the limited sensitivity of IP, among other factors, might account for this. However, as the Reviewer rightly questioned this explanation, we acknowledge additional potential reasons, including limitations of the SHP-1 antibody, interference of the interaction with the antibody's epitope, or unknown components within the complex affecting antibody recognition. It is also possible that this interaction is highly sensitive to minor experimental variations.

Despite the lack of definitive evidence regarding CEACAM1-SHP-1 interaction kinetics, our key conclusion remains unchanged: the observed increased SYK phosphorylation following anti-IgM treatment reflects a positive signaling outcome driven by CEACAM1 expression in JEKO-1 and MINO cells. This contrasts with the inhibitory role of CEACAM1 when introduced into Z-138 cells, where it suppresses BCR signaling. Accordingly, we have revised our conclusions to reflect these limitations in interpreting the data in **Fig. 5c, d** and **Fig. 6a** on **page 14 and 15** as follows:

“However, while the early SYK kinetics reached statistical significance, the slower SHP-1 kinetics did not, potentially due to a combination of factors such as limitations of the SHP-1 antibody, epitope masking caused by the interaction itself, or unidentified components within the complex interfering with antibody recognition. It is also possible that this interaction is particularly sensitive to minor experimental variations.”

Except for Fig.5c, 5d, 7a and 7c, no Western blot is quantified! For Fig.7a and 7c statistical analysis based on TWO independent experiments (as stated in the figure legend)! How can you apply a statistical analysis on these data?
???

Response: We are unsure whether the Reviewer received our earlier response to this comment; therefore, we are re-posting it here and welcome any further clarification or suggestions the Reviewer may have.

Signal quantifications have now been included for those blots with missing values.

For the statistical analysis in **Fig. 7a,c**, we used two-way ANOVA to compare two categorical independent variables (that is, two cell lines and time variables). Although our sample sizes are small, with only two replicates that may or may not be statistically significant, the time variables from the two cell lines under comparison are independent, making the comparison between the cell lines statistically significant. We include a few examples of two-way ANOVA tests below. A complete list of all the tests for these figures can be found in the corresponding **Source Data Fig 7** in the **Supplementary Files**.

Table Analyzed					Table Analyzed				
Fig 7A pSYK Y352					Fig 7C-JEKO-1, Z138 SYK				
Two-way RM ANOVA	Matching: Across row				Two-way RM ANOVA	Matching: Across row			
Assume sphericity?	Yes				Assume sphericity?	Yes			
Alpha	0.05				Alpha	0.05			
Source of Variation	% of total variation	P value	P value sum	Significant?	Source of Variation	% of total variation	P value	P value summary	Significant?
Replicates x EV vs 4L	1.047	0.4852	ns	No	Replicates x Jeko-1 vs Z-138	0.5044	0.421	ns	No
Replicates	33.12	0.0917	ns	No	Replicates	1.63	0.3218	ns	No
EV vs 4L	55.9	0.025	*	Yes	Jeko-1 vs Z-138	85.42	<0.0001	****	Yes
Induction time	7.023	0.2926	ns	No	Induction time	8.389	0.1992	ns	No

Table Analyzed					Table Analyzed				
Fig 7A pSYK Y352					Fig 7C-JEKO-1, Z138 SHP-1				
Two-way RM ANOVA	Matching: Across row				Two-way RM ANOVA	Matching: Across row			
Assume sphericity?	Yes				Assume sphericity?	Yes			
Alpha	0.05				Alpha	0.05			
Source of Variation	% of total variation	P value	P value sum	Significant?	Source of Variation	% of total variation	P value	P value summary	Significant?
Replicates x EV vs 4S	10.46	0.014	*	Yes	Replicates x Jeko-1 vs Z-138	32.75	0.009	**	Yes
Replicates	33.94	0.2926	ns	No	Replicates	21.54	0.005	**	Yes
EV vs 4S	21.41	0.0069	**	Yes	Jeko-1 vs Z-138	25.13	0.0159	*	Yes
Induction time	33.88	0.0088	**	Yes	Induction time	6.943	0.7841	ns	No

In summary, regarding the inconsistencies/ weaknesses of the study that compromise two central findings of the study (lipid raft stabilization by CEACAM1, altered signaling molecule interaction depending on CEACAM1 expression level) I recommend rejection of the study.

Response: As discussed above, lipid raft stabilization is not the primary focus of this study, and we have revised the title and abstract accordingly to better reflect this (see below). Our conclusion that CEACAM1 recruits SYK early during IgM stimulation in CEACAM1^{high} cells, thereby enhancing BCR activation, is strongly supported by both PLA and IP approaches, regardless of conclusive evidence concerning SHP-1 recruitment kinetics. In the cellular context of low or absent CEACAM1 expression, our data clearly demonstrate a marked reduction in SYK recruitment by CEACAM1, while SHP recruitment become more predominant. We hope that these robust data and our acknowledgement of the limitations in detecting the CEACAM1-SHP-1 interaction—without any impact on our main conclusion—will support the Reviewer’s consideration for publication.

Title: CEACAM1 as a Mediator of B-cell Receptor Signaling in Mantle Cell Lymphoma

Abstract:

B-cell receptor (BCR) signaling plays an important role in the pathogenesis of mantle cell lymphoma (MCL), but the detailed mechanisms are not fully understood. In this study, through a genome-wide loss-of-function screen, we have identified carcinoembryonic antigen-related cell adhesion molecule 1 (CEACAM1) as an essential factor in a subset of MCL tumors. Our signal transduction studies revealed that CEACAM1 plays a critical role in BCR activation through involvement in two dynamic processes. First, following BCR engagement, CEACAM1 co-localizes to the membrane microdomains (lipid rafts) by anchoring to the F-actin cytoskeleton through the adaptor protein filamin A. Second, CEACAM1 recruits and increases the abundance of SYK in the BCR complex leading to BCR activation. These activities of CEACAM1 require its cytoplasmic tail and the N-terminal ectodomain. Considering that previous studies have extensively characterized CEACAM1 as an ITIM-bearing inhibitory receptor, our findings regarding its activating role are both surprising and context-dependent, which may have implications for BCR-targeting therapies.